# AI as Humanity's Salieri:
# Quantifying Linguistic Creativity of Language Models via Systematic Attribution of Machine Text against Web Text

**Ximing Lu**[♡♠]       **Melanie Sclar**[♡]       **Skyler Hallinan**[♡]       **Niloofar Mireshghallah**[♡]
**Jiacheng Liu**[♡♠]   **Seungju Han**[♠]   **Allyson Ettinger**[♠]   **Liwei Jiang**[♡]   **Khyathi Chandu**[♠]
**Nouha Dziri**[♠]       **Yejin Choi**[♡]

[♡]University of Washington       [♠]Allen Institute for Artificial Intelligence
{lux32,yejin}@cs.washington.edu

## Abstract

Creativity has long been considered one of the most difficult aspect of human intelligence for AI to mimic. However, the rise of Large Language Models (LLMs), like ChatGPT, has raised questions about whether AI can match or even surpass human creativity. We present Creativity Index as the first step to quantify the linguistic creativity of a text by reconstructing it from existing text snippets on the web. Creativity Index is motivated by the hypothesis that the seemingly remarkable creativity of LLMs may be attributable in large part to the creativity of human-written texts on the web. To compute Creativity Index efficiently, we introduce DJ Search, a novel dynamic programming algorithm that can search verbatim and near-verbatim matches of text snippets from a given document against the web. Experiments reveal that the Creativity Index of professional human authors is on average 66.2% higher than that of LLMs, and that alignment reduces the Creativity Index of LLMs by an average of 30.1%. In addition, we find that distinguished authors like Hemingway exhibit measurably higher Creativity Index compared to other human writers. Finally, we demonstrate that Creativity Index can be used as a surprisingly effective criterion for zero-shot machine text detection, surpassing the strongest existing zero-shot system, DetectGPT, by a significant margin of 30.2%, and even outperforming the strongest supervised system, GhostBuster, in five out of six domains.

## 1 Introduction

Creativity has long been considered one of the most challenging "holy grail" of human intelligence for AI to mimic (Hasselberger & Lott, 2023). However, Large Language Models (LLMs) such as ChatGPT have taken the world by storm with their creative power. From generating poetry (Sawicki et al.; Deng et al., 2024b; Sawicki et al., 2023) and composing music (Ding et al., 2024; Deng et al., 2024a; Liang et al., 2024) to designing artwork (Makatura et al., 2024; Jignasu et al., 2023; Lim et al., 2024) and crafting compelling narratives (Yuan et al., 2022; Mirowski et al., 2023a; Ippolito et al., 2022), LLMs take only seconds to produce outputs that would rival or even surpass the work of human creators. This proficiency has even sparked a growing trend of using LLMs for content creation in industrial settings. For example, major studios in Hollywood have integrated LLMs into production processes such as movie scriptwriting (Carnevale, 2023). While studio executives are optimistic about using LLMs to streamline production and reduce costs, Hollywood writers are deeply concerned about being replaced by the rapid integration of LLMs in the industry, leading to a five-month writers' strike (Koblin & John, 2023).

While science fiction writer Ted Chiang characterizes LLMs as a blurry JPEG of the web (Hubert et al., 2024), many others wonder whether AI can indeed match or surpass the creativity of humanity. After all, LLMs have consumed orders of magnitude more works of writing than any single human

could ever read, thus it may seem possible that LLMs could consequently reach a new level of literary sophistication and creativity beyond that of humanity at large.

To answer this question, the first step is to assess the level of creativity in machine texts compared to human texts. Creativity is a complex and ambiguous process that is challenging to define and quantify (Csikszentmihalyi, 1997; Glaveanu et al., 2020; Eagleman & Brandt, 2017; Paeth). Several previous studies have attempted to quantify creativity in writing by developing specific rubrics and asking human evaluators to score the writing based on these criteria. Vaezi & Rezaei (2018) developed a comprehensive rubric to assess fiction writing, while Biggs & Collis (1982) used a taxonomy of structural complexity to categorize creative writing. More recently, Chakrabarty et al. (2024) applied the Torrance Test of Creative Thinking to evaluate the creativity of short stories generated by LLMs in terms of fluency, flexibility, originality and elaboration. While these rubric-based methods are valuable, scaling them up to evaluate large amounts of texts generated by LLMs is impractical due to the reliance on human evaluators.

In this work, we propose CREATIVITY INDEX, a novel statistical measure of creativity in text. The key intuition underlying CREATIVITY INDEX is to quantify the degree of linguistic creativity of a given text by reconstructing that text via mixing and matching of a vast amount of existing text snippets on the web (See Figure 1a; 24 additional examples in Appendix Fig. 5 to Fig. 30). The underlying premise of our work is that the seemingly remarkable creativity of LLMs may be in large part attributable to the remarkable creativity of human-written texts on the web. This contrasts with distinguished human authors such as Hemingway, whose original content and unique writing style cannot be easily replicated by simply assembling snippets from other works. To test this, we provide a novel computational approach to systematically attribute machine text to web texts. Specifically, we introduce DJ SEARCH,[1] a novel dynamic programming algorithm that can efficiently search for verbatim and near-verbatim matches of text snippets from a given document against the web. Here, near-verbatim matches are defined as close paraphrases, characterized by high semantic similarity. Our algorithm combines strict verbatim matching using Infini-gram (Liu et al., 2024), which allows for fast retrieval of any existing sequence of words, with near-verbatim semantic matching achieved through a novel application of Word Mover's Distance (WMD) (Kusner et al., 2015) computed on the word embeddings of text snippets.

The contribution of our work is threefold: First, we introduce the CREATIVITY INDEX to reveal novel insights about machine creativity and human creativity. We find that the CREATIVITY INDEX of human authors—specifically professional writers and historical figures—is on average 66.2% higher than that of LLMs. This creativity gap is consistent across various domains—novel snippets, modern poems, and speech transcripts—at both verbatim and semantic levels. Moreover, we notice that Reinforcement Learning from Human Feedback (RLHF), a widely used alignment method, dramatically reduces the CREATIVITY INDEX of LLMs, by an average of 30.1%. This reduction is more significant at the verbatim level than the semantic level, indicating that LLMs may have converged to certain linguistic style preferred by humans during alignment. Furthermore, we explore creativity differences among various groups of humans. Despite in-group variance, famous authors of classic literature, like Hemingway and Dickens, exhibit the highest levels of creativity, consistent with their levels of renown.

Second, we introduce DJ SEARCH as an efficient algorithmic tool to trace the usage of existing text snippets from the web that LLMs incorporate to compose new generations. The power of LLMs arises from training exhaustively on existing human-written texts on the web, and it is meaningful to trace back and acknowledge the human writers whose work empowers these models' outputs—just as we credit original composers when enjoying a DJ's remix.

Finally, we demonstrate a novel use of CREATIVITY INDEX as a surprisingly effective criterion for zero-shot black-box machine text detection. Our method is ready to deploy out-of-the-box, requiring no training or prior knowledge of the text generator. It not only surpasses the strongest zero-shot baseline, DetectGPT (Mitchell et al., 2023a), by a significant margin of 30.2%, but also outperforms the strongest supervised baseline, GhostBuster (Verma et al., 2024)—which requires expensive data collection for supervised training—in five out of six domains.

We believe that our study will enhance the understanding of LLMs and guide informed usage of content created by LLMs, by providing an interoperable and scalable measurement to assess creativity

---

[1] The name DJ SEARCH is inspired by the way a DJ creates a remix by blending pieces of existing music.

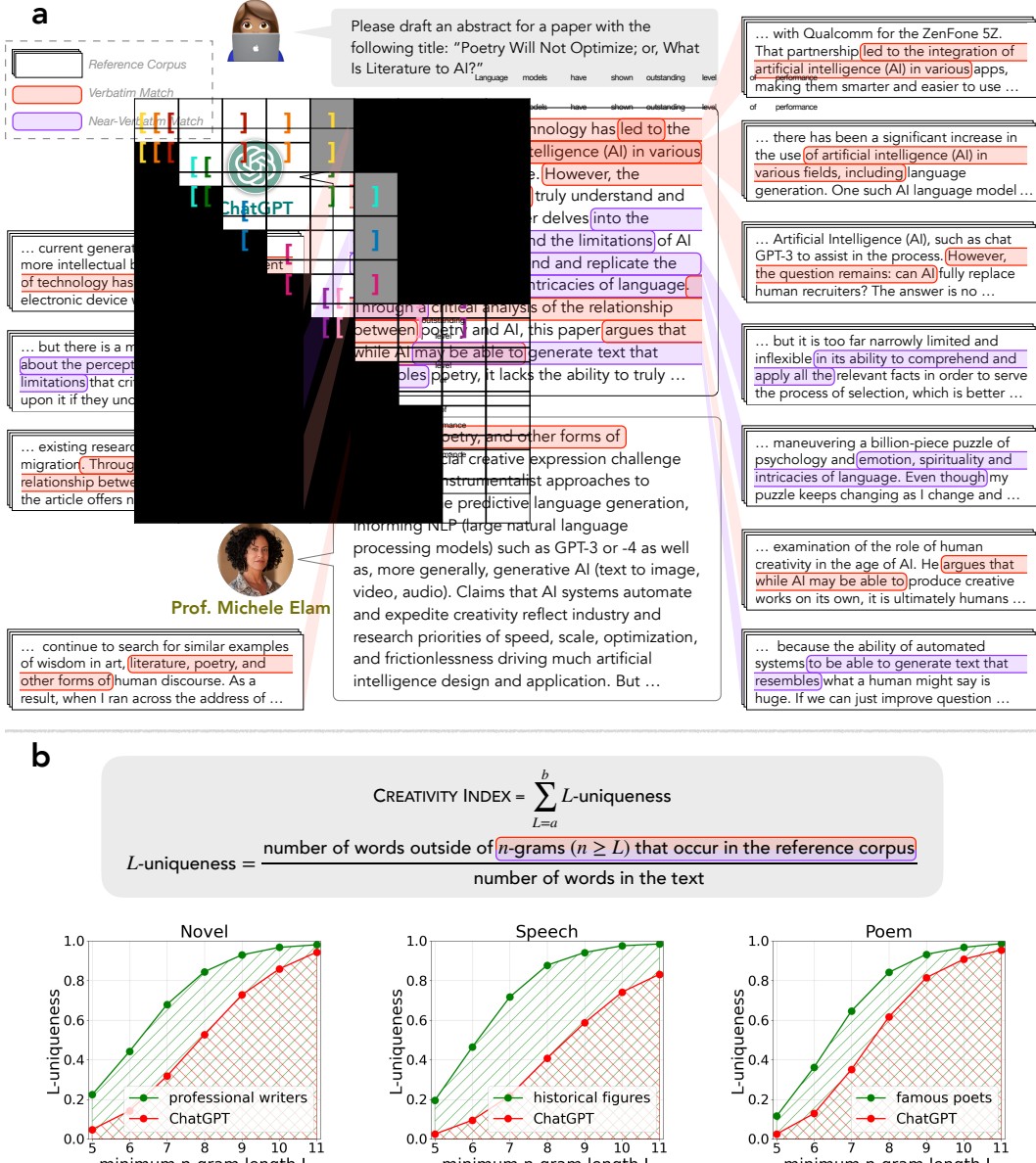

Figure 1: **a**: **Example outputs from DJ SEARCH.** We asked ChatGPT to generate an abstract based on the title of Prof. Michele Elam's paper, "Poetry Will Not Optimize; or, What Is Literature to AI?" (Elam, 2023) The abstract generated by ChatGPT contains significantly more verbatim and near-verbatim matches with existing texts on the web compared to the original abstract written by Prof. Elam. **b**: **Definition of CREATIVITY INDEX.** CREATIVITY INDEX is mathematically equivalent to the area under the $L$-uniqueness curve across a range of minimum $n$-gram lengths $L$. The $L$-uniqueness of ChatGPT is noticeably lower than that of proficient human writers across various context granularities (i.e., $n$-gram lengths) in all domains, leading to a significantly higher CREATIVITY INDEX for human writers compared to ChatGPT.

in machine texts. Additionally, we hope that the out-of-the-box machine text detection enabled by the CREATIVITY INDEX can empower individuals to discern between human texts and machine texts, fostering a more informed and critical engagement with information in the digital age.

## 2 METHOD

**CREATIVITY INDEX**  The key intuition underlying CREATIVITY INDEX is to quantify the degree of linguistic creativity of a given text by estimating how much of that text can be reconstructed by mixing and matching a vast amount of existing text snippets on the web, as shown in Figure 1a. Specifically, CREATIVITY INDEX assesses the extent to which the content of the text can be traced back to similar or identical contexts found in other existing texts. This metric is grounded in the notion of originality from creative thinking in psychology literature, which is defined as the statistic rarity of a response or an idea (Torrance, 1966; Crossley et al., 2016).

Concretely, let $\mathbf{x}$ be a text whose creativity we aim to quantify, such as a speech transcript or a poem, either human written or machine generated. Let an $n$-gram of $\mathbf{x}$ be any contiguous sequence of $n$ words of $\mathbf{x}$, and let $\mathbf{x}_{i:i+n}$ be the $n$-gram of $\mathbf{x}$ starting in the $i$-th word. Let $C$ be a massive reference corpus of publicly available texts on the web , and let $f$ be a binary function that determines whether an $n$-gram $\mathbf{x}_{i:i+n}$ occurs anywhere in the corpus $C$. We define the $L$-uniqueness of a text $\mathbf{x}$ as the proportion of words $w \in \mathbf{x}$ such that none of the $n$-grams in $\mathbf{x}$ that include $w$ occur in the corpus $C$ for $n \geq L$—denoted uniq$(\mathbf{x}, L)$. Intuitively, $L$-uniqueness measures the proportion of $\mathbf{x}$'s words that are used in novel contexts (here, $n$-grams), unseen across a vast text collection $C$. Thus, a higher $L$-uniqueness implies a higher level of originality of $\mathbf{x}$. Formally, uniq$(\mathbf{x}, L) = \sum_{k=1}^{\|\mathbf{x}\|} \mathbb{1}\{f(\mathbf{x}_{i:i+n}, C) = 0 \ \forall i \in (k-n, k], \ n \geq L\}/\|\mathbf{x}\|$, where trivially uniq$(\mathbf{x}, L) \in [0, 1]$.

Note that when fixing $\mathbf{x}$, the function uniq$(\mathbf{x}, L)$ is monotonically increasing as $L$ grows. Its improper integral—$\sum_{n \geq L}$ uniq$(\mathbf{x}, n)$—is an indicator of the overall uniqueness of $\mathbf{x}$ across various context granularities (i.e., $n$-gram lengths), and because of uniq$(\mathbf{x}, L)$'s monotonicity it indirectly measures uniqueness growth speed. We thus define CREATIVITY INDEX as $\sum_{n \geq L}$ uniq$(\mathbf{x}, n)$, with higher CREATIVITY INDEX indicating greater linguistic originality with respect to the corpus $C$, as shown in Figure 1b.

When a text $\mathbf{x}$ is part of the reference corpus $C$, its CREATIVITY INDEX would trivially become zero. This issue often arises with works from famous authors, as their writings are widely available online. To address this, for human texts written before the cutoff date of the reference corpus, we exclude any document $\mathbf{d} \in C$ that contains copies, quotations, or citations of $\mathbf{x}$ and compute CREATIVITY INDEX using this filtered corpus, detailed in Appendix A.3.

**DJ SEARCH**  To enable the use of our CREATIVITY INDEX it is vital to compute it efficiently. For the efficient computation, we introduce DJ SEARCH, a dynamic programming algorithm designed to radpily identify the set of all $\mathbf{x}$'s $n$-grams ($n \geq L$) that occur in the corpus $C$.

A brute force approach would independently check if every $n$-gram of $\mathbf{x}$ occurs in $C$, performing a quadratic number of $f$ evaluations with respect to $\mathbf{x}$'s length, and thus making it too computationally expensive. Instead, we design a two-pointer method (Laaksonen, 2020) that takes only a linear number of $f$ evaluations, as illustrated in Figure 2. The key idea is to reduce finding all $n$-grams occurring in $C$ to identifying the longest $n$-gram occurring in $C$ starting at each index $i$: once those have been found, it is trivial to deduce all the $n$-gram occurring in $C$ by computing their subsequences. Concretely, we progressively analyze the whole document $\mathbf{x}$ by iteratively searching for the longest $n$-gram that starts at each index $i$ and occurs in $C$, using $f$ as the assessment. Once we have found such longest $n$-gram starting at $i$, we crucially reuse computations for $i + 1$ by noting that $f(\mathbf{x}_{i:i+n}, C) = 1$ implies $f(\mathbf{x}_{i+1:i+n}, C) = 1$. Thus, we always analyze $n$-grams starting and/or ending at a later endpoint than before, which upper bounds the number of analyzed $n$-grams (i.e., the number of $f$ calls) to at most $2\|\mathbf{x}\|$. The implementation is detailed in Appendix A.1.

In addition to minimizing the number $f$ evaluations, DJ SEARCH optimizes the time complexity of each evaluation. $f$ determines whether a $n$-gram $\mathbf{x}_{i:i+n}$ occurs in the corpus $C$ either exactly or in a semantically similar way—e.g., a paraphrase of $\mathbf{x}_{i:i+n}$ exists in $C$. Semantic similarity is often computed using text embeddings, which are fixed-length vector representations of text meanings. This reduces measuring text similarity to computing vector distance. Text embeddings, typically

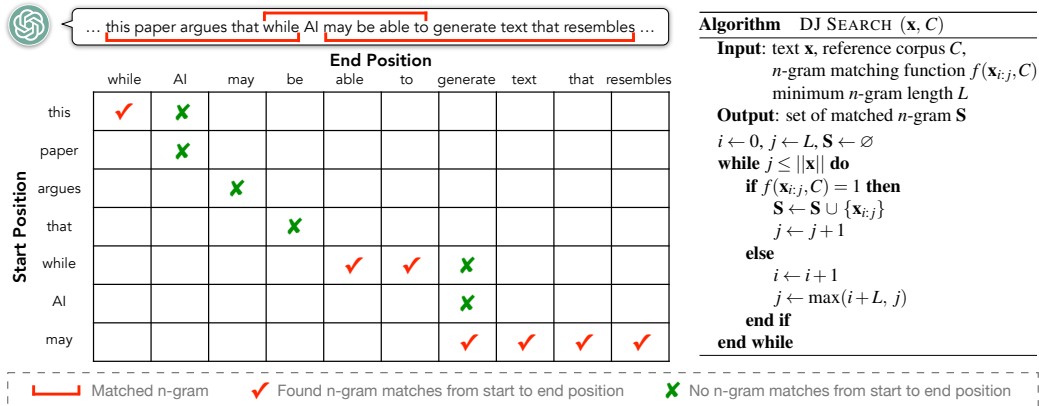

Figure 2: **An illustration of DJ Search algorithm.** A brute force approach would independently check if every $n$-gram of $\mathbf{x}$ occurs in $C$, performing a quadratic number of $f$ evaluations with respect to $\mathbf{x}$'s length (i.e., checking every cell in the grid). DJ SEARCH is a two-pointer method that takes only a linear number of $f$ evaluations. By progressively analyzing $n$-grams starting and/or ending at a later endpoint than before, DJ SEARCH limits the total number of $f$ evaluations to $2||\mathbf{x}||$. In this example, the minimum $n$-gram length $L$ is set to 5.

generated by complex models (e.g., BERT (Devlin et al., 2019), RoBERTa (Liu et al., 2019), Span-BERT (Joshi et al., 2020)) lack linearity, requiring independent computation for each $n$-gram in $\mathbf{x}$ and $C$. To alleviate this issue we use Word Mover's Distance (WMD) (Kusner et al., 2015), an optimal transport-inspired metric that measures distance between two $n$-grams by combining word embedding distances between each $n$-gram's words. WMD enables optimizing $f$'s computation, as pairwise distances between word embeddings can be pre-computed for every pair of words, and then be reused in every function call of $f$ to identify $n$-grams in $C$ that are semantically similar to the ones in $\mathbf{x}$. The implementation is detailed in Appendix A.2.

To further boost efficiency, and given that occurrences of $\mathbf{x}_{i:i+n}$ are more likely in texts similar to $\mathbf{x}$, we estimate $f$ by computing WMD only for the texts in $C$ most similar to $\mathbf{x}$, as identified by BM25 (Robertson & Walker, 1994). Moreover, exact occurrences of $\mathbf{x}_{i:i+n}$ in $C$ represent a less costly special case in computing $f$. We further optimize $f$'s computation by using Infini-gram (Liu et al., 2024), which finds exact matches of $\mathbf{x}_{i:i+n}$ in $C$ in milliseconds; WMD is computed only if no matches are found by Infini-gram.

## 3 EVALUATION

**How does the creativity of language models compare to humans?** We compute the CREATIVITY INDEX for machine texts and human texts across three creative writing tasks: novel writing, poetry composition, and speech drafting. For human texts, we use book snippets in the Book-MIA (Shi et al., 2024) dataset, popular modern poems collected by PoemHunter.com, and famous speeches from the American Rhetoric speech bank. For machine texts, we prompt LLMs to generate several paragraphs of novels, poems, or speeches, starting with an initial sentence from existing human writings in each category (see Appendix B.1 for details). We experiment with state-of-the-art LLMs, including GPT-3 (Brown et al., 2020), ChatGPT (Ouyang et al., 2022), LLaMA 2 Chat (Touvron et al., 2023), Tulu 2 (Ivison et al., 2023), and OLMo Instruct (Groeneveld et al., 2024). For open-source and open-weight models, we use the largest model size available from each model family. We use RedPajama (Computer, 2023), a large-scale English corpus with 900 million web documents, as the reference corpus. The models we analyze are primarily pre-trained on the web data available before the cutoff date of the reference corpus RedPajama. We will discuss later how to handle newer models, such as GPT-4 (OpenAI et al., 2023), given that it was largely trained on more recent web data and third-party private data, both of which fall outside the reference corpus. We restrict the matching criteria to verbatim matches only in the first experiment. We will ablate the effect of different matching criteria, prompt formats, decoding strategies, context length, and model sizes in later experiments.

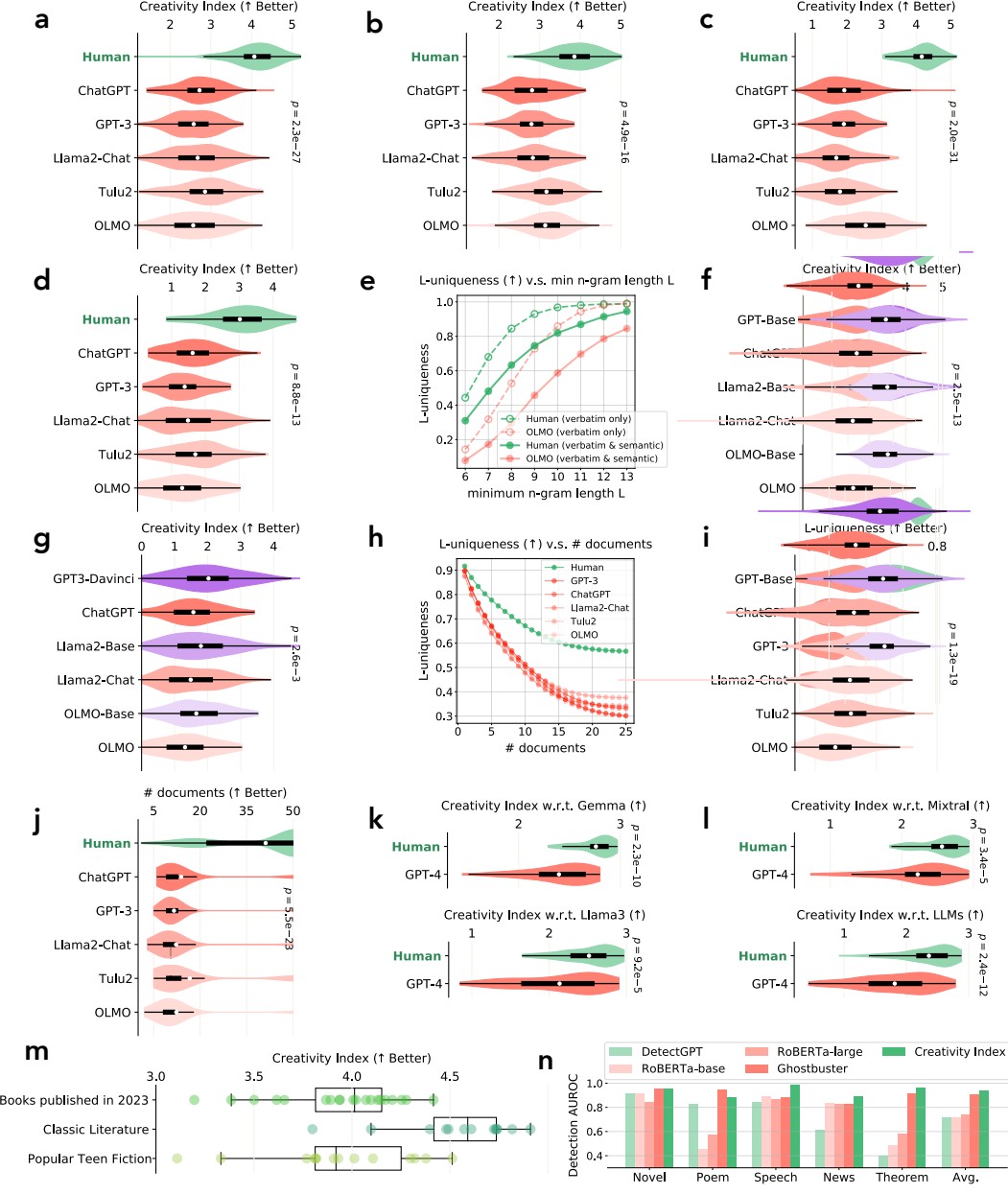

Figure 3: **a-c**: CREATIVITY INDEX in novel writing (**a**), poetry composition (**b**) and speech writing (**c**) based solely on verbatim matches. **d**: CREATIVITY INDEX in novel writing considering both verbatim and semantic matches. **e**: $L$-uniqueness in novel writing with respect to the minimum $n$-gram length $L$ for humans and OLMo. **f-g**: CREATIVITY INDEX of LLMs before and after RLHF in novel writing, based solely on verbatim matches (**f**) and based on both verbatim and semantic matches (**g**). **h**: $L$-uniqueness in novel writing with respect to number of documents in the reference corpus. **i**: $L$-uniqueness when search over the top 50 documents in novel writing. **j**: The number of reference documents required to keep $L$-uniqueness below 50% in novel writing. **k-l**: CREATIVITY INDEX of GPT-4 compared to humans in novel writing based on verbatim matches, using a machine-generated reference corpus sourced from the instruction-aligned version of Gemma-7B, Llama3-8B, and Mixtral-7B, as well as a combination of all three. **m**: CREATIVITY INDEX of different groups of human writers. **n**: Detection AUROC across various domains: our approach sets a new state-of-the-art for zero-shot detection, even surpassing supervised baselines.

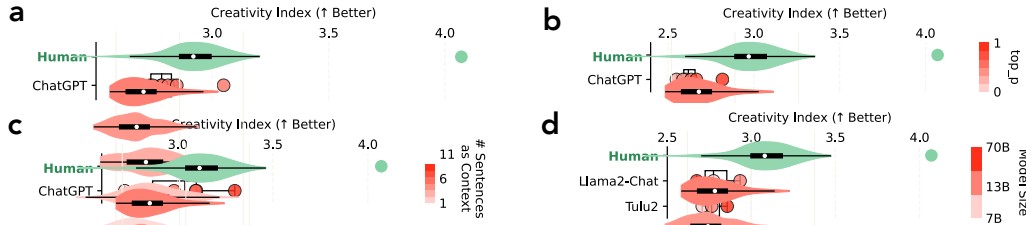

Figure 4: **a-c**: CREATIVITY INDEX of ChatGPT in novel writing based on verbatim matches, with different prompt formats (**a**), $p$ values in top-p decoding (**b**) and prompt length (**c**). **d**: CREATIVITY INDEX of LLaMA 2 Chat and Tulu 2 with different model sizes.

Our primary finding is that *humans consistently exhibit a much higher level of creativity compared to any LLM across all tasks* (Fig. 3**a-c**). Averaged across all models, the CREATIVITY INDEX of humans is 52.2% higher[2] than LLMs in novel writing ($p = 6.9 \times 10^{-27}$, by Mann-Whitney U test unless otherwise specified; $N = 600$), 31.1% higher in poetry composition ($p = 1.5 \times 10^{-15}$; $N = 600$) and 115.3% higher in speech drafting ($p = 6.1 \times 10^{-31}$, $N = 600$). This suggests that human writings are composed of far more unique combinations of words and phrases compared to model generations. On the other hand, the differences in model creativity are much smaller and show very low statistical significance ($p = 0.09$; $N = 1500$).

Furthermore, we experiment with different prompt formats on top of ChatGPT, intentionally encouraging creativity in the model's generations by incorporating instructions such as 'push for creative ideas, unique emotions, and original twists,' 'be bold and creative,' or 'you are a creative writer.' (Fig. 4**a**) For a full list of the prompts we used, please see Appendix B.1. We found that the difference in the CREATIVITY INDEX of ChatGPT across different prompts is minimal, with no statistical significance ($p = 0.23$; $N = 600$). We also experimented with different decoding strategies by varying the $p$ value in top-p decoding (Fig. 4**b**). Although a higher $p$ value resulted in a marginally higher CREATIVITY INDEX, the difference was minimal and not statistically significant ($p = 0.23$; $N = 600$). Moreover, we ablate the effect of prompt length by varying the number of sentences from human writings included in the prompt (Fig. 4**c**). We found that longer prompts tended to result in a slightly higher CREATIVITY INDEX, likely due to the model copying more from the longer human text in the prompt. However, the statistical significance of these differences is very low ($p = 0.13$; $N = 600$). Lastly, we analyze the effect of different model sizes for LLaMA 2 Chat and Tulu 2, but do not observe a consistent trend ($p = 0.12$; $N = 600$) (Fig. 4**d**).

**How do different matching criteria affect creativity measurement?** We experiment with restricting valid matches to verbatim only, and with allowing both verbatim and semantic matches. First, *the creativity gap between humans and LLMs becomes even larger when considering semantic matches in addition to verbatim matches* (Fig. 3**d**). Averaged across all models, the CREATIVITY INDEX of human, based on both verbatim and semantic matches, is 102.5% higher than LLMs in novel writing ($p = 2.6 \times 10^{-12}$; $N = 600$), whereas based on verbatim matches alone, the CREATIVITY INDEX of human is 52.2% higher than LLMs. Second, *semantic matches provide more signal for analyzing the uniqueness of longer n-grams* (Fig. 3**e**). For example, while the gap in $L$-uniqueness at $L = 11$ between human text and machine text from OLMo Instruct is 3.7% based on verbatim matches alone, this gap widens to 16.3% when considering both verbatim and semantic matches ($p = 3.1 \times 10^{-7}$; $N = 600$). This indicates that although some of the longer $n$-grams in machine text may appear unique at the verbatim level, they are similar to certain text snippets in the reference corpus at the content level.

**What impact does RLHF have on model creativity?** RLHF aims to align model's outputs with human preferences, enhancing LLMs' ability to follow instructions and improving their safety and adaptability. To understand the impact of RLHF on model creativity, we compare the CREATIVITY INDEX of the LLMs before and after RLHF alignment. Specifically, we experiment with GPT Base (Brown et al., 2020), LLaMA 2 Base (Touvron et al., 2023), and OLMo Base (Groeneveld et al., 2024) and compare their creativity with their counterparts post-RLHF alignment. Our main

---

[2]The percentage difference computed using the formula: $\frac{\text{CREATIVITY INDEX (human)} - \text{CREATIVITY INDEX (model)}}{\text{CREATIVITY INDEX (model)}}$

finding is that *the* CREATIVITY INDEX *of models after RLHF alignment is much lower than those before RLHF* (Fig. 3**f**-**g**). Based on verbatim match alone, the CREATIVITY INDEX of LLMs reduces by an average of 30.1% after RLHF ($p = 1.3 \times 10^{-12}$; $N = 600$). Based on both verbatim and semantic matches, the CREATIVITY INDEX of LLMs decreases by an average of 8.9% after RLHF ($p = 0.01$; $N = 600$). We notice that *the reduction of* CREATIVITY INDEX *after RLHF is noticeably larger when considering verbatim matches alone.* We speculate that models might have learned certain linguistic styles preferred by humans during RLHF, leading to a decreased surface form diversity in its outputs.

**How do overlapped n-grams distribute in the reference corpus?**  In addition to measuring the amount of matched $n$-grams in a given text, we also investigate the distribution of these $n$-grams in the reference corpus. We aim to understand whether these matched $n$-grams are spread across many documents or concentrated in a few. Specifically, we identify the top $N$ documents that contain the highest amount of matched $n$-grams and result in the minimum $L$-uniqueness for a given text. This problem can be reduced to the maximum coverage problem (Nemhauser et al., 1978) and approximated using a greedy algorithm. Here, we consider both verbatim and semantic matches. Our main finding is that *the matched $n$-grams in machine texts are concentrated in fewer documents compared to human texts* (Fig. 3**h**-**j**). When searching over the top 50 documents, the averaged $L$-uniqueness ($L = 5$) for machine texts is 32.8%, which is 73.4% lower than human texts (mean: 56.6%; $p = 3.9 \times 10^{-19}$; $N = 600$). Conversely, keeping $L$-uniqueness below 50% requires searching through an average of 41.2 documents for human texts, which is 213.7% more than for machine texts (mean: 13.4; $p = 1.6 \times 10^{-22}$; $N = 600$). This implies that it's more likely to find some existing documents resemble models' generations than human writings.

**How to measure creativity in LLMs trained on data outside of the reference corpus?**  The CREATIVITY INDEX of GPT-4 would be significantly inflated if computed using the RedPajama corpus, as RedPajama's cutoff date is two years earlier than GPT-4's knowledge cutoff, and GPT-4 is additionally trained on third-party private data that we don't have access to. We hypothesize that LLMs pre-trained on similar web data are likely to memorize and replicate similar patterns. As a result, when comparing the generations of these models, we expect them to be more similar to each other than to human texts, which often contain long-tail patterns. Therefore, to compare the creativity level of GPT-4 with humans, we use a model-generated reference corpus from newer open-weight models with knowledge cutoff dates similar to GPT-4, including the instruction-aligned versions of Gemma-7B (Team et al., 2024), Llama3-8B (AI@Meta, 2024), and Mixtral-7B (Jiang et al., 2023). Specifically, we randomly sample 150k sentences from the RedPajama corpus and prompt these models to generate document-level continuations. *Based on the model-generated reference corpus, the average* CREATIVITY INDEX *of humans is 30.3% higher than GPT-4 in novel writing* ($p = 2.3 \times 10^{-12}$; $N = 600$) (Fig. 3**k**-**l**). This suggests that while newer LLMs like GPT-4 may appear more creative when compared to public data, they still learn common patterns from their private training data and tend to emit similar patterns as other LLMs trained on comparable data.

**How does the creativity vary among different groups of human?**  We compare the creativity levels among three categories of writings: books published in 2023 from the BookMIA (Shi et al., 2024) dataset, classic literature by famous authors, and popular young adult fictions, both sampled from Goodreads' book lists. Our main finding is that *classic literature exhibits a higher creativity level than the other two categories* (Fig. 3**m**). The average CREATIVITY INDEX of classic literature is 21.6% higher than young adult fictions ($p = 2.7 \times 10^{-90}$; $N = 3000$), and 13.8% higher than books published in 2023 ($p = 4.3 \times 10^{-120}$; $N = 3000$). Though CREATIVITY INDEX may reflect some inherent differences in creativity, it could also be influenced by factors like writing style and era of composition. For instance, some classic literature is written in older English, making it harder to reconstruct from web texts that primarily use modern English. In addition to the differences across categories, we also observed noticeable variance in creativity within each category. For example, the CREATIVITY INDEX of 'The Hunger Games' is 35.4% higher than 'Twilight' ($p = 1.5 \times 10^{-19}$; $N = 200$), even though both books belong to the category of popular young adult fiction.

**Can we leverage differences in creativity for detecting machine-generated text?**  Based on the creativity difference between humans and LLMs, we propose to use CREATIVITY INDEX as a criterion for zero-shot black-box machine text detection. Texts with higher creativity are more likely

to be written by human. Our approach is ready to deploy out-of-the-box, requiring no training or prior knowledge of the text generator. In addition to creative writing tasks, we also test our method on detecting machine-generated fake news and theorem proofs. Detecting fake news is crucial for protecting the public from misinformation, while identifying model-generated solutions is important for regulating students' use of LLMs in their coursework. To obtain additional test data, we prompt LLMs to generate news articles based on the fake news headlines from the Misinfo Reaction Frames (Gabriel et al., 2022) and compare them with the real news articles from the XSum (Narayan et al., 2018) dataset. Meanwhile, we prompt LLMs to generate proofs for theorems from the NaturalProofs (Welleck et al., 2022) benchmark, and compare them with the ground-truth human-written proofs. The baselines we compare against includes the state-of-the-art zero-shot detector, Detect-GPT (Mitchell et al., 2023a), which uses the curvature of log probability as the detection criterion, as well as several supervised methods. These include OpenAI's RoBERTa-based detector, fine-tuned on millions of generations from various GPT-2 sized models, and the state-of-the-art supervised detector, Ghostbuster (Verma et al., 2024), fine-tuned on thousands of generations from ChatGPT. We measure performance using the area under the receiver operating characteristic curve (AUROC), which represents the probability that a classifier correctly ranks a randomly-selected human-written example higher than a randomly selected machine-generated example. Our method achieves new state-of-the-art performance in zero-shot detection: it consistently surpasses DetectGPT and OpenAI's detector across all domains, with significant improvements in AUROC—30.2% and 26.9%, respectively. It also outperforms the strongest supervised baseline, Ghostbuster—which requires expensive training and data collection—in five out of six domains, achieving an average AUROC improvement of 3.5% (Fig. 3n).

## 4  DISCUSSION

This work investigates the level of linguistic creativity in texts generated by LLMs and written by humans. Our findings suggest that the content and writing style of machine-generated texts may be less original and unique, as they contain significantly more semantic and verbatim matches with existing web texts compared to high-quality human writings. We hypothesize that this limited creativity in models may result from the current data-driven paradigm used to train LLMs. In this paradigm, models are trained to mimic human-written texts during the pre-training stage, and to produce outputs aligned with human preferences during the RLHF stage. As a result, models learn to generate fluent and coherent texts by absorbing and replicating common patterns observed in their training data. This reliance on existing text patterns can restrict their originality, as their outputs are inherently shaped by previously seen examples. In contrast, accomplished authors such as Hemingway go beyond simply mimicking the great writings of others; they craft their own narratives to express their unique opinions, perspectives, and insights, drawing from their personal experiences, emotions, and backgrounds, which translates to the more creative compositions of words and phrases that our method detects. Just as a DJ remixes existing tracks while a composer creates original music, we speculate that LLMs behave more like DJs, blending existing texts to produce impressive new outputs, while skilled human authors, similar to music composers, craft original works.

This work also faces the following limitations. Firstly, the computation of the CREATIVITY INDEX is constrained by the reference corpus used for DJ SEARCH. While open-source LLMs such as OLMo rely on publicly available texts from the internet for their training data, major companies like OpenAI additionally curate private data to train their closed-source LLMs such as ChatGPT. Without incorporating these private data into the reference corpus of DJ SEARCH, the CREATIVITY INDEX of closed-source LLMs may be somewhat inflated. Secondly, the overlap with existing texts identified by DJ SEARCH in models' generations may not conclusively indicate memorization of a specific document. It's possible that these text fragments, or their variations, appear in multiple documents that the model has been trained on, including those outside the reference corpus of DJ SEARCH. Thirdly, the human authors that this work focuses on are those with relatively high-quality writings available in existing public datasets. While some human writings can be mediocre, tedious and unoriginal, we aim to assess how the creativity levels in impressive LLM outputs compare against the high-quality writings produced by professional human authors. Lastly, we acknowledge that the discussion surrounding the use of LLMs in social and industrial settings is highly complex, and our work here speaks only to a part of it. Besides the creativity of machine-generated content, other considerations in this discussion include socioeconomic factors and ethical implications, which fall beyond the scope of this paper.

## 5 RELATED WORK

**Measuring Creativity in Ideas:**   Measuring creative thinking and problem solving takes root in early work in psychology (Torrance, 1966), where researchers defined four pillars for creative thinking: fluency, flexibility, originality and elaboration. Crossley et al. (2016) later on developed this notion and built on it to expand this to measuring creative writing in students, where they also adopted $n$-gram novelty for a measure of originality. However, these prior work focus on creativity in humans, and they also do not introduce any automated metrics or measurements.

**Measuring Creativity in Machine-generated Text Using Expert Annotators:**   Closely related to CREATIVITY INDEX is a recent line of work in the generative AI literature comparing the creativity of human writers to that of large language models in different domains such as story telling and journalism (Chakrabarty et al., 2023; 2024; Anonymous, 2024). Similar to us, the approach in this direction often involves prompting an LLM to write an original story or news article, based on some existing premise or press release, and then comparing the machine-generated text to the human-written counterparts. These works, however, take a rather subjective approach, where they define and measure creativity based on human expert annotations and whether people perceive the text to be more creative, rather than an objective quantification of novelty that we provide.

**Measuring Novelty of $N$-grams:**   Finally, closely related to our work in terms of techniques is Nguyen (2024) and Merrill et al. (2024). The former attempts at finding $n$-gram rules that would cover and predict generations from transformer models, showing that more than $70\%$ of the times transformers follow some pre-set patterns and rules. The latter is more similar to our work as they also measure the novelty of generated $n$-grams and compare it to human-written text, however they differ from us in tow major ways: (1) they only find verbatim matches, whereas we also match to approximate, semantically similar blocks of text and (2) they compute the percentage of $n$-grams of a certain length in a text that can be found in the reference corpus, whereas we measure how much of the text can be reconstructed by mixing and matching a vast amount of existing text snippets of varying lengths from the web.

**Machine Text Detection:**   Detecting machine-generated text has been explored for several years using a variety of methods (Jawahar et al., 2020; Uchendu et al., 2021). Gehrmann et al. (2019) and Dugan et al. (2023) demonstrate that even humans tend to struggle to differentiate between text written by humans and machines, highlighting the need for automated detection solutions. Some approaches involve training a classifier in a supervised manner to identify machine-generated text (Bakhtin et al., 2019; Uchendu et al., 2020), while others use a zero-shot detection method (Solaiman et al., 2019; Ippolito et al., 2020). Additionally, there is research on bot detection through question answering (Wang et al., 2023; Chew & Baird, 2003). Recently, Mitchell et al. (2023b) introduced DetectGPT, a zero-shot method based on the hypothesis that texts produced by a large language model (LLM) are located at local maxima, and thus exhibit negative curvature, in the model's probability distribution. Follow-up work build on DetectGPT by making it faster (Bao et al., 2024) and using cross-detection when the target model is unknown (Mireshghallah et al., 2024).

## 6 CONCLUSION

We introduce CREATIVITY INDEX, an interoperable and scalable metric designed to quantify the linguistic creativity of a given text by estimating how much of that text can be reconstructed by mixing and matching a vast amount of existing text snippets on the web. To efficiently compute the CREATIVITY INDEX, we developed DJ SEARCH, a novel dynamic programming algorithm that can search verbatim and near-verbatim matches of text snippets from a given document against the web. We find that the creativity index of professional human writers is, on average, 66.2% higher than that of LLMs. Notably, RLHF dramatically reduces the creativity index of LLMs by an average of 30.1%. Furthermore, we demonstrate that CREATIVITY INDEX can be used as a surprisingly effective criterion for zero-shot black-box machine text detection. Our method not only surpasses the strongest zero-shot baseline, DetectGPT, by a significant margin of 30.2%, but also outperforms the strongest supervised baseline, GhostBuster, in five out of six domains. We hope that this study enhances the understanding of LLMs through the lens of linguistic creativity, and fosters informed usage of content created by LLMs in real-world applications.

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

# A  METHOD DETAILS

## A.1  IMPLEMENTATION DETAILS OF DJ SEARCH

As discussed in the main text, the deployment of the CREATIVITY INDEX relies on efficiently determining whether each $n$-gram $\mathbf{x}_{i+i+n} \in \mathbf{x}$ can be found anywhere in the massive reference corpus $C$ of publicly available texts. The function $f(\mathbf{x}_{i+i+n}, C)$ is a binary indicator that determines whether an $n$-gram $\mathbf{x}_{i:i+n}$ occurs in $C$. In line with the definition of CREATIVITY INDEX, we only consider the $n$-grams $\mathbf{x}_{i+i+n}$ such that $n \geq L$ for some fixed constant $L$.

While a naive approach to checking whether $\mathbf{x}_{i+i+n}$ appears in $C$ for every $n$-gram $\mathbf{x}_{i+i+n} \in \mathbf{x}$ would take $O(|\mathbf{x}|^2)$ calls[3] to $f$ (see Algorithm 1), using a two-pointer approach we can radically reduce this to $O(|\mathbf{x}|)$ calls (see Algorithm 2). Note that a two-pointer approach does $O(|\mathbf{x}|)$ calls to $f$ since in each iteration we advance at least one of the two pointers $i$ and $j$ by 1, and $0 \leq i, j \leq |\mathbf{x}|$.

---

**Algorithm 1** Naive Computation

---

$\text{NGramsFound}_{i,j} \leftarrow \text{False} \quad \forall \, i \in [0..|\mathbf{x}|) \text{ and } j \in [0..|\mathbf{x}|) \quad \triangleright \text{matrix to store } n\text{-gram occurrence}$
**for** $i \in [0, 1, ..., |\mathbf{x}| - L)$ **do**
    **for** $j \in [i + L, ..., |\mathbf{x}|)$ **do**
        $\text{NGramsFound}(i, j) \leftarrow f(\mathbf{x}_{i:j}, C)$
    **end for**
**end for**
**return** NGramsFound

---

**Algorithm 2** Efficient computation of DJ SEARCH$(\mathbf{x}, C)$

---

$\text{NGramsFound}_{i,j} \leftarrow \text{False} \quad \forall \, i \in [0..|\mathbf{x}|) \text{ and } j \in [0..|\mathbf{x}|) \quad \triangleright \text{matrix to store } n\text{-gram occurrence}$
$i \leftarrow 0, \, j \leftarrow L$
**while** $j < |\mathbf{x}|$ **do**
    $\text{NGramsFound}(i, j) = f(\mathbf{x}_{i:j}, C)$
    **if** $\text{NGramsFound}(i, j)$ **then**
        $j \leftarrow j + 1$                          $\triangleright \text{we will search for } \mathbf{x}_{i:j+1} \text{ next}$
    **else**
        $i \leftarrow i + 1$        $\triangleright \text{since } \mathbf{x}_{i:j} \text{ was not found, } \mathbf{x}_{i:j+k} \text{ will not be found for all } k > 0$
        $j \leftarrow \max(i + L, j)$               $\triangleright \text{we only explore } L\text{-grams and beyond}$
    **end if**
**end while**
**return** NGramsFound

---

## A.2  IMPLEMENTATION DETAILS OF WORD MOVER'S DISTANCE

Let $w$ be an $n$-gram. Let $f(w, C)$ be the function that determines whether $w$ appears in any text $\mathbf{d} \in C$, either exactly or as a phrase that is highly similar in meaning to $w$ (e.g., a paraphrase of $w$). Trivially, $f(w, C) := \bigcup_{\mathbf{d} \in C} f(w, \mathbf{d})$, and here on we will only discuss how to compute $f(w, \mathbf{d})$.

An established approach for finding semantically similar phrases to a given $n$-gram $w$ is to compute its embedding—embedding$(w)$—and then independently compute its similarity to the embeddings of all other $n$-grams to be analyzed. An *embedding* of a $n$-gram is a vector that represents the meaning of such $n$-gram in an $k$-th dimensional space of fixed size, enabling the comparison of similarity between concepts expressed in different surface forms. This comparison is typically done using *cosine similarity*, the scaled dot product between the two embeddings being compared. Text embeddings are generated by models specifically trained to this effect (e.g., BERT (Devlin et al., 2019), RoBERTa (Liu et al., 2019), SpanBERT (Joshi et al., 2020)) making their computation expensive at a large scale. Notably, text embeddings usually do not possess linearity, i.e. the embedding of concatenating $n$-grams $w$ and $v$ cannot be deduced from knowing embedding$(w)$ and embedding$(v)$, and instead needs to be computed from scratch.

---

[3]There are $(|\mathbf{x}| - L)(|\mathbf{x}| - L + 3)/2$ spans to analyze if $L$ is the minimum $n$-gram length to be considered.

Since our goal is to find the $n$-grams of $\mathbf{d}$ that are highly similar to $w$, using the traditional approach would entail comparing embedding($w$) with the embeddings of all $n$-grams in $C$, which are approximately $\sum_{d \in C} |d|^2$ in number. Note that this also implies independently computing $\approx \sum_{d \in C} |d|^2$ embeddings, which increases the computation costs significantly. Instead we use Word Mover's Distance (Kusner et al., 2015) (WMD), a method to estimate similarity between two $n$-grams by combining comparisons between pairs of **_word_** embeddings. This enables lifting the requirement to independently computing the embedding for each $n$-gram in $C$. Concretely, the Word Movers' Distance between two $n$-grams $w$ and $v$ is defined as follows:

$$D_{w \to v} := \frac{1}{|w|} \sum_{i \in [0..|w|)} \min_{j \in [0..|v|)} 1 - \mathrm{cosine\_similarity}(\mathrm{embedding}(v_j), \mathrm{embedding}(w_i))$$

$$= 1 - \frac{1}{|w|} \sum_{i \in [0..|w|)} \max_{j \in [0..|v|)} \mathrm{cosine\_similarity}(\mathrm{embedding}(v_j), \mathrm{embedding}(w_i))$$

$$\mathrm{WMD}(w, v) := \max(D_{w \to v}, D_{v \to w})$$

WMD also pre-filters the words considered in $w$ and $v$ to only include the *content words* in the analysis (i.e, discards *stop-words*, such as *the, a, an, it, on, ...*).

Note that $D_{w \to v}$'s definition is asymmetric ($D_{w \to v} \neq D_{v \to w}$). Thus, we consider the Word Movers' Distance of two $n$-grams $w$ and $v$ as the maximum of $D_{w \to v}$ and $D_{v \to w}$: $w$ and $v$ are highly similar if their distance is below a threshold $\delta$ for both $D_{w \to v}$ and $D_{v \to w}$ (See Algorithm 3):

$$\mathrm{WMD}(w, v) = \max(D_{w \to v}, D_{v \to w}) < \delta$$

---

**Algorithm 3** Conceptual writeup of $f(w, \mathbf{d})$ using Word Mover Distance (WMD) to find the $n$-grams of a single text $\mathbf{d} \in C$ that are highly similar to the $n$-gram $w$ and are of length $\geq L$.

---

  **procedure** DIRECTIONALWMD($w, v$)
    **return** 1 - $\frac{1}{|v|} \sum_{j \in [0..|v|)} \max_{i \in [0..|w|)} \mathrm{cosine\_similarity}(\mathrm{embedding}(w_i), \mathrm{embedding}(v_j))$
  **end procedure**
  **for** $a \in [0, 1, ..., |\mathbf{d}|)$ **do**
    **for** $b \in [a + L, ..., |\mathbf{d}|]$ **do**
      symmetricWMD $\leftarrow \max(\mathrm{directionalWMD}(\mathbf{d}[a : b], w), \mathrm{directionalWMD}(w, \mathbf{d}[a : b]))$
      **if** symmetricWMD $< \delta$ **then**
        **return** True
      **end if**
    **end for**
  **end for**
  **return** False

---

Avid readers may notice that Algorithm 3 repeatedly computes the maximum over the same set, and sums of contiguous similarity scores; these can be pre-computed. Algorithm 4 shows these optimizations, resulting in an algorithm of time complexity $O(|d| \cdot |w| + |d|^2|w|) = O(|d|^2|w|)$, assuming already computed word embeddings. Note that because there is a fixed vocabulary, all word embeddings as well as cosine similarities of word embedding pairs can be pre-computed.

We described how to compute $f(w, \mathbf{d})$ for a single document $\mathbf{d} \in C$, as we have already established that $f(w, C) = \bigcup_{\mathbf{d} \in C} f(w, \mathbf{d})$. To accelerate computation, and given that similar $n$-grams to $\mathbf{x}_{i:i+n}$ are more likely to occur in texts similar to $\mathbf{x}$, we select $C$'s top most likely documents to contain $w$ using a BM25Robertson & Walker (1994) index, denoted $C'$. We then approximate $f(w, C) \approx \bigcup_{\mathbf{d} \in C'} f(w, \mathbf{d})$.

As a final optimization, we note that it is unnecessary to compute the costly $f(w, C)$ for finding semantically similar matches for $w$ in the case where $w$ appears exactly in $C$. To check if $w$ appears exactly in $C$, we can leverage the existing, less expensive approach Infini-Gram (Liu et al., 2024) and search for the semantic similar matches only if Infini-Gram could not find any exact matches.

---

**Algorithm 4** Efficient Computation of $f(w, \mathbf{d})$ (optimization of Algorithm 3)

---

token_similarity$_{i,j}$ ← cosine_similarity(embedding($w_i$), embedding($\mathbf{d}_j$)) $\quad \forall\, i \in [0..|w|)$ and $j \in [0..|\mathbf{d}|)$

**for** $j \in [1, ..., |\mathbf{d}|]$ **do**
 $\quad$ doc_prefix_similarity$_j$ ← doc_prefix_similarity$_{j-1}$ + $\max_{i \in [0..|w|)}$ token_similarity$_{i,j-1}$
**end for**

**for** $a \in [0, 1, ..., |\mathbf{d}|)$ **do**
 $\quad$ **for** $b \in [a + L, ..., |\mathbf{d}|]$ **do**
 $\quad\quad$ computed_WMD($\mathbf{d}[a : b), w$) ← 1 − (doc_prefix_similarity$_b$ − doc_prefix_similarity$_a$)/($b - a$)
 $\quad\quad$ computed_WMD($w, \mathbf{d}[a : b)$) ← 1 − $\frac{1}{|w|} \sum_{i \in [0..|w|)} \max_{j \in [a..b)}$ token_similarity$_{i,j}$
 $\quad\quad$ symmetric_WMD ← max(computed_WMD($\mathbf{d}[a : b), w$), computed_WMD($w, \mathbf{d}[a : b)$))
 $\quad\quad$ **if** symmetric_WMD $< \delta$ **then**
 $\quad\quad\quad$ **return** True
 $\quad\quad$ **end if**
 $\quad$ **end for**
**end for**
**return** False

---

## A.3 Deduplication of the Reference Corpus

When a text $\mathbf{x}$ is part of the reference corpus $C$, its CREATIVITY INDEX would trivially become zero. This issue often arises when analyzing the works of famous authors, as their writings are frequently copied, quoted, or cited online. To address this, when analyzing human texts written before the cutoff date of the reference corpus, we exclude any document $\mathbf{d} \in C$ that contains copies, quotations, or citations of the text $\mathbf{x}$ from the reference corpus $C$, and compute CREATIVITY INDEX of $\mathbf{x}$ using this filtered reference corpus.

Specifically, we measure the degree of overlap between $\mathbf{x}$ and $\mathbf{d}$ by calculating the length of the longest common subsequence (LCS) between them, normalized by the length of $\mathbf{x}$. Formally, $S(\mathbf{x}, \mathbf{d}) = \frac{||\text{LCS}(\mathbf{x}, \mathbf{d})||}{||\mathbf{x}||}$. If $\mathbf{x}$ and $\mathbf{d}$ have a high degree of overlap (i.e., $S(\mathbf{x}, \mathbf{d}) \geq \alpha$), it's very likely that $\mathbf{d}$ contains an exact copy of $\mathbf{x}$. If $\mathbf{x}$ and $\mathbf{d}$ show a moderate amount of overlap (i.e., $\beta \leq S(\mathbf{x}, \mathbf{d}) < \alpha$), we prompt a LLM to determine whether $\mathbf{d}$ contains copies or quotations of $\mathbf{x}$ using in-context examples provided below. Additionally, if $\mathbf{d}$ includes the author name or title of $\mathbf{x}$, it is highly likely that $\mathbf{d}$ contains a citation of $\mathbf{x}$. In practice, we set the values of $\alpha$ and $\beta$ to 0.9 and 0.3, respectively, and use LLaMA 2 Chat as the LLM to check for copies and quotations.

To determine the optimal value for $\alpha$ and $\beta$, we analyzed the relationship between the degree of overlap $S(\mathbf{x}, \mathbf{d})$ and the likelihood that the reference document $\mathbf{d}$ contains an exact copy or quotations of the input text $\mathbf{x}$.

| Degree of Overlap $S(\mathbf{x}, \mathbf{d})$ | 0.1 | 0.2 | 0.3 | 0.4 | 0.5 | 0.6 | 0.7 | 0.8 | 0.9 | 1.0 |
|---|---|---|---|---|---|---|---|---|---|---|
| **Percentage of Exact Copy** | 0.0 | 0.0 | 0.0 | 0.03 | 0.12 | 0.28 | 0.49 | 0.77 | 0.94 | 1.0 |
| **Percentage of Quotation** | 0.0 | 0.02 | 0.04 | 0.11 | 0.15 | 0.13 | 0.05 | 0.02 | 0.0 | 0.0 |

We found that when $S(\mathbf{x}, \mathbf{d}) < 0.3$, there are no exact copies, and almost no quotations of the input text $\mathbf{x}$ in the reference document $\mathbf{d}$. Conversely, when $S(\mathbf{x}, \mathbf{d}) > 0.9$, the reference document $\mathbf{d}$ almost always contains an exact copy of $\mathbf{x}$. For cases where $0.3 < S(\mathbf{x}, \mathbf{d}) < 0.9$, all three scenarios (exact copy, quotation, or none) are plausible. Therefore, we set the values of $\alpha$ and $\beta$ to 0.9 and 0.3 to ensure accurate deduplication.

```
Please check if paragraph A contains any copies or quotations from
paragraph B.

Here are some examples:
Paragraph A: In the end though, I did the required reading, complained
bitterly about being bored, wrote the requisite essay, and promptly
forgot all about it.  "He was an old man who fished alone in a skiff
in the Gulf Stream and he had gone eighty-four days now without taking
a fish.  In the first forty days ...
Paragraph B: He was an old man who fished alone in a skiff in the Gulf
Stream and he had gone eighty-four days now without taking a fish.  In
the first forty days a boy had been with him.  But after forty days
without a fish the boy's parents had told him that the old man was now
definitely and finally salao ...
Answer:  Yes

Paragraph A: He was an old man who fished alone in a lobster boat off the
Maine coast and he had gone 117 days without taking a crustacean.  His
luck was not bad, rather his judgment was good (don't fish the Atlantic
in winter).  Then he met us and for all I know his luck changed.  El
Campion is due for a change of luck ...
Paragraph B: He was an old man who fished alone in a skiff in the Gulf
Stream and he had gone eighty-four days now without taking a fish.  In
the first forty days a boy had been with him.  But after forty days
without a fish the boy's parents had told him that the old man was now
definitely and finally salao ...
Answer:  No

Paragraph A: Santiago, the "old man who fished alone," in Hemingway's
"The Old Man and the Sea" appears as one who has an undefeatable
character, a loving, cheerful character, and very humble.  The writer
describes him in this way:  "Everything about him was old except his
eyes, and they were the same color as the sea ...
Paragraph B: He was an old man who fished alone in a skiff in the Gulf
Stream and he had gone eighty-four days now without taking a fish.  In
the first forty days a boy had been with him.  But after forty days
without a fish the boy's parents had told him that the old man was now
definitely and finally salao ...
Answer:  Yes

Paragraph A: He was an old man who could see the form of his god, and
a monk, moreover.  Izzie had limited ability to communicate directly with
her own deity.  Much of her life she had proceeded by vague impressions
and only glimpsed the great god's image briefly in the depths of
meditation ...
Paragraph B: He was an old man who fished alone in a skiff in the Gulf
Stream and he had gone eighty-four days now without taking a fish.  In
the first forty days a boy had been with him.  But after forty days
without a fish the boy's parents had told him that the old man was now
definitely and finally salao ...
Answer:  No

Here is the test example:
Paragraph A: [A]
Paragraph B: [B]
Answer:
```

# B EVALUATION

## B.1 MACHINE TEXT GENERATION

We experiment with state-of-the-art LLMs: GPT-3 (Brown et al., 2020) (`text-davinci-003`), ChatGPT (Ouyang et al., 2022) (`gpt-3.5-turbo`), LLaMA 2 Chat (Touvron et al., 2023), Tulu 2 (Ivison et al., 2023) and OLMo Instruct (Groeneveld et al., 2024) along with their base model before RLHF: GPT Base (Brown et al., 2020) (`davinci-002`), LLaMA 2 Base (Touvron et al., 2023) and OLMo Base (Groeneveld et al., 2024). These models are primarily pre-trained on the web data available before the cutoff date of the reference corpus RedPajama (Computer, 2023). We additionally discuss how to handle newer models, such as GPT-4 (OpenAI et al., 2023), which are largely trained on more recent web data and third-party private data, both of which fall outside the reference corpus RedPajama.

To obtain machine texts, we prompt LLMs to generate several paragraphs of novels, poems, or speeches, starting with an initial sentence taken from existing human writings in each category. To construct test data for machine text detection, we further prompt LLMs to generate news articles based on the fake news headlines from the Misinfo Reaction Frames (Gabriel et al., 2022) and to generate theorem proofs for questions from the NaturalProofs (Welleck et al., 2022) benchmark. The prompts used for each task are illustrated below. For all generations, we use nucleus sampling with $p = 0.9$ and set the maximum length of the generated texts to 288 tokens.

```
Please write a few paragraphs for a novel starting with the following
prompt:  [PROMPT SENTENCE]
```

```
Please write a poem starting with the following line:  [PROMPT LINE]
```

```
Please write a speech starting with the following sentence:  [PROMPT
SENTENCE]
```

```
Please write a news article based on the given headline:  [NEWS
HEADLINE]
```

```
Please provide a proof for the following theorem:  [THEOREM QUESTION]
```

To obtain model-generated reference corpus to compare the CREATIVITY INDEX of GPT-4 with humans, we randomly sample 150k sentences from the RedPajama corpus and prompt open-weight LLMs with knowledge cutoff dates similar to GPT-4 to generate document-level continuations. The models we use are the instruction-aligned versions of Gemma-7B (Team et al., 2024) (`gemma-7b-it`), Llama3-8B (`Meta-Llama-3-8B`) (AI@Meta, 2024), and Mixtral-7B (`Mistral-7B-v0.1`) (Jiang et al., 2023). The prompt used to generate continuations is illustrated below. We use nucleus sampling with $p = 0.9$ and set the maximum length of the generated texts to 2048 tokens.

```
Please generate a continuation for the following sentence:  [PROMPT
SENTENCE]
```

We additionally experiment with different prompt formats, intentionally encouraging creativity in models' generations by incorporating instructions such as 'push for creative ideas, unique emotions, and original twists,' 'be bold and creative,' or 'you are a creative writer.' Please see blow for a full list of the prompts we tried.

```
Write a few paragraphs for a novel from the following prompt, pushing
for creative ideas, unique emotions, and original twists.
Prompt:  [PROMPT SENTENCE]
```

```
Use the following prompt to write a few paragraphs for a novel with
creative, unqiue perspectives or twists. Let your originality shine.
Prompt: [PROMPT SENTENCE]
```

```
Create a few paragraphs from the following prompt for a novel, focusing
on novel ideas, emotions, or perspectives. Be as creative as possible.
Prompt: [PROMPT SENTENCE]
```

```
Write a few paragraphs for a novel based on the following prompt,
exploring unexpected twists, emotions, or unique perspectives. Be bold
and creative.
Prompt: [PROMPT SENTENCE]
```

```
Based on the following prompt, and write a few paragraphs for a novel
that explore unexpected twists, deep emotions, or unique perspectives.
Let your creativity flow, and don't be afraid to experiment with
unconventional ideas or characters
Prompt: [PROMPT SENTENCE]
```

```
As a creative agent, write a few paragraphs for a novel based on the
following prompt, bringing your novel ideas and original emotions to
life.
Prompt: [PROMPT SENTENCE]
```

```
You are a creative writer, write a few paragraphs for a novel based
on the following prompt. Explore unique perspectives and unexpected
twists, and let your creativity guide you.
Prompt: [PROMPT SENTENCE]
```

```
You are a creative agent, free to shape this story in any direction.
Write a few paragraphs for a novel based on the following prompt, using
your imagination to uncover surprises and depth.
Prompt: [PROMPT SENTENCE]
```

```
As a creative writer, your task is to write a few paragraphs for a
novel based on the following prompt. Dive into original ideas, explore
emotions, and surprise yourself.
Prompt: [PROMPT SENTENCE]
```

```
You are a creative writer who brings stories to life. Write a few
paragraphs for a novel based on the following prompt, letting your
imagination take bold, unexpected turns.
Prompt: [PROMPT SENTENCE]
```

## B.2 DATASET DETAILS

**Reference Corpus:** We use RedPajama (Computer, 2023), the largest web data collection available at the time of this study, as our reference corpus. RedPajama contains 100 billion text documents with 100+ trillion raw tokens from 84 CommonCrawl dumps.

**Novel:** For human-written novels, we use book snippets from the BookMIA (Shi et al., 2024) dataset. The BookMIA dataset contains approximately 10k book snippets, with an average length of around 650 words per snippet. We randomly sample 100 book snippets from the BookMIA dataset and select the first $K$ sentences of each snippet such that their total length exceeds 256 words, to use as human text. Since novels we use were published after the cutoff date of RedPajama, there's no need for deduplication before DJ SEARCH.

**Speech:** For the transcripts of human speeches, we randomly sample 100 speeches from the famous speeches available in the American Rhetoric speech bank. For each speech, we randomly sample continuous $K$ sentences such that their total length exceeds 256 words, to use as human text. Since these speeches were made before the cutoff date of RedPajama, deduplication is needed before DJ SEARCH.

**Poem:** For human-written poems, we randomly sample 100 poems from the popular modern poems collected by PoemHunter.com. Since these poems were published before the cutoff date of RedPajama, deduplication is needed before DJ SEARCH.

**News Article:** We use news articles from the XSum (Narayan et al., 2018) dataset as the human text for the machine text detection task. The Xsum dataset contains around 200k new articles, with an average length of around 380 words per article. We randomly sample 500 articles to use as human text. Since these news articles were released before the cutoff date of RedPajama, deduplication is needed before DJ SEARCH. For machine-generated fake news, we randomly sample 500 fake news headlines from the Misinfo Reaction Frames (Gabriel et al., 2022), and based on these headlines, LLMs are asked to generate corresponding news articles.

**Theorem Proof:** We use the ground-truth human-written proofs from the NaturalProofs (Welleck et al., 2022) dataset as the human text for the machine text detection task. The NaturalProofs dataset contains approximately 24k theorems and their corresponding proofs. We randomly sample 500 theorem-proof pairs and use the ground-truth proofs as human text. Since the NaturalProofs dataset was curated after the cutoff date of RedPajama, there's no need for deduplication before DJ SEARCH. For machine-generated math proofs, we prompt LLMs to write proofs for the 500 theorems we sampled.

### B.3 PARAMETERS OF DJ SEARCH

We set the minimum $n$-gram length $L$ in DJ SEARCH to 5, and set the threshold for Word Mover's Distance to 0.95 for semantic matches. We observe that the $L$-uniqueness is close to zero for most human and machine texts when $L \leq 5$ and close to one when $L \geq 12$. Therefore, in practice, we sum up the $L$-uniqueness for $5 \leq L \leq 12$ when computing CREATIVITY INDEX.

The only experiment with slightly different parameters is to compare the creativity of GPT-4 with humans. We observed that the $L$-uniqueness is close to one when $L \geq 7$ based on the model-generated reference corpus. Therefore, we sum up the $L$-uniqueness for $5 \leq L \leq 7$ when computing CREATIVITY INDEX.

### B.4 RUNTIME OF DJ SEARCH

To verify linear complexity, we measured the runtime of DJ SEARCH with respect to increasing input text lengths, using RedPajama as the reference corpus, restricting the matching criteria to verbatim matches only and averaging the results over 500 examples.

| Input Length | 64 | 128 | 256 | 512 |
|---|---|---|---|---|
| Runtime (seconds) | 9.2 | 16.9 | 30.7 | 62.3 |

We observed that the runtime of DJ SEARCH indeed increases linearly with the length of the input text. We also measured the runtime of DJ SEARCH with respect to increasing sizes of the reference corpus while keeping the input length fixed at 256 words. In addition to RedPajama, which contains approximately 1.4 trillion tokens, we experimented with the Pile (380B tokens) and DOLMA (2.6 trillion tokens).

| Reference Corpus Size | 380B Tokens | 1.4T Tokens | 2.6T Tokens |
|---|---|---|---|
| Runtime (seconds) | 24.1 | 30.7 | 37.1 |

We found that the runtime increase for DJ SEARCH is minimal as the size of the reference corpus grows. This is likely due to the high optimization of the Infini-gram, which enables it to retrieve matched $n$-grams within milliseconds.

## B.5 Variation of Word Mover's Distance

We experimented with different variations of transport-inspired distances, including relaxed WMD, supervised WMD (Huang et al., 2016), and Sinkhorn distance (Cuturi, 2013).

We computed the CREATIVITY INDEX for ChatGPT and human writing in the novel domain based on semantic matches using relaxed WMD, supervised WMD, and Sinkhorn distance. Regardless of the distance metric used, human writings consistently exhibited a significantly higher CREATIVITY INDEX than ChatGPT, aligning with our main results. Specifically, based on relaxed WMD, the CREATIVITY INDEX of human authors was 97.6% higher than ChatGPT ($p = 2.5 \times 10^{-13}$; $N = 100$). Based on supervised WMD, the CREATIVITY INDEX of human authors was 79.3% higher than ChatGPT ($p = 1.9 \times 10^{-7}$; $N = 100$). Based on Sinkhorn distance, the CREATIVITY INDEX of human authors was 91.3% higher than ChatGPT ($p = 3.5 \times 10^{-12}$; $N = 100$).

As mentioned in the main text, the motivation for using transport-inspired distances—which combine word embedding distances for each $n$-gram's words rather than directly computing the Euclidean distance between embeddings of two $n$-grams—is rooted in efficiency. The latter requires computing $n$-gram embeddings for every $n$-gram in the input text and every document in the reference corpus during DJ SEARCH, which is computationally infeasible. Ideally, a good transport-inspired distance should approximate the Euclidean distance between the embeddings of two $n$-grams. To evaluate this, we computed the Pearson correlation between our implementations of WMD, relaxed WMD, supervised WMD, and Sinkhorn distance with the Euclidean distance between the embeddings of two $n$-grams to assess how closely they correlate.

|  | Our MWD | Relaxed MWD | Supervised MWD | Sinkhorn Distance |
|---|---|---|---|---|
| **r-value** | 0.842 | 0.823 | 0.716 | 0.808 |
| **p-value** | $5.7 \times 10^{-11}$ | $4.1 \times 10^{-10}$ | $8.3 \times 10^{-4}$ | $9.1 \times 10^{-8}$ |

We found that all transport-inspired distances correlate highly with Euclidean distance, showing strong statistical significance. The slightly lower correlation with supervised MWD may be due to its training objective, which focuses on distinguishing between different categories of documents.

## B.6 Compare with Existing Creativity Metrics

A recent study by Chakrabarty et al. (2024) evaluates the creativity of stories written by LLMs and professional human authors through expert evaluations based on a carefully designed rubric inspired by the Torrance Test of Creative Thinking (TTCT). This work introduces the Torrance Test of Creative Writing (TTCW), which consists of 14 binary questions grouped into four dimensions: Fluency, Flexibility, Originality, and Elaboration. To conduct the evaluation, ten expert creative writers were recruited to evaluate 48 stories—written either by LLMs or published in The New York Times—using the TTCW rubric.

For each story, three expert evaluators are tasked with answering 14 binary questions, each designed to quantify a specific dimension of creativity. For example, within the Fluency dimension, "Understandability & Coherence" is assessed by asking, "Do the different elements of the story work together to form a unified, engaging, and satisfying whole?" In the Flexibility dimension, "Emotional Flexibility" is evaluated with the question, "Does the story achieve a good balance between interiority and exteriority, in a way that feels emotionally flexible?" Similarly, within the Originality dimension, "Originality in Thought" is measured by asking, "Is the story an original piece of writing without any clichés?" For each of the 14 binary questions, the final answer is determined by majority vote among the expert evaluators.

To evaluate the correlation between our CREATIVITY INDEX and Chakrabarty et al.'s TTCW, we compute the CREATIVITY INDEX for each of the 48 stories. For each binary question in the TTCT (e.g., "Understandability & Coherence" or "Emotional Flexibility"), we calculate the point biserial correlation between the CREATIVITY INDEX values and the corresponding binary expert ratings of that question across all 48 stories.

We found that the CREATIVITY INDEX positively correlates with all dimensions of TTCW, showing particularly strong correlations with high statistical significance in dimensions such as "Emotional Flexibility," "Narrative Pacing," "Structural Flexibility," and "Originality in Thought."

|  | Emotional Flexibility | Narrative Pacing | Structural Flexibility | Originality in Thought | Scene vs Summary | Rhetorical Complexity | Language Proficiency |
|---|---|---|---|---|---|---|---|
| **r-value** | 0.6826 | 0.6524 | 0.5968 | 0.5968 | 0.5812 | 0.5810 | 0.5702 |
| **p-value** | 0.0028 | 0.0051 | 0.0135 | 0.0135 | 0.0135 | 0.0135 | 0.0204 |

|  | World Building & Setting | Narrative Ending | Originality in Form & Structure | Understandability & Coherence | Perspective & Voice Flexibility | Originality in Theme & Content | Character Development |
|---|---|---|---|---|---|---|---|
| **r-value** | 0.5170 | 0.4871 | 0.4626 | 0.4495 | 0.4467 | 0.3582 | 0.2299 |
| **p-value** | 0.0426 | 0.0617 | 0.0816 | 0.0941 | 0.0969 | 0.1231 | 0.1453 |

## C  RELATED WORK

**Measuring Creativity in Ideas:**  Measuring creative thinking and problem solving takes root in early work in psychology (Torrance, 1966), where researchers defined four pillars for creative thinking: fluency, flexibility, originality and elaboration. Crossley et al. (2016) later on developed this notion and built on it to expand this to measuring creative writing in students, where they also adopted $n$-gram novelty for a measure of originality. However, these prior work focus on creativity in humans, and they also do not introduce any automated metrics or measurements.

**Measuring Creativity in Machine-generated Text Using Expert Annotators:**  Closely related to CREATIVITY INDEX is a recent line of work in the generative AI literature comparing the creativity of human writers to that of large language models in different domains such as story telling and journalism (Chakrabarty et al., 2023; 2024; Anonymous, 2024). Similar to us, the approach in this direction often involves prompting an LLM to write an original story or news article, based on some existing premise or press release, and then comparing the machine-generated text to the human-written counterparts. These works, however, take a rather subjective approach, where they define and measure creativity based on human expert annotations and whether people perceive the text to be more creative, rather than an objective quantification of novelty that we provide.

**Measuring Novelty of $N$-grams:**  Finally, closely related to our work in terms of techniques is Nguyen (2024) and Merrill et al. (2024). The former attempts at finding $n$-gram rules that would cover and predict generations from transformer models, showing that more than 70% of the times transformers follow some pre-set patterns and rules. The latter is more similar to our work as they also measure the novelty of generated $n$-grams and compare it to human-written text, however they differ from us in tow major ways: (1) they only find verbatim matches, whereas we also match to approximate, semantically similar blocks of text and (2) they compute the percentage of $n$-grams of a certain length in a text that can be found in the reference corpus, whereas we measure how much of the text can be reconstructed by mixing and matching a vast amount of existing text snippets of varying lengths from the web.

**Machine Text Detection:**  Detecting machine-generated text has been explored for several years using a variety of methods (Jawahar et al., 2020; Uchendu et al., 2021). Gehrmann et al. (2019) and Dugan et al. (2023) demonstrate that even humans tend to struggle to differentiate between text written by humans and machines, highlighting the need for automated detection solutions. Some approaches involve training a classifier in a supervised manner to identify machine-generated text (Bakhtin et al., 2019; Uchendu et al., 2020), while others use a zero-shot detection method (Solaiman et al., 2019; Ippolito et al., 2020). Additionally, there is research on bot detection through question answering (Wang et al., 2023; Chew & Baird, 2003). Recently, Mitchell et al. (2023b) introduced DetectGPT, a zero-shot method based on the hypothesis that texts produced by a large language model (LLM) are located at local maxima, and thus exhibit negative curvature, in the model's probability distribution. Follow-up work build on DetectGPT by making it faster (Bao et al., 2024) and proposing to use cross-detection when the target model is unknown (Mireshghallah et al., 2024).

Various strategies have been developed to detect machine-generated text in real-world settings. One notable approach is watermarking, which embeds algorithmically detectable patterns into the generated text while maintaining the quality and diversity of the language model's outputs. Initial watermarking techniques for natural language were proposed by Atallah et al. (2001) and have been adapted for neural language model outputs (Fang et al., 2017; Ziegler et al., 2019). Recent advance-

ments include Abdelnabi & Fritz (2021) work on an adversarial watermarking transformer (AWT) for transformer-based language models. Unlike methods dependent on specific model architectures, Kirchenbauer et al. (2023) introduce a watermarking technique applicable to texts generated by any common autoregressive language model.

**Application of LLMs in Creative Writing:** Recent advancements have highlighted the potential of LLMs in supporting various creative writing endeavors, ranging from short stories (Yang et al., 2022) to screenplays (Mirowski et al., 2023b). Enhancing LLMs to produce text that aligns more closely with human preferences has made them adept at following user instructions, thereby turning them into valuable tools for individuals without technical expertise. This progress has boosted the commercial viability of LLMs as writing aids, which can continue a narrative, describe scenes, or offer feedback. Chung et al. (2021) conducted a review of literature on creativity support tools across various arts, leading to the development of a taxonomy that includes roles, interactions, and technologies. In contrast, Frich et al. (2019) and Palani et al. (2022) focused on how creative practitioners select new tools, highlighting their emphasis on functionality, workflow integration, and performance, and noting that personal recommendations often guide their choices. Additionally, Gero et al. (2022) created a space based on the cognitive process model of writing, influencing interface design decisions. Gero et al. (2023) further explored the social dynamics of AI in creative tasks, revealing a disconnect between writers' objectives and the support provided by computer tools.

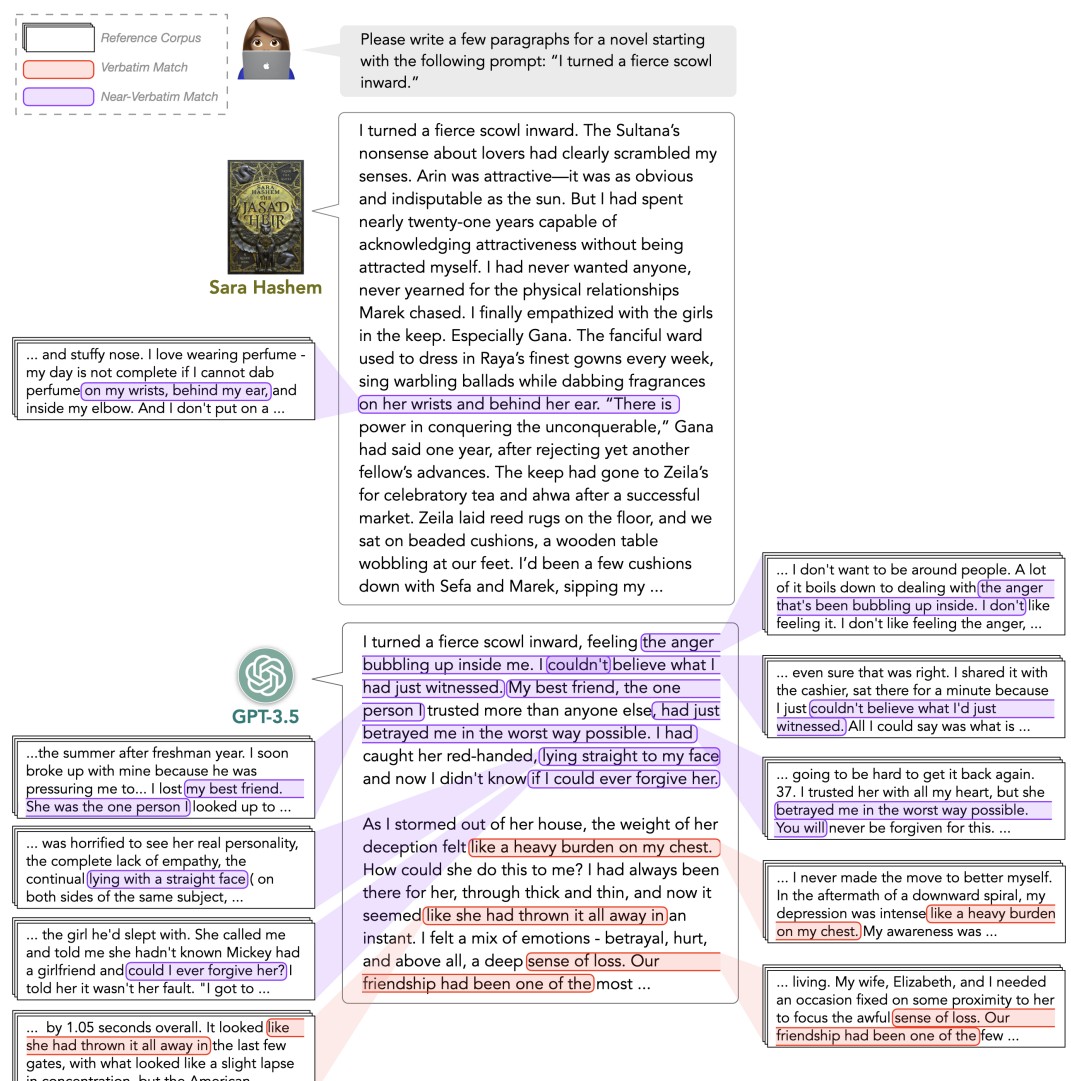

Figure 5: Example outputs from DJ SEARCH based on both verbatim and semantic matches. We prompt LLMs to generate a few paragraphs of a novel, beginning with a first sentence taken from a human-written novel snippet.

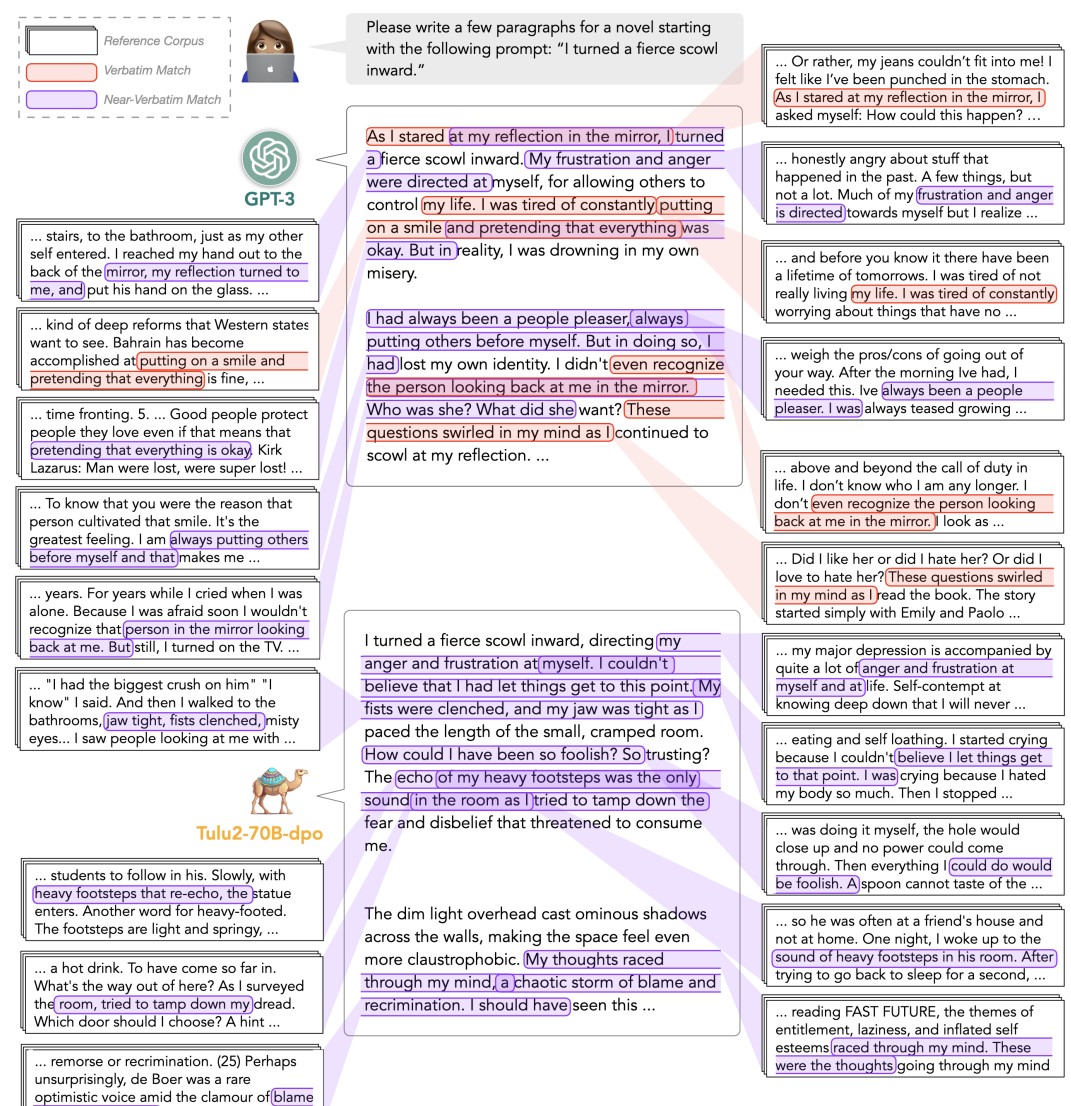

Figure 6: Example outputs from DJ SEARCH based on both verbatim and semantic matches. We prompt LLMs to generate a few paragraphs of a novel, beginning with a first sentence taken from a human-written novel snippet.

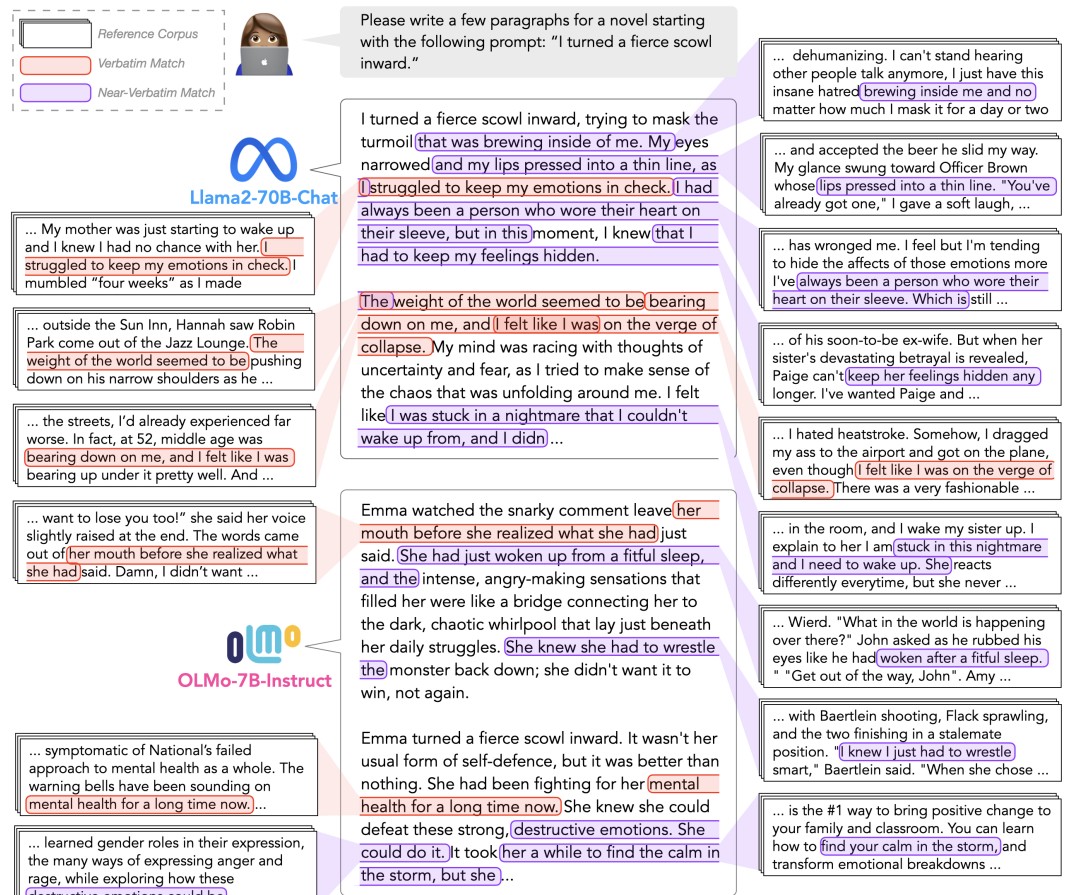

Figure 7: Example outputs from DJ SEARCH based on both verbatim and semantic matches. We prompt LLMs to generate a few paragraphs of a novel, beginning with a first sentence taken from a human-written novel snippet.

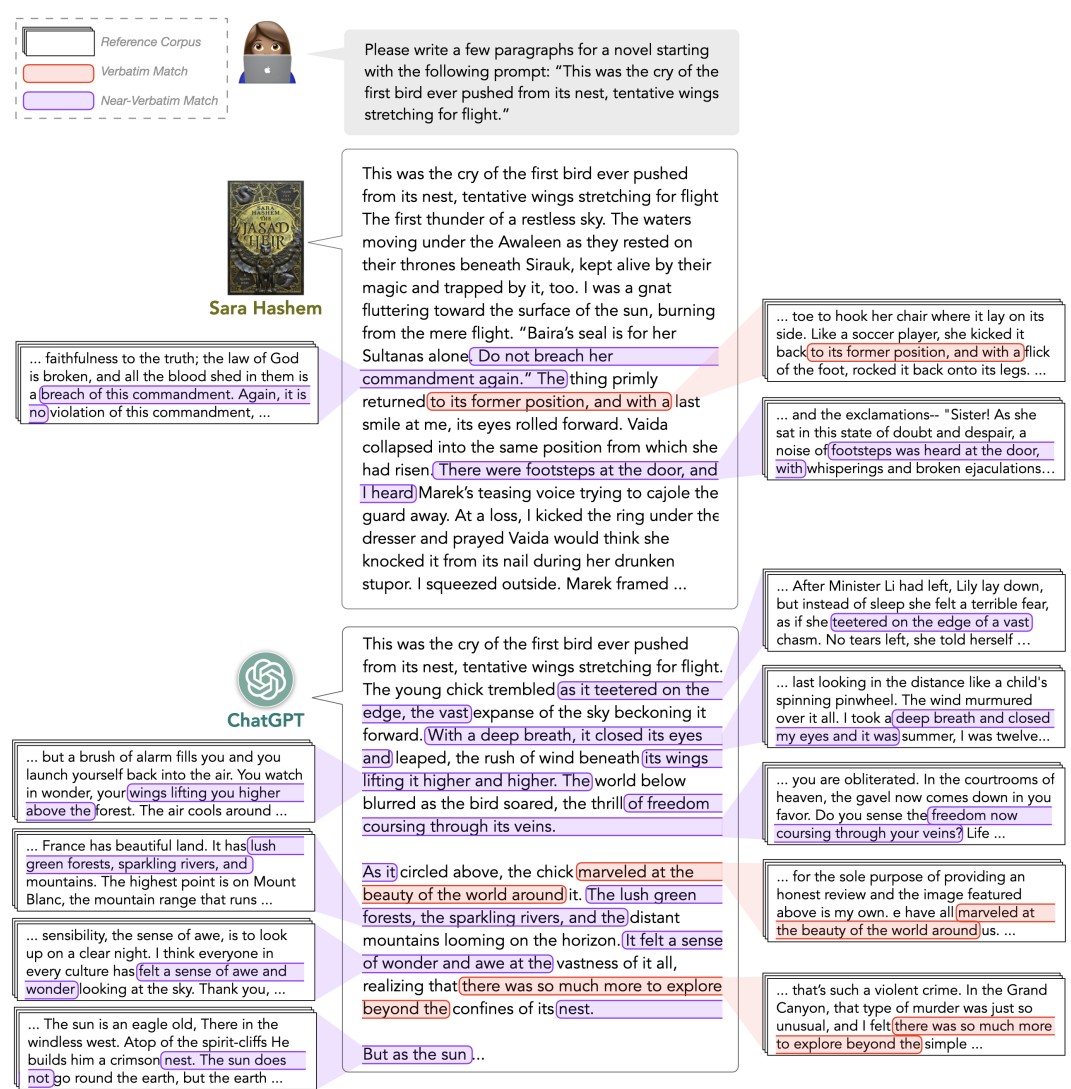

Figure 8: Example outputs from DJ SEARCH based on both verbatim and semantic matches. We prompt LLMs to generate a few paragraphs of a novel, beginning with a first sentence taken from a human-written novel snippet.

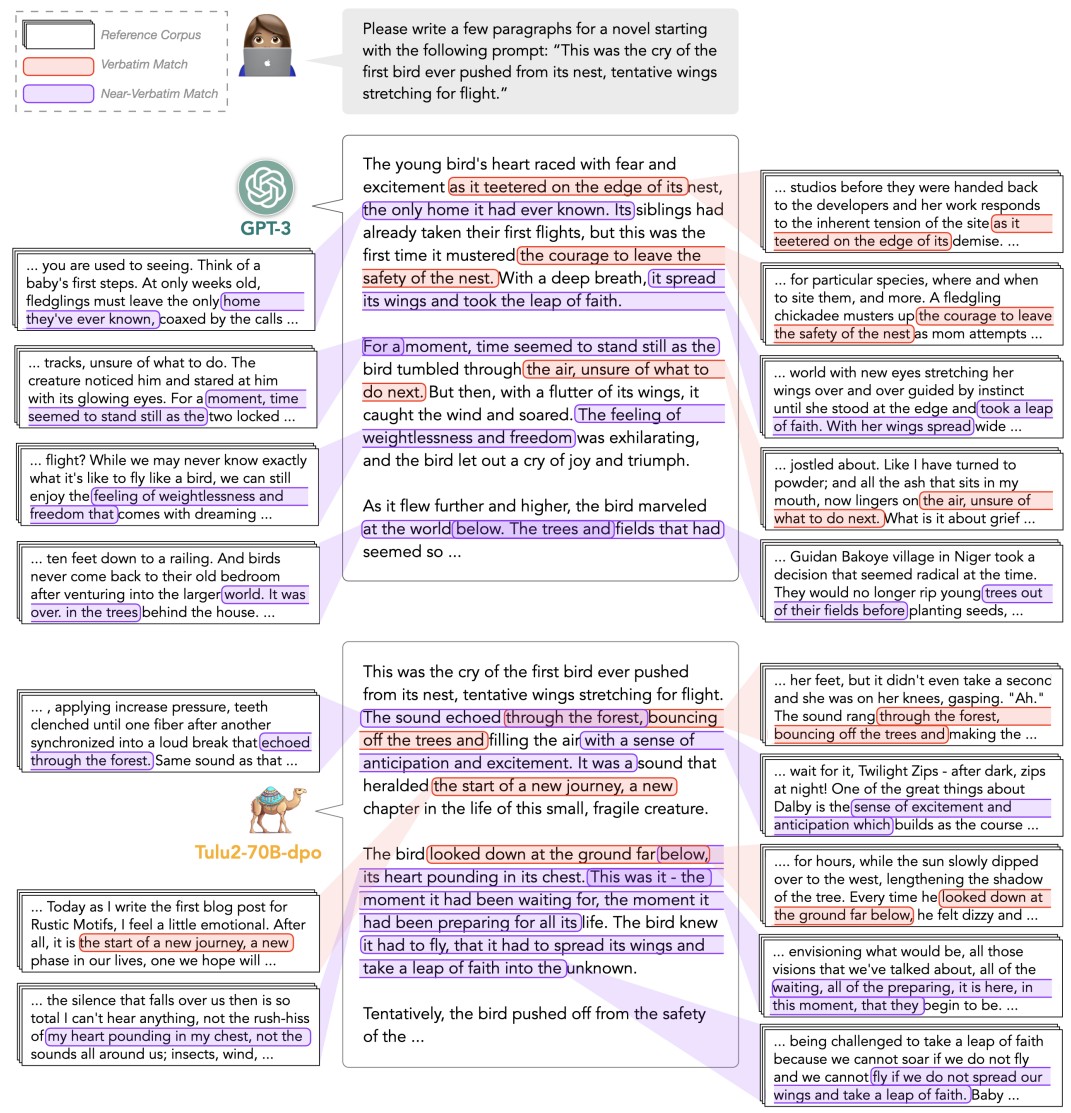

Figure 9: Example outputs from DJ SEARCH based on both verbatim and semantic matches. We prompt LLMs to generate a few paragraphs of a novel, beginning with a first sentence taken from a human-written novel snippet.

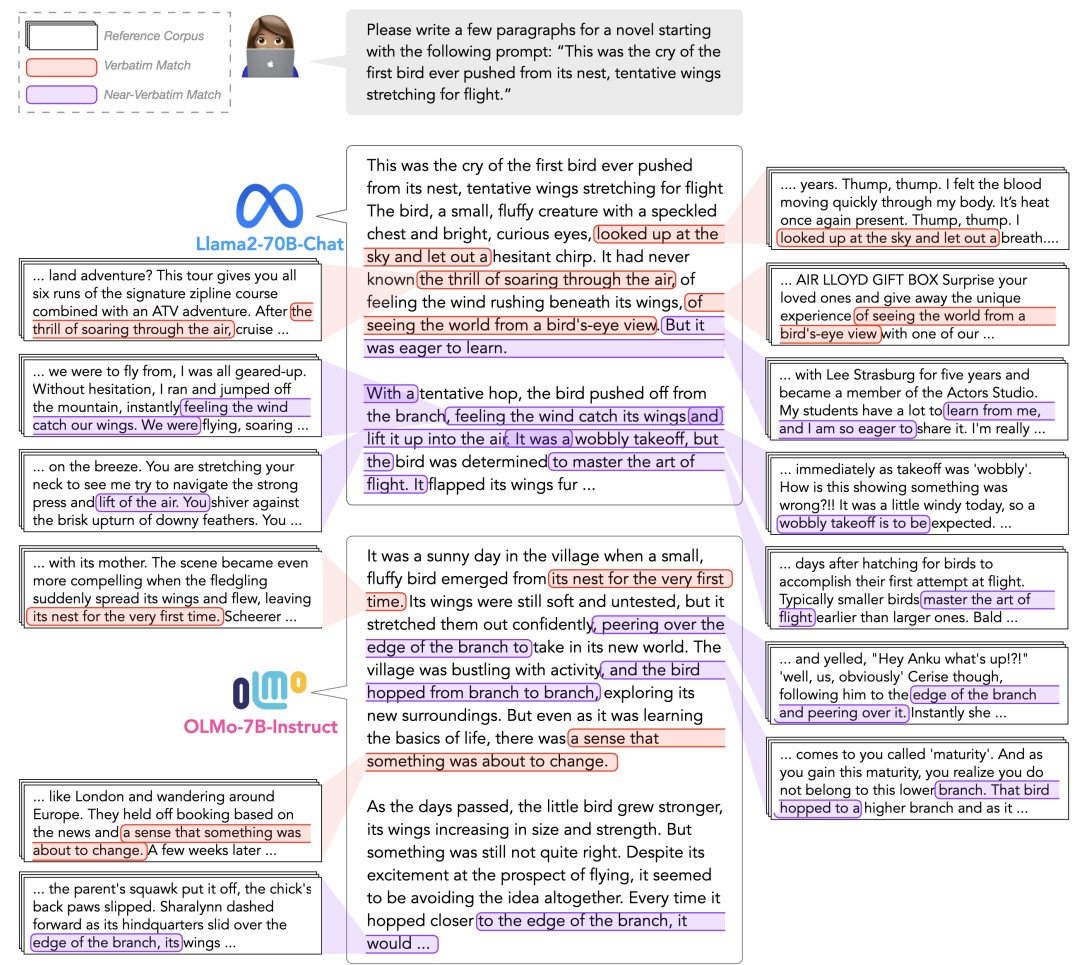

Figure 10: Example outputs from DJ SEARCH based on both verbatim and semantic matches. We prompt LLMs to generate a few paragraphs of a novel, beginning with a first sentence taken from a human-written novel snippet.

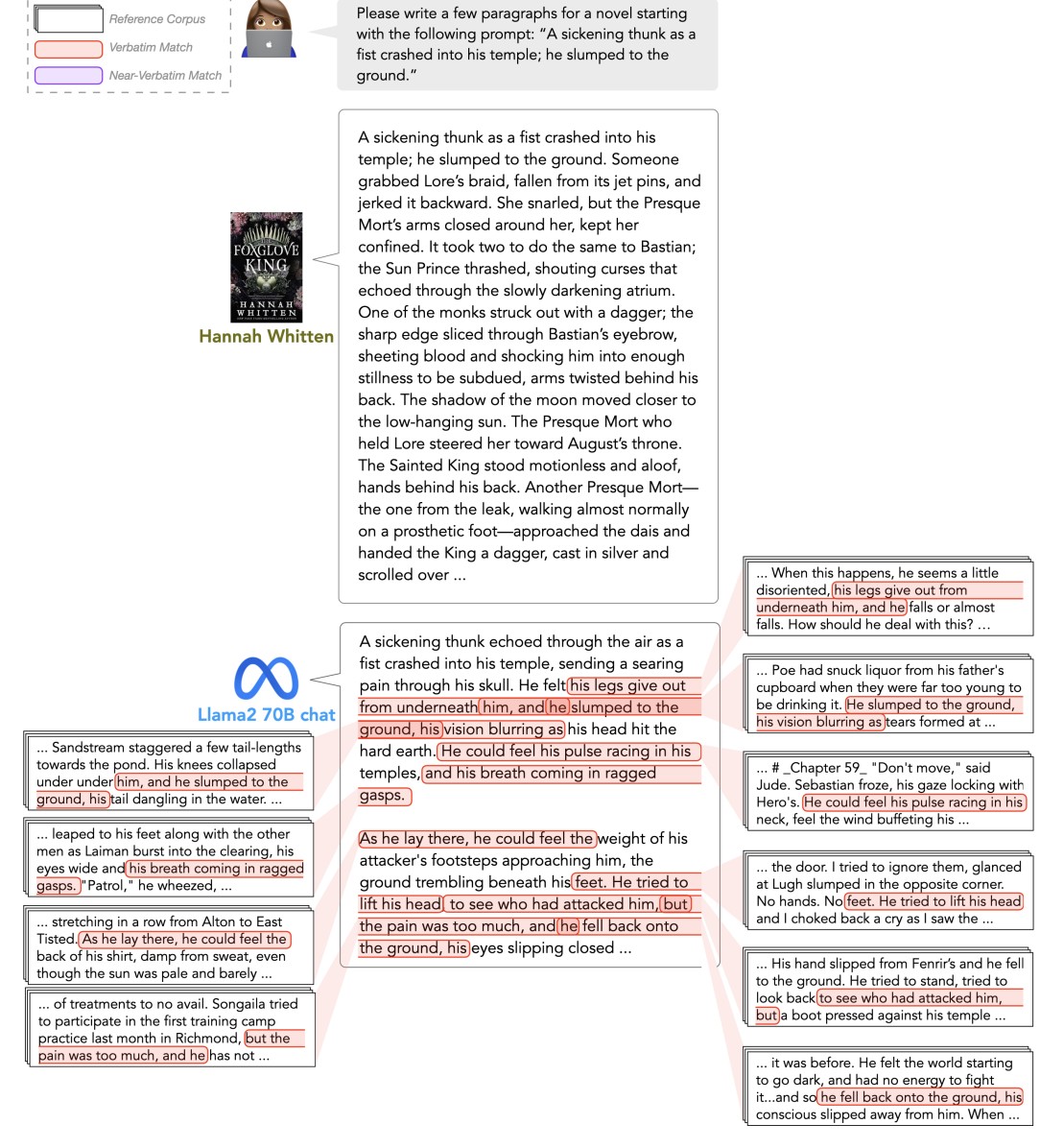

Figure 11: Example outputs from DJ SEARCH based on verbatim matches. We prompt LLMs to generate a few paragraphs of a novel, beginning with a first sentence taken from a human-written novel snippet.

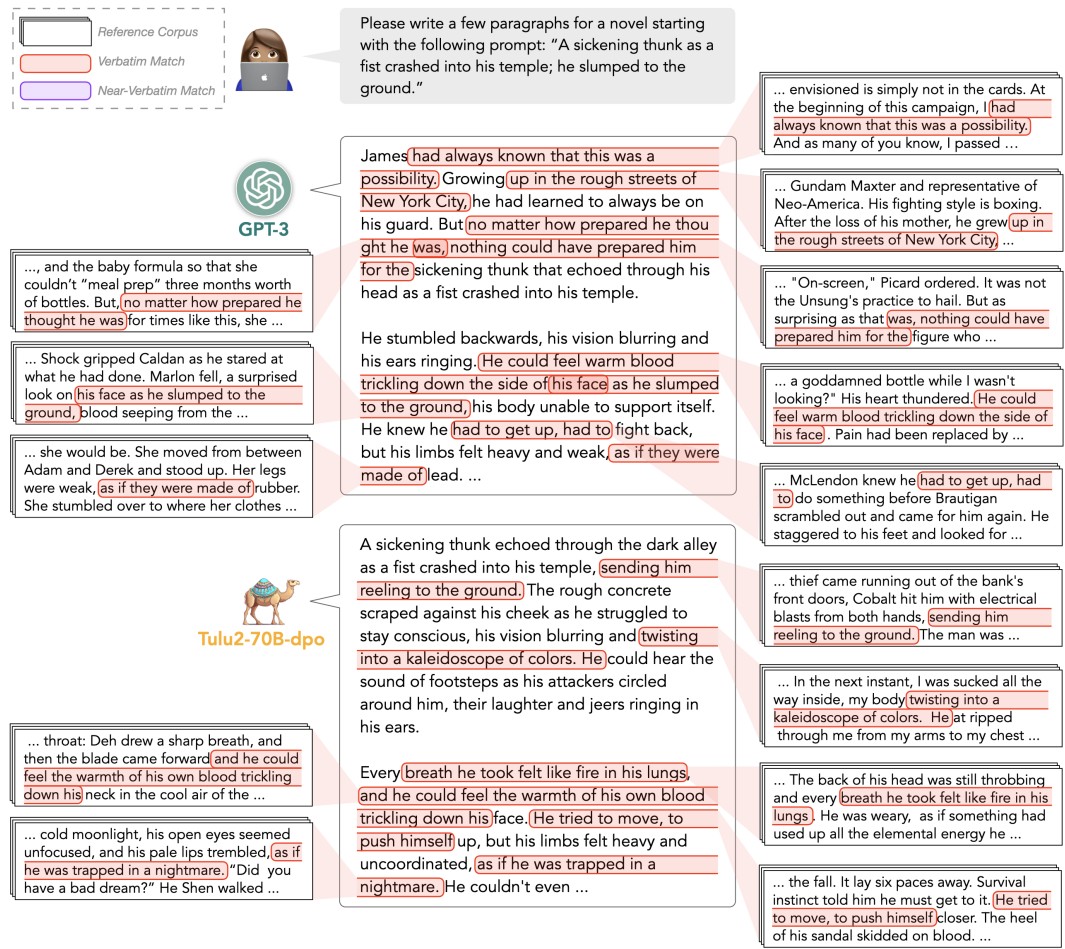

Figure 12: Example outputs from DJ SEARCH based on verbatim matches. We prompt LLMs to generate a few paragraphs of a novel, beginning with a first sentence taken from a human-written novel snippet.

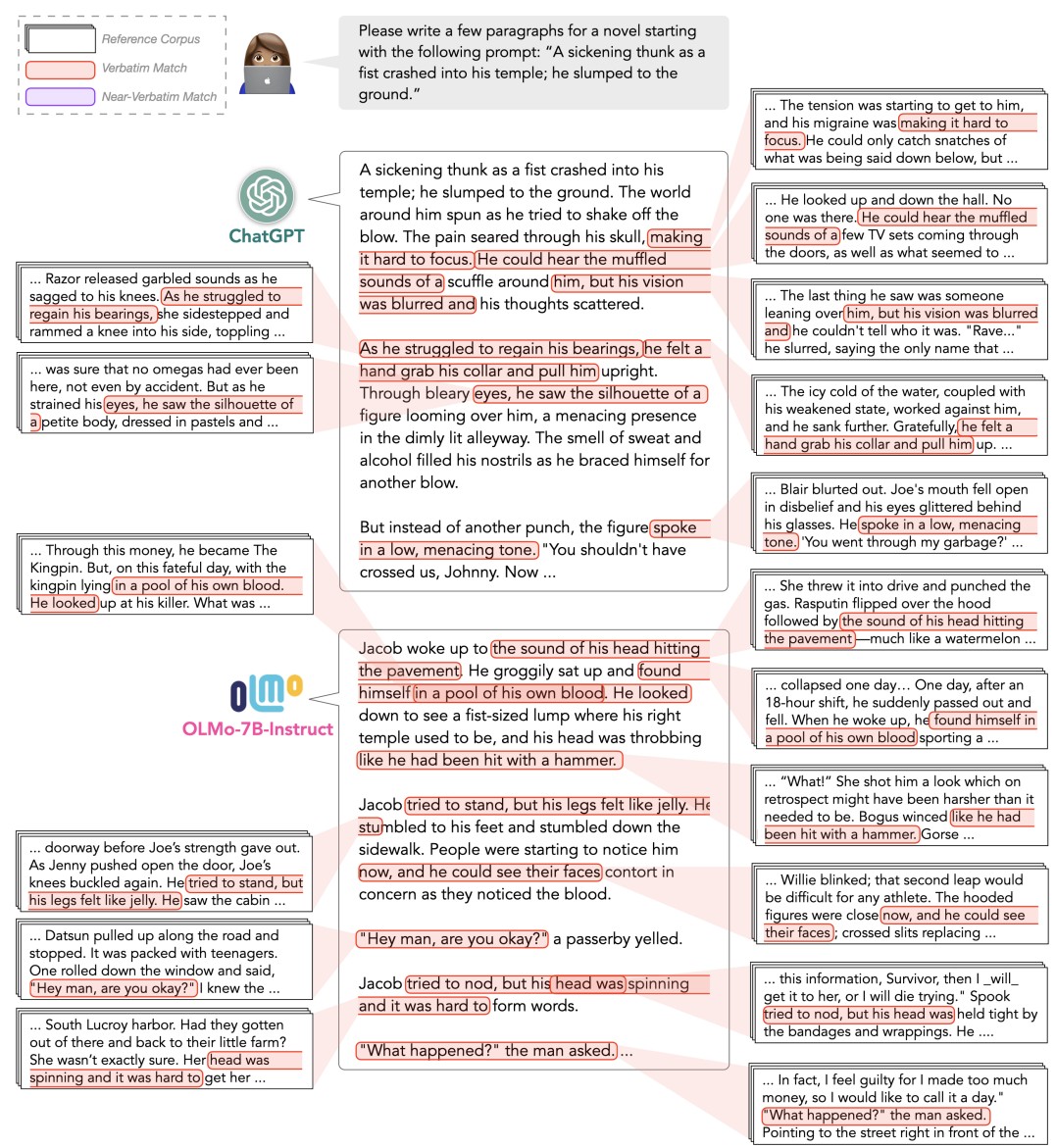

Figure 13: Example outputs from DJ SEARCH based on verbatim matches. We prompt LLMs to generate a few paragraphs of a novel, beginning with a first sentence taken from a human-written novel snippet.

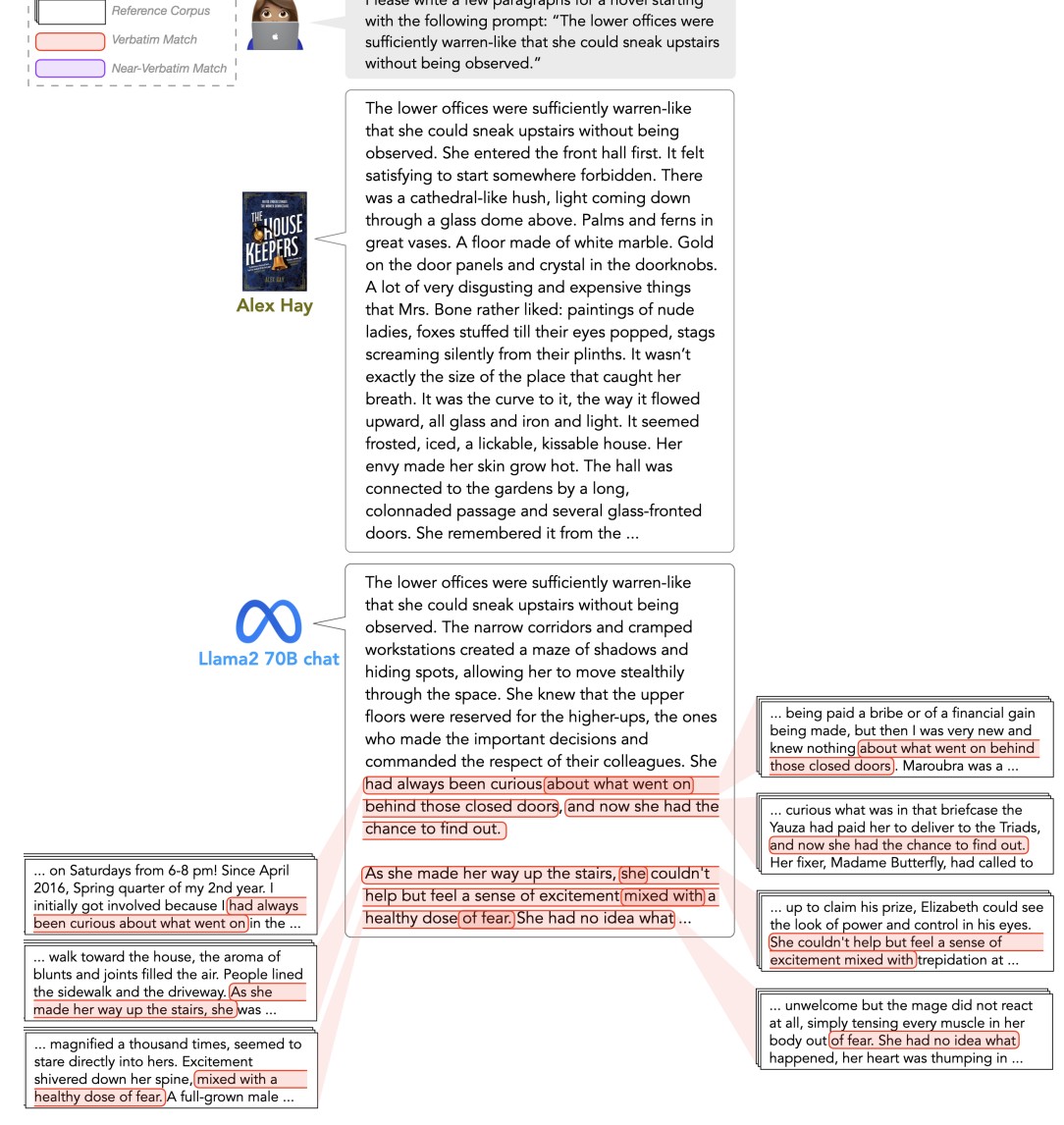

Figure 14: Example outputs from DJ SEARCH based on verbatim matches. We prompt LLMs to generate a few paragraphs of a novel, beginning with a first sentence taken from a human-written novel snippet.

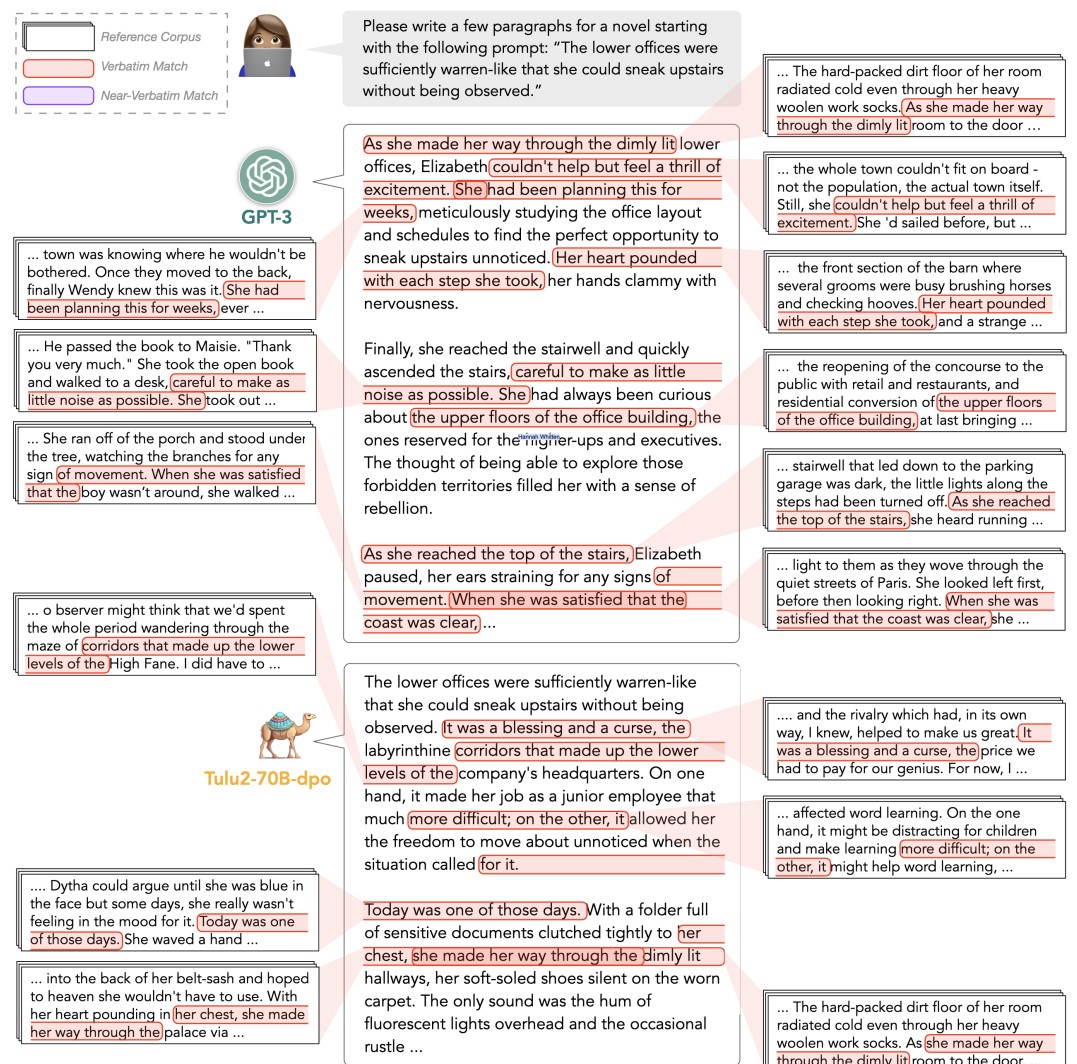

Figure 15: Example outputs from DJ SEARCH based on verbatim matches. We prompt LLMs to generate a few paragraphs of a novel, beginning with a first sentence taken from a human-written novel snippet.

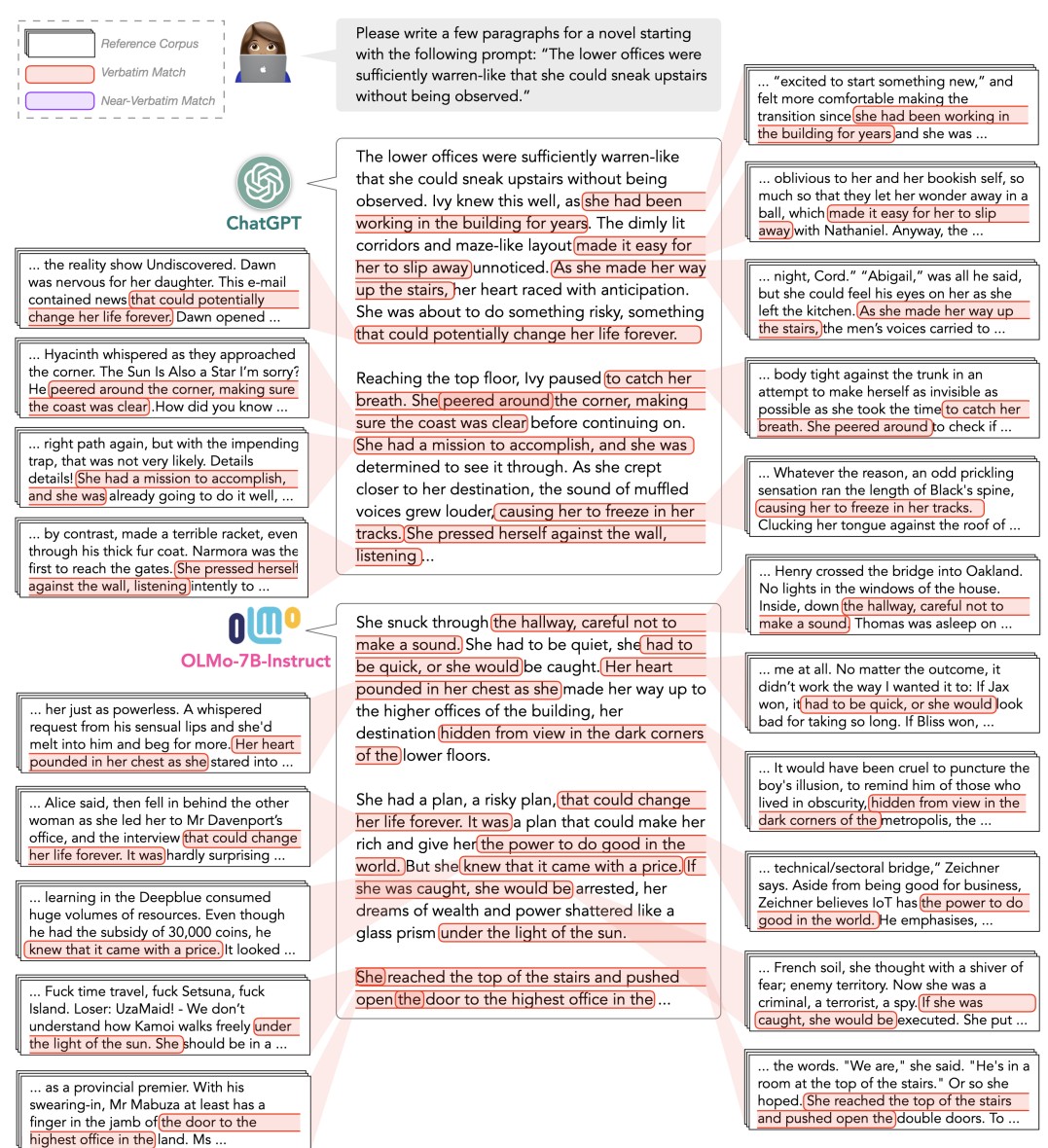

Figure 16: Example outputs from DJ SEARCH based on verbatim matches. We prompt LLMs to generate a few paragraphs of a novel, beginning with a first sentence taken from a human-written novel snippet.

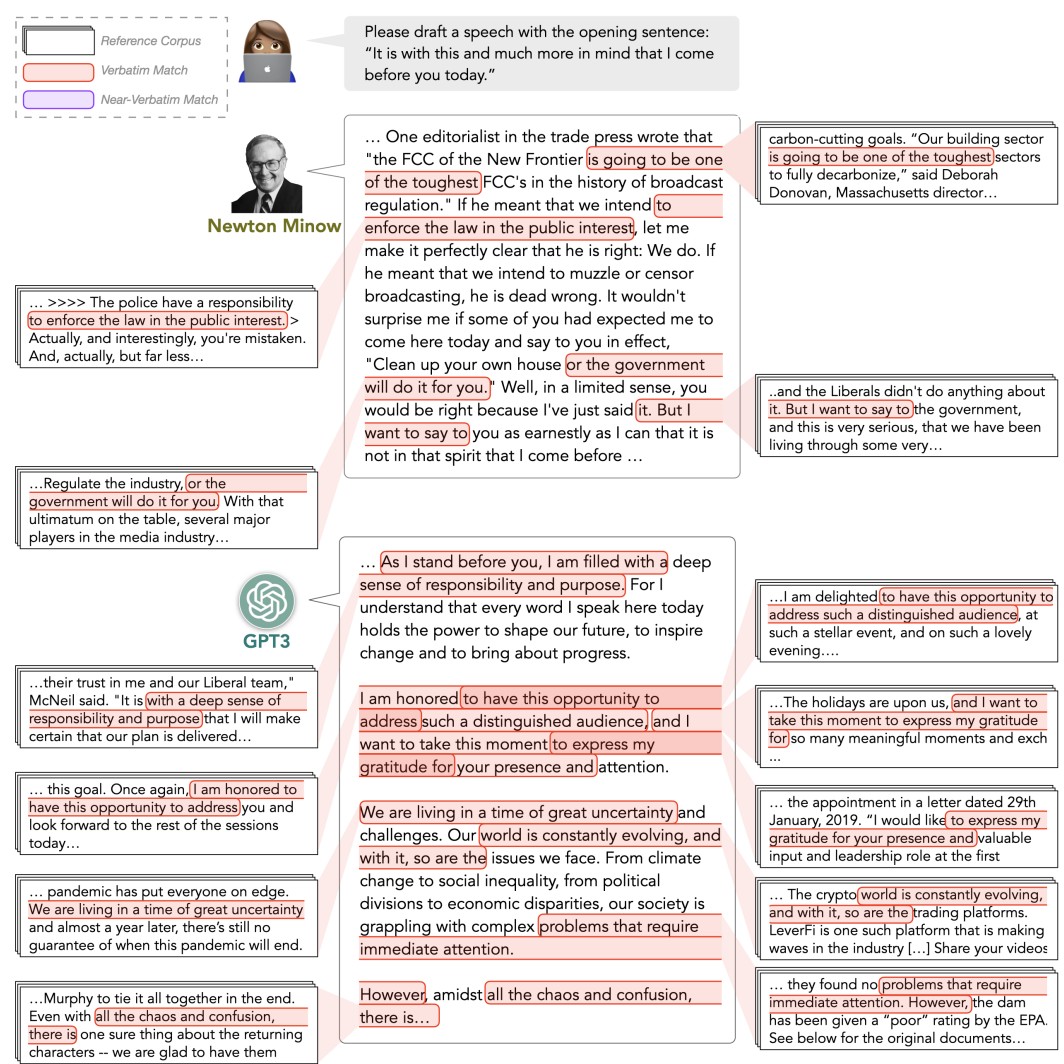

Figure 17: Example outputs from DJ SEARCH based on verbatim matches. We prompt LLMs to generate a speech starting with the opening sentence of a human speech transcript.

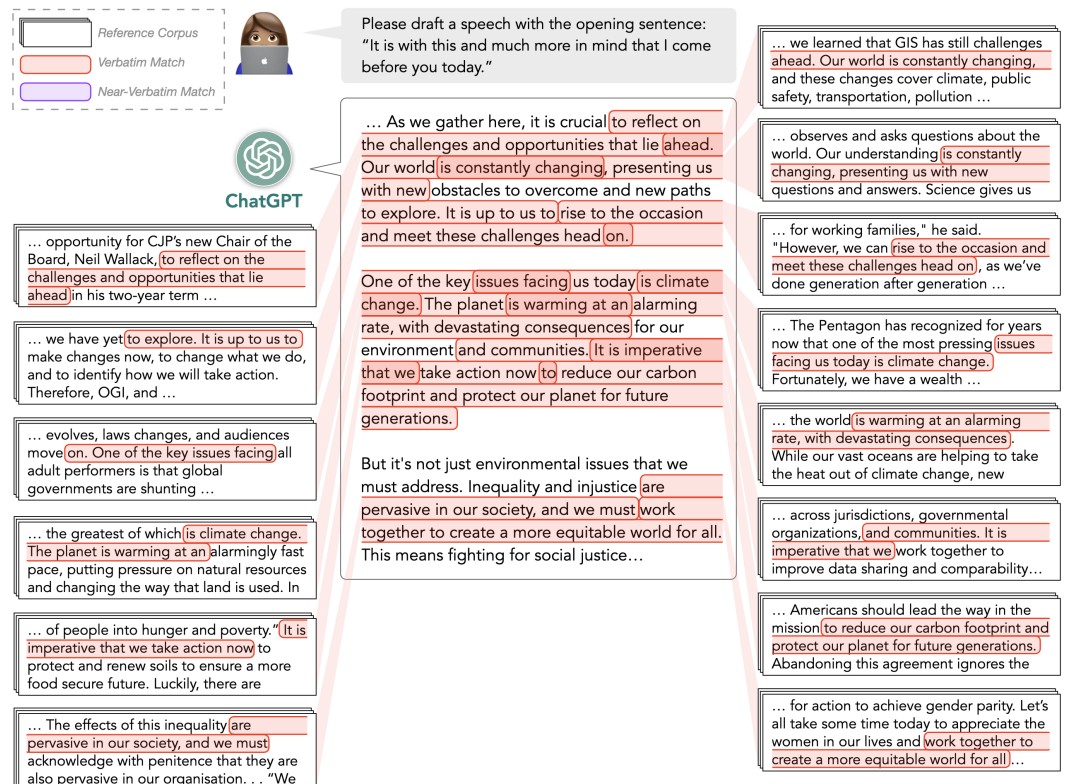

Figure 18: Example outputs from DJ SEARCH based on verbatim matches. We prompt LLMs to generate a speech starting with the opening sentence of a human speech transcript.

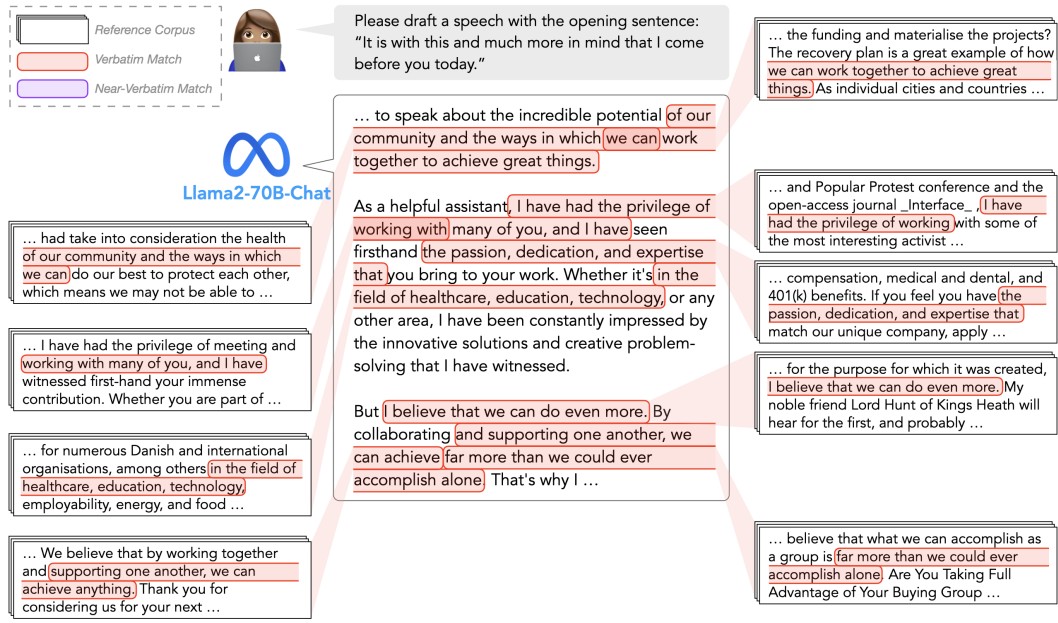

Figure 19: Example outputs from DJ SEARCH based on verbatim matches. We prompt LLMs to generate a speech starting with the opening sentence of a human speech transcript.

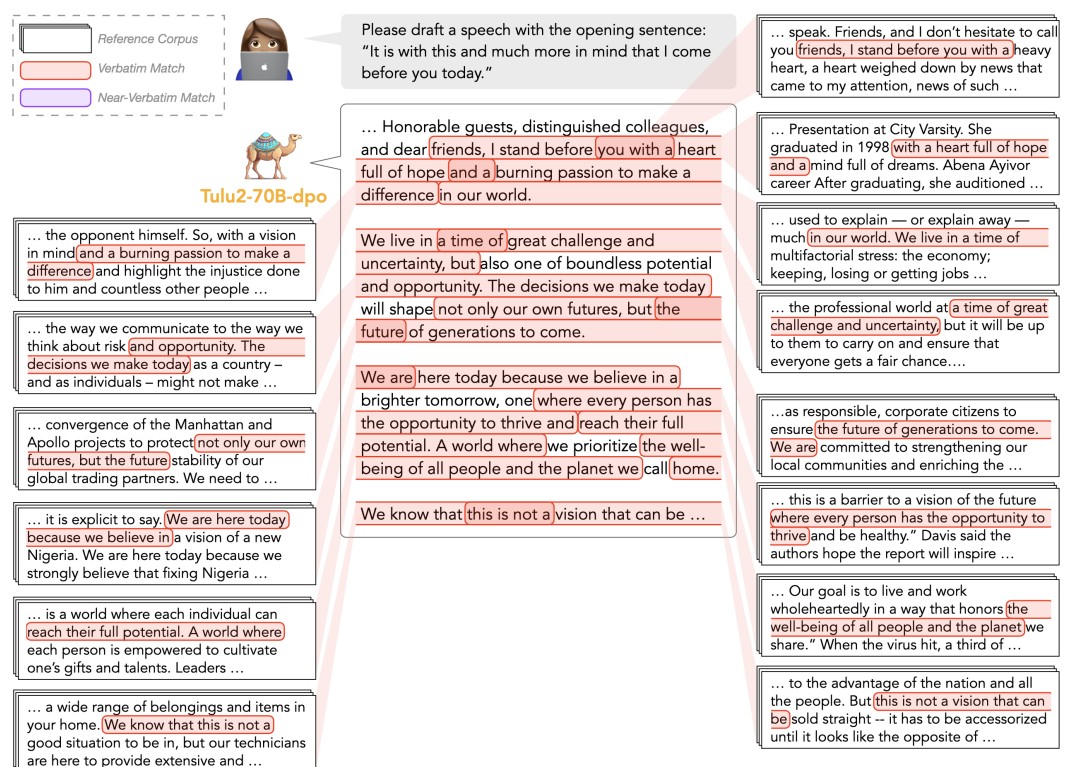

Figure 20: Example outputs from DJ SEARCH based on verbatim matches. We prompt LLMs to generate a speech starting with the opening sentence of a human speech transcript.

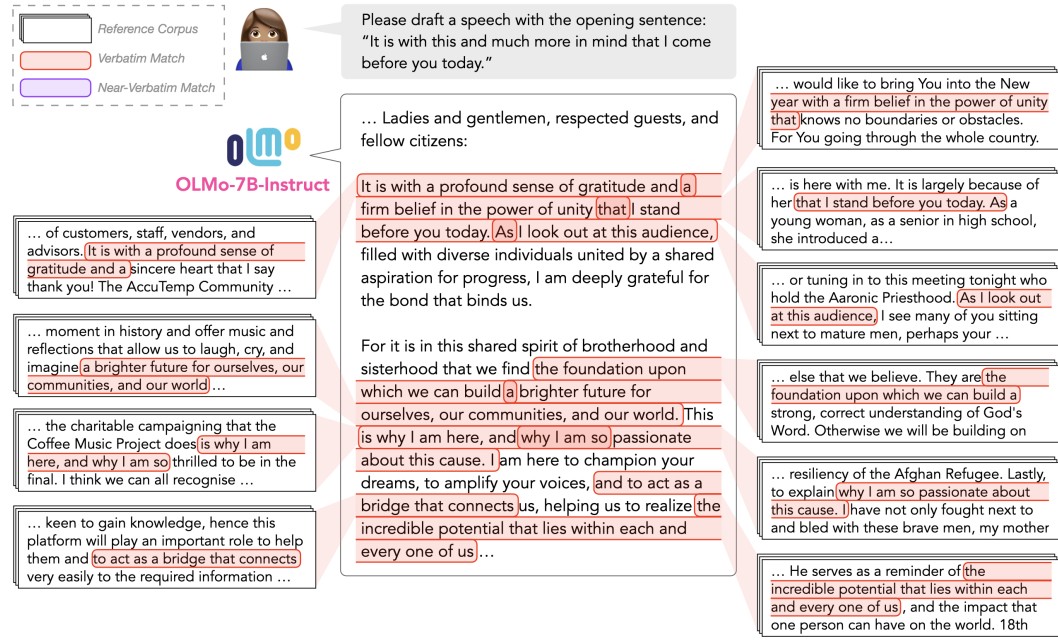

Figure 21: Example outputs from DJ SEARCH based on verbatim matches. We prompt LLMs to generate a speech starting with the opening sentence of a human speech transcript.

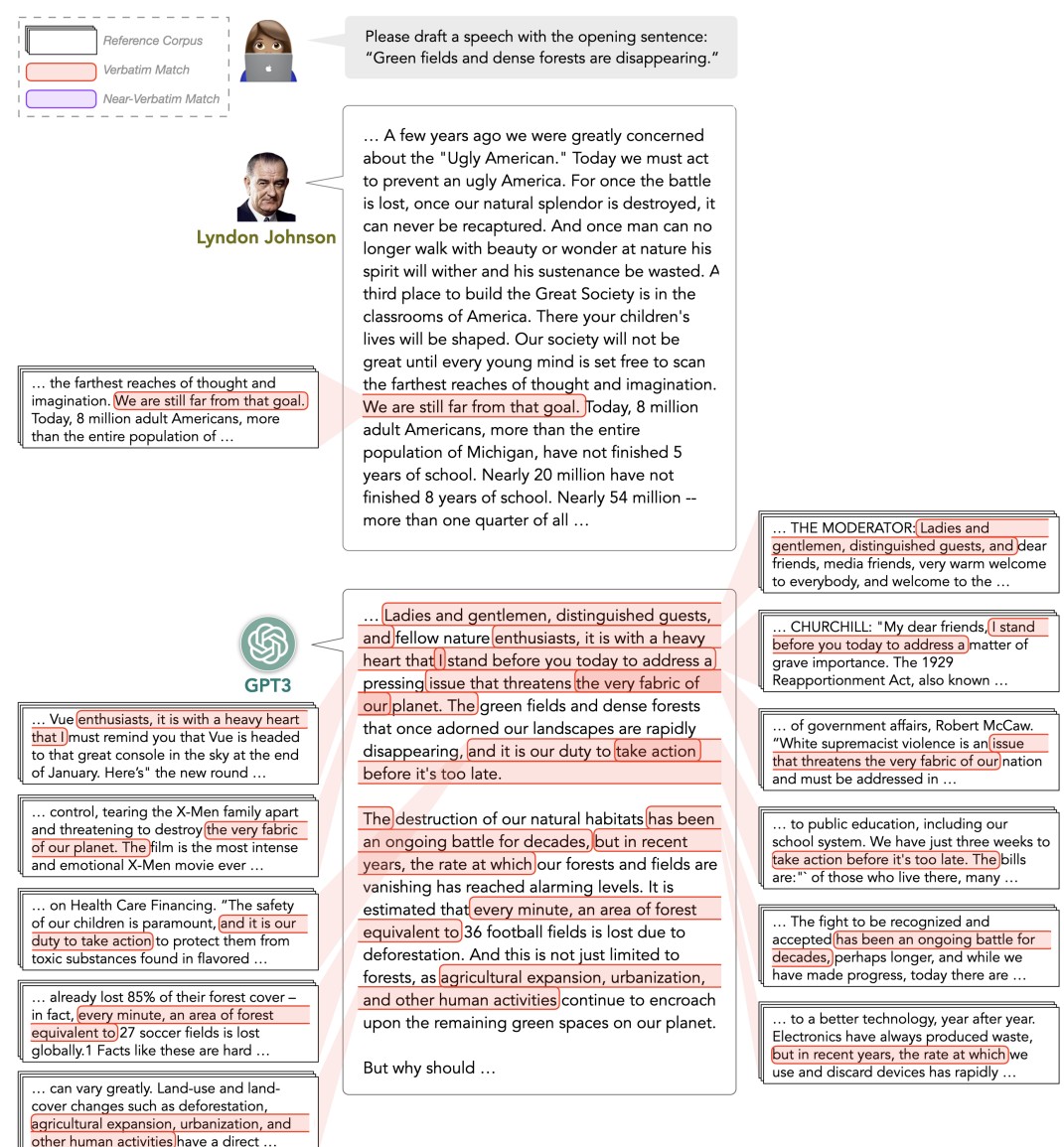

Figure 22: Example outputs from DJ SEARCH based on verbatim matches. We prompt LLMs to generate a speech starting with the opening sentence of a human speech transcript.

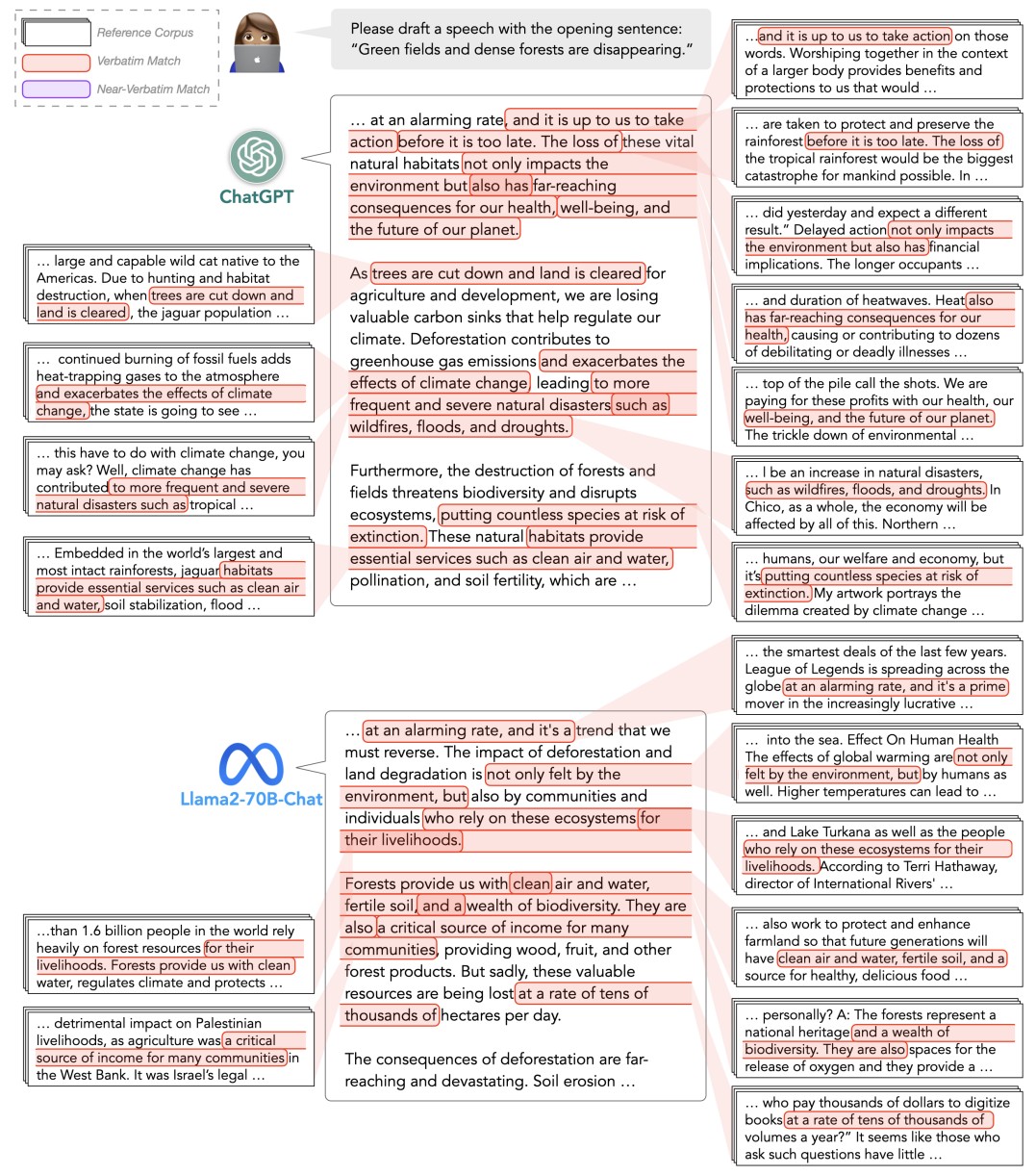

Figure 23: Example outputs from DJ SEARCH based on verbatim matches. We prompt LLMs to generate a speech starting with the opening sentence of a human speech transcript.

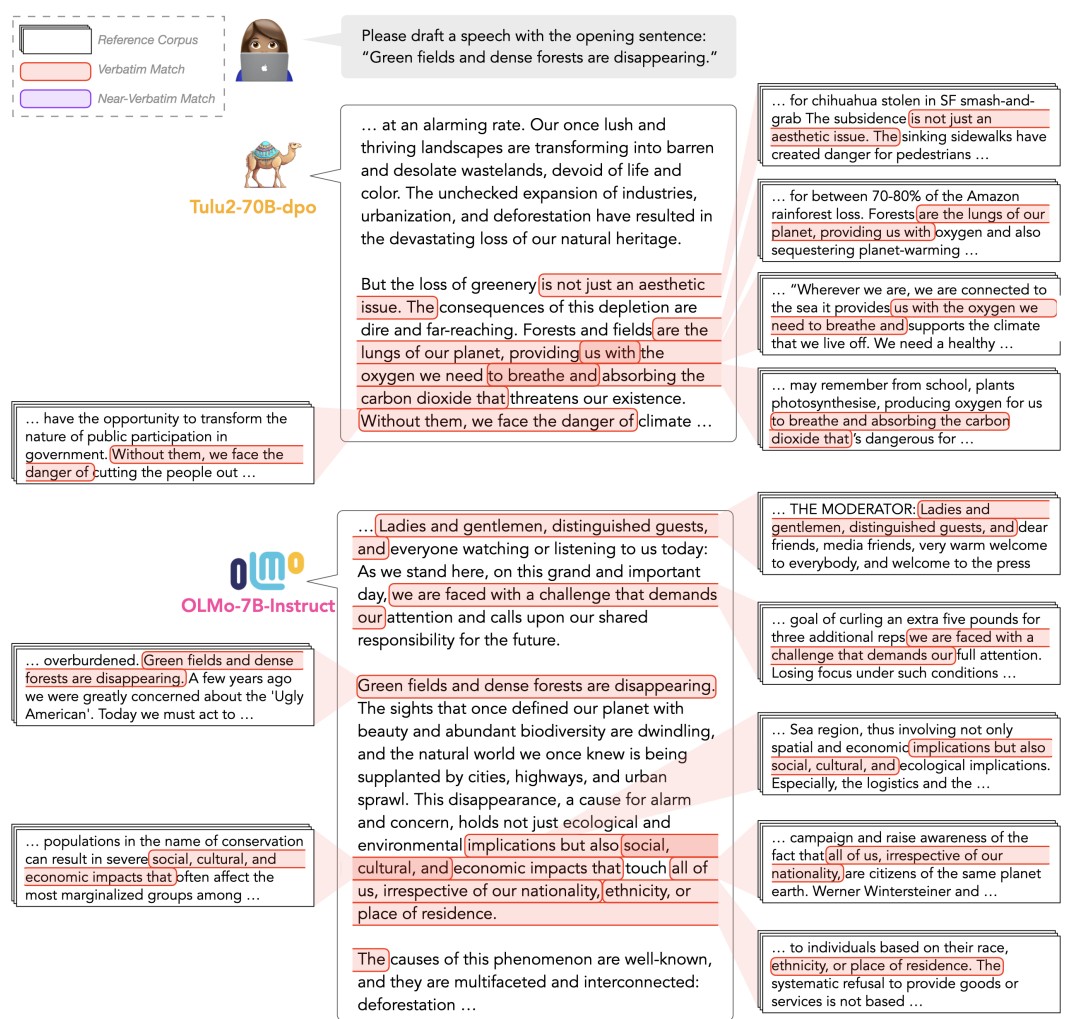

Figure 24: Example outputs from DJ SEARCH based on verbatim matches. We prompt LLMs to generate a speech starting with the opening sentence of a human speech transcript.

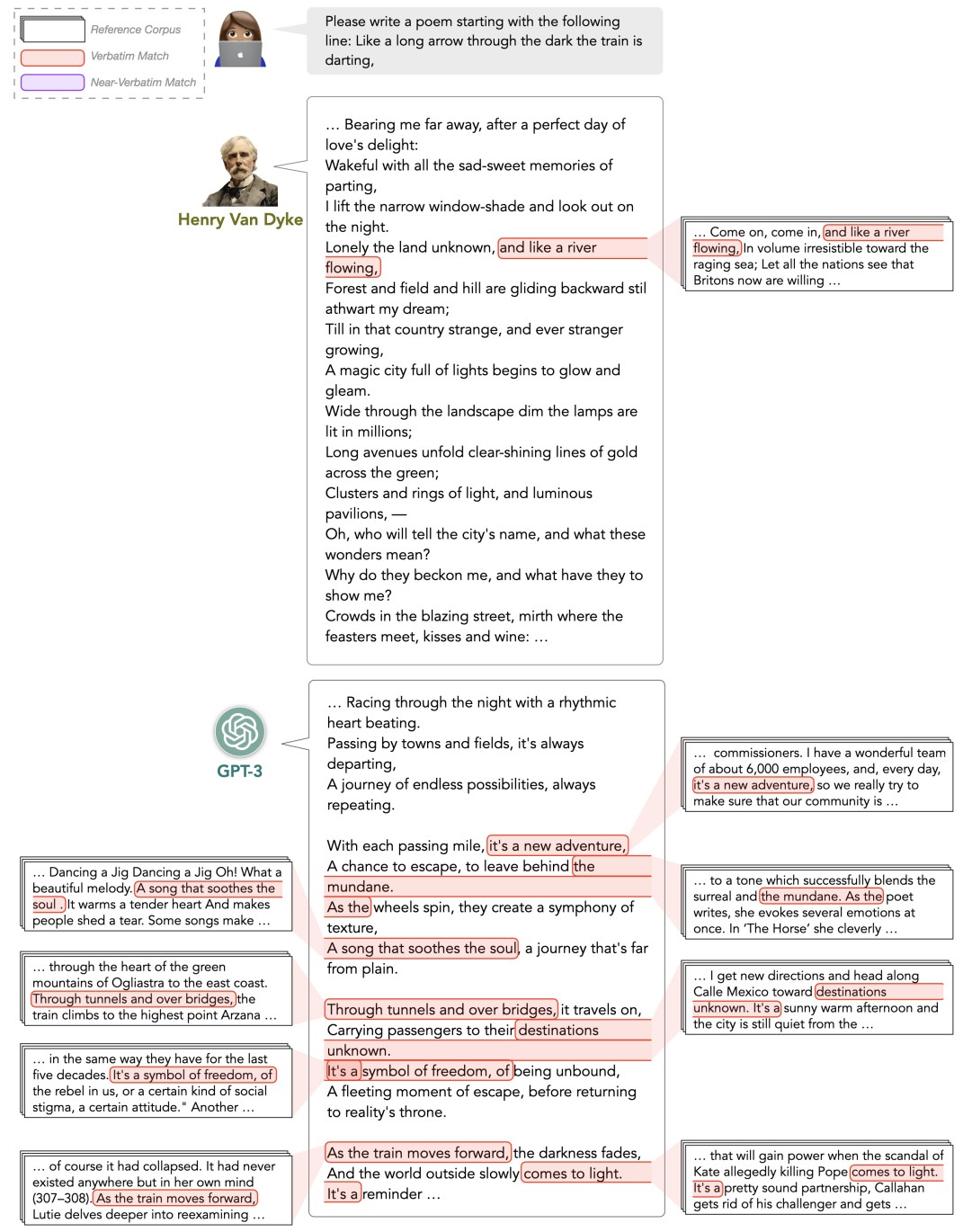

Figure 25: Example outputs from DJ SEARCH based on verbatim matches. We prompt LLMs to generate a poem starting with the first line of a human-written poem.

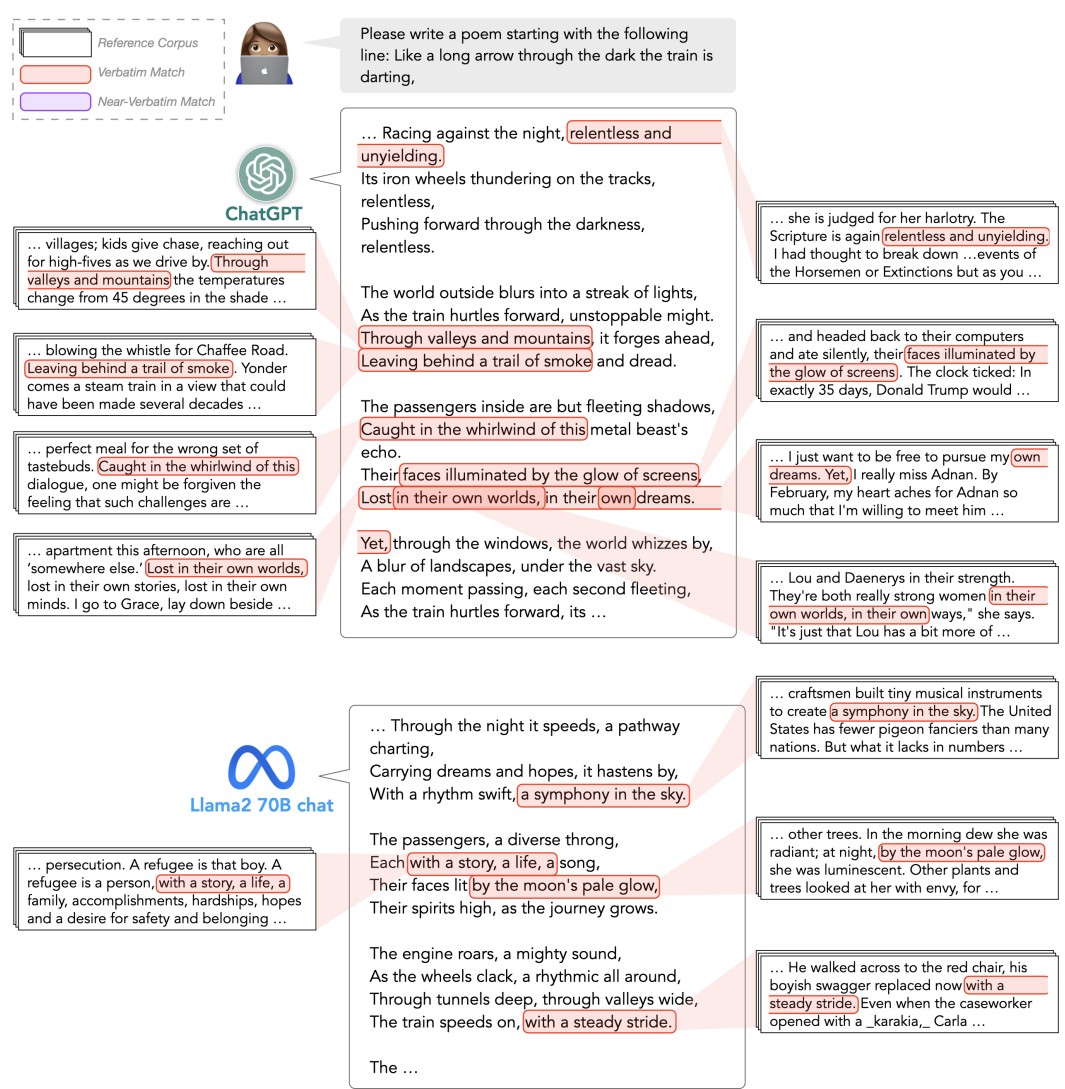

Figure 26: Example outputs from DJ SEARCH based on verbatim matches. We prompt LLMs to generate a poem starting with the first line of a human-written poem.

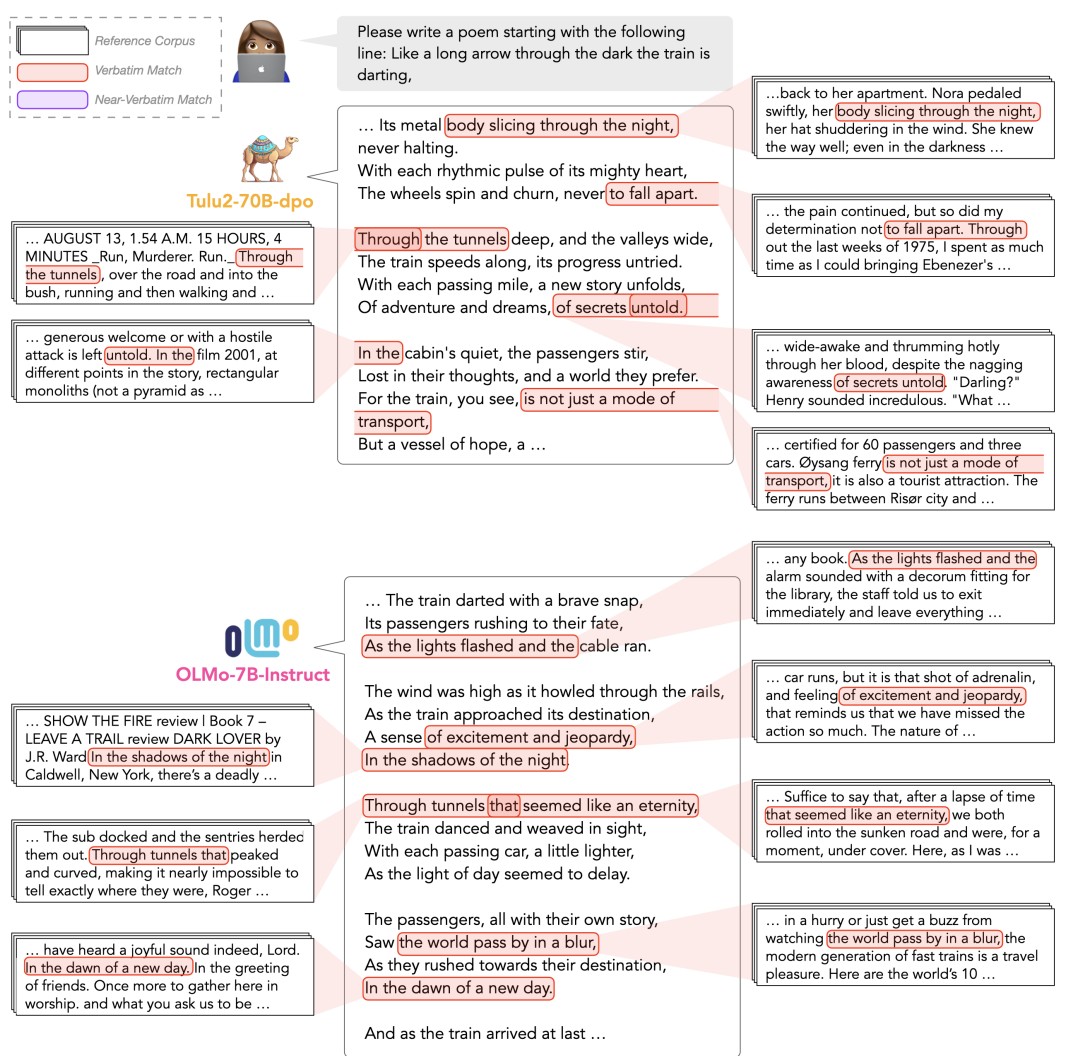

Figure 27: Example outputs from DJ SEARCH based on verbatim matches. We prompt LLMs to generate a poem starting with the first line of a human-written poem.

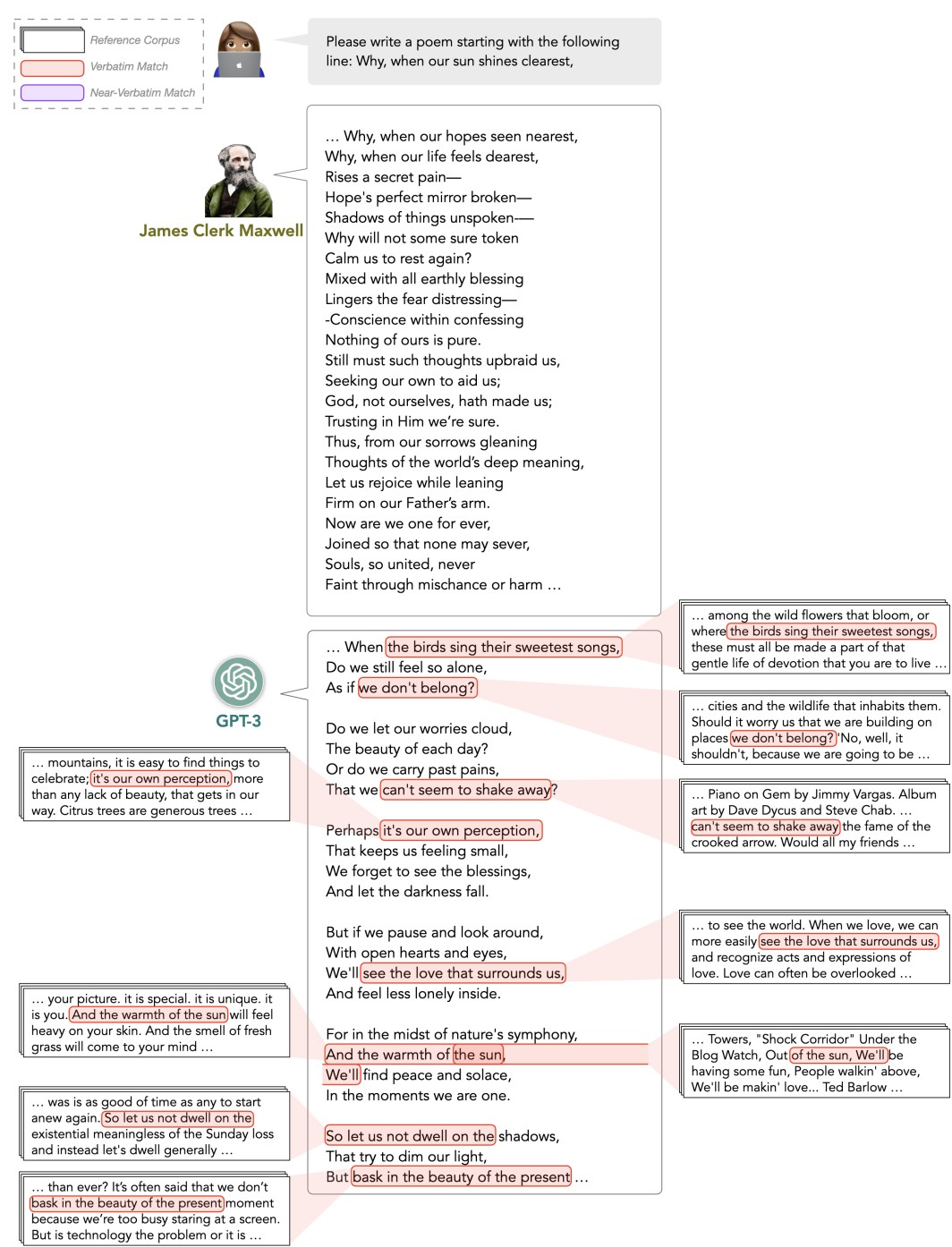

Figure 28: Example outputs from DJ SEARCH based on verbatim matches. We prompt LLMs to generate a poem starting with the first line of a human-written poem.

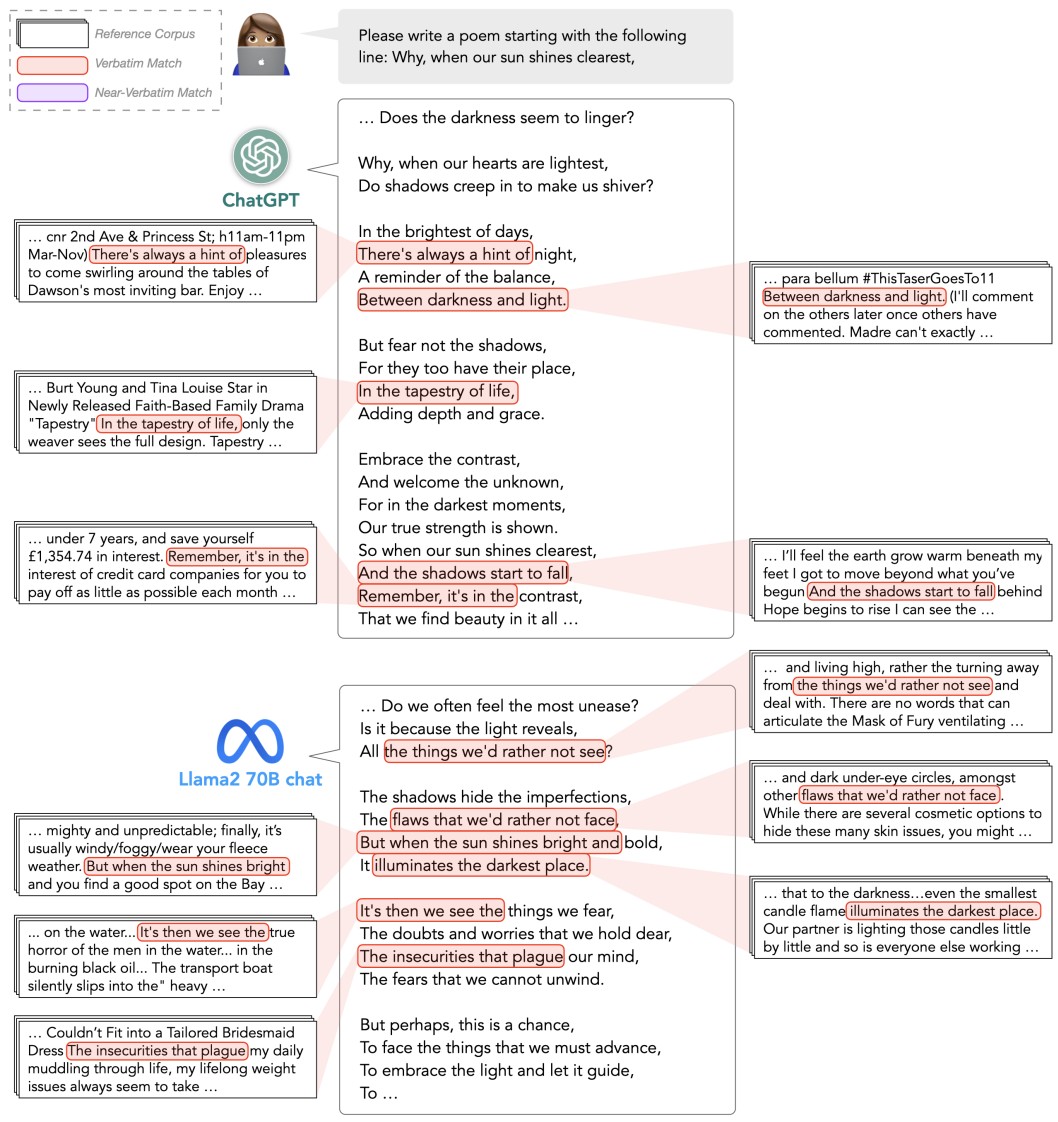

Figure 29: Example outputs from DJ SEARCH based on verbatim matches. We prompt LLMs to generate a poem starting with the first line of a human-written poem.

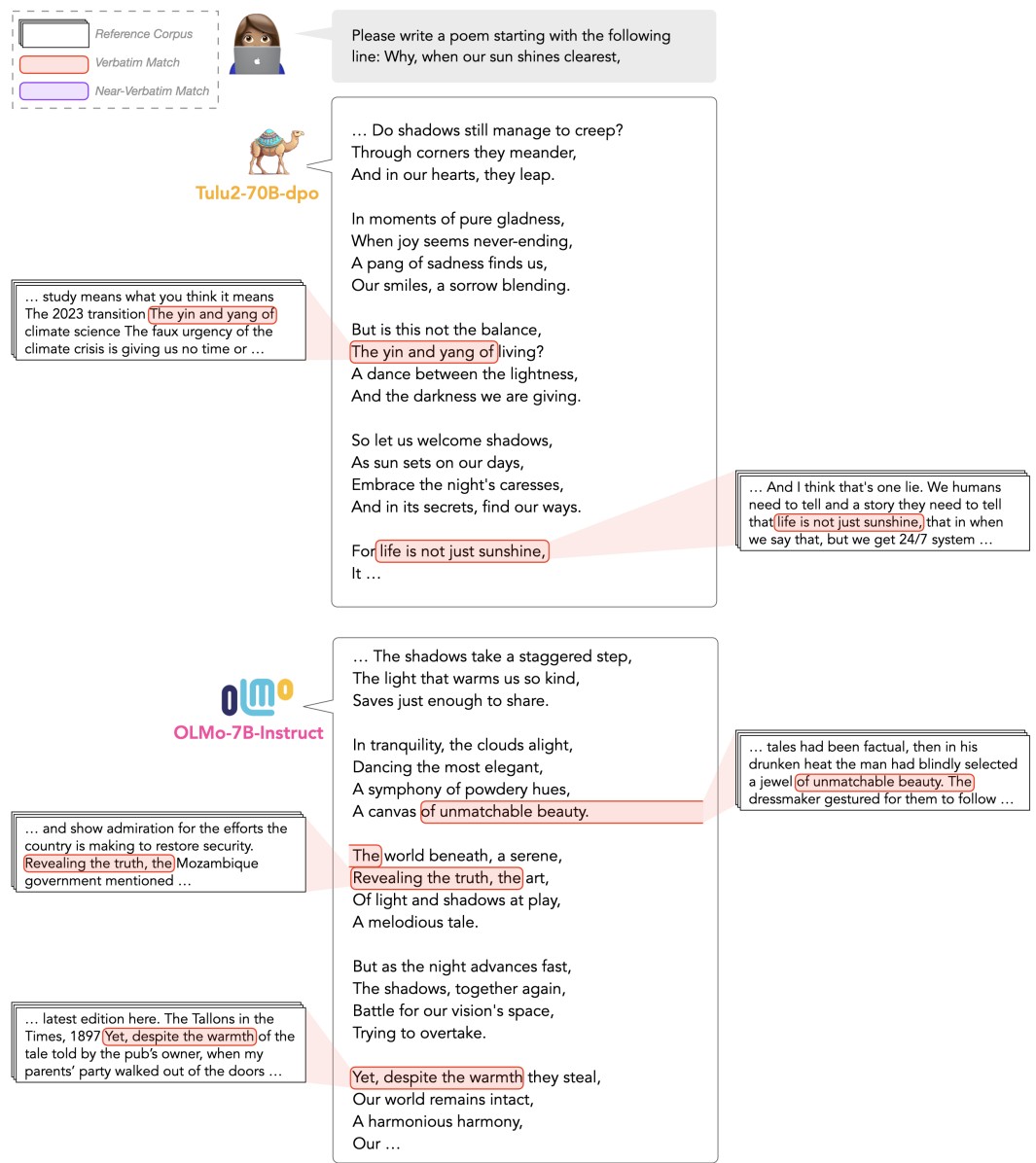

Figure 30: Example outputs from DJ SEARCH based on verbatim matches. We prompt LLMs to generate a poem starting with the first line of a human-written poem.

