# OpenReview forum: "AI as Humanity’s Salieri: Quantifying Linguistic Creativity of Language Models via Systematic Attribution of Machine Text against Web Text"
_ICLR.cc/2025/Conference — ICLR 2025 Oral_

### Official Review · Reviewer_8K9m · 2024-10-28

**Soundness:** 2
**Presentation:** 4
**Contribution:** 2
**Rating:** 6
**Confidence:** 5

**Summary:**

This paper tries to create a statistical metric to quantify creativity called CREATIVITY INDEX. The premise is fairly simple. They try to quantify creativity by checking what proportion of words are novel i.e those words are not part of ngrams that verbatim occur in pre-training data or is an approximate semantic match. They test a whole bunch of LMs both base as well as RLHF versions and experiment with various settings (nucleus sampling p, prompt format , model size). They also test on 3 type of writing tasks (novel, poetry, speech). The findings reveal that LLMs generated text have more verbatim matches from pre-training compared to human writing and hence they have lower CREATIVITY INDEX. The paper also show that CREATIVITY INDEX also adds as tool to detect machine generated text and outperforms DetectGPT and popular supervised baselines

**Strengths:**

The paper has many detailed experiments which I very much appreciate. Lots of graphs and figures to show different settings. The idea is very intuitive and from an engineering standpoint its of-course a very valuable contribution. Being able to trace text from pre-training is an excellent opportunity for the future of work and creative labor especially at a time when LLMs impact the livelihoods of professional writers. The structure and organization of the paper is also great.

**Weaknesses:**

There is so much to like about the paper. But there are obviously many limitations which makes me hesitant to give this paper a positive rating. I will try to be constructive here

1) From an experimental standpoint I think its important to fix top p to a higher value and vary temperature. Fig 4b plots CREATIVITY INDEX with respect to top p but we all know that when p = 0 the text is of course going to be repetitive. It does show that CREATIVITY INDEX increases with higher p. So this effect requires more rigorous testing where you fix the p and vary temperature

2) This is more technical term. But I do not think this paper is measuring Creativity. Authors mention "This metric is grounded in the notion of originality from creative thinking in psychology literature, which is defined as the statistical rarity of a response or an idea (Torrance, 1966; Crossley et al., 2016)." But Torrance Test has 4 broader components and just being Original is not enough for Creativity. This is my biggest complaint with the paper because the way it quantifies Creativity it would be super easy to game this metric by generating gibberish incoherent stuff that is not there in the pre-training. By authors' definition that is original wrt to Reference Corpus but is it Creative? I mean this as one of the biggest limitations of the paper. I do not think that novelty of ngram alone qualifies something to be more creative than the rest. So I request authors to update the paper and discuss these details. I think your metric quantifies Originality wrt Pre-training but that does not quantify creativity

3) One very big concern I had with this metric is how this can cause a lot of implicit bias towards writing that is simple yet evocative. I want to demonstrate some examples. I carefully checked that these texts occur in writings from other people (outside of the duplicate removal strategy which u applied)

   - **i) The context sentence**: "In the kitchen, she found herself reaching for the cabinet where her mother always kept the coffee, only to stop short"

     - **Expert Writer Written Continuation**: "This time, though, she was alone. Her mother would never come back."
     - **AI Written Continuation**: "The realization that she was alone here, truly alone, settled over her like a heavy blanket."

     Now obviously in context the AI written continuation contains a very cliched metaphor and is absolutely not original or creative in terms of writing. However, the human-written sentence has a richer subtext about the death of someone's mother which is expressed through simplicity and economical prose without using any flourish. Now I went to https://huggingface.co/spaces/liujch1998/infini-gram and copy-pasted the Human Written Continuation and found 6 occurrences of "This time, though, she was alone." and 94 occurrences of "mother would never come back." This means by your metric the expert-written continuation is less creative which I find untrue and somewhat problematic.

   - **ii) Coming to poetry**. Richie Hoffmann, one of the best contemporary poets, wrote a beautiful poem in *The New Yorker* titled "The French Novel" (https://www.newyorker.com/magazine/2019/04/08/french-novel). I mention the quatrain:

     > "[You were my second lover].
     > You had dark eyes and hair,
     > like a painting of a man
     > We lay on our stomachs reading books in your bed."

     This was published in 2019. Now I searched Infinigram and found a poetry published in 2017 had one same sentence verbatim (https://cosmicrubble.com/2017/03/11/false-dawn/):

     > "[You were my second lover] ever in life,
     > but I often wish you were my first"

     Now "dark eyes and hair" occurs 248 times in Infinigram; "like a painting of a man" occurs 24 times. "lay on our stomachs" occurs 604 times and "reading books in your bed" occurs 24 times. So based on your CREATIVITY INDEX you are saying the first quatrain is not Creative but they are originally written by an ex-Stanford Stegner Fellow.

   - **iii) Annie Ernaux** (Nobel Prize winner in Literature 2022) is popularly known for "Flat Writing." See this link https://www.nobelprize.org/prizes/literature/2022/ernaux/prose/

     Take the first excerpt from *Shame*: "One evening ‒ it was our last day ‒ in Tours, we had dinner in a brightly-lit restaurant where the walls were lined with mirrors, frequented by a sophisticated clientele. My father and I were seated at the end of a long table [....]"

     Infinigram Counts:
     - "it was our last day" → 8877
     - "dinner in a brightly-lit restaurant" → 1
     - "walls were lined with mirrors" → 147
     - "frequented by a sophisticated clientele" → 4
     - "My father and I were seated" → 15
     - "at the end of a long table" → 1826

     You need to be super careful with these sorts of claims in your paper. Because I think even though your intention is good, your metric accuses original writers or writers who aren't necessarily producing ngrams not in the reference corpus as not CREATIVE. There is a whole spectrum of writing. Verbatim matching text from pretraining does not reduce creativity; it’s more about juxtaposition of thoughts, how novel it is given the context/situation.

4) What impact does RLHF have on model creativity? This has already been studied by several papers so I don't see as any surprising thing.

5) How to measure creativity in LLMs trained on data outside of the reference corpus? Here again I appreciate that you found the similar models as GPT-4 cut off date like Gemma and Llama but I don't think you can use model-generated reference corpus. Especially there is evidence from ScaleAI that is used by Meta for RLHF data that workers themselves use ChatGPT / GPT4 https://www.wsj.com/tech/ai/alexandr-wang-scale-ai-d7c6efd7

6) This is a hasty and not well thought out claim "Our main finding is that classic literature exhibits a higher creativity level than the other two categories." What do you exactly mean by this? Books published in 2023 from the BookMIA (Shi et al., 2024) are a limited resource. In good faith, I cannot agree that books published now are less creative than Hemingway. That's like saying Murakami, Zadie Smith, Sally Rooney are less creative than Ernest Hemingway and William Faulkner. If you read literature, you know this is obviously not true.

This is in no way a harsh criticism but trying to make you aware of the issues that exist with any sort of creativity evaluation in Machine Learning Research. I look forward to your rebuttal, and I will see if I can increase scores.

**Questions:**

Have you considered doing some human eval on Creativity Ratings and measuring correlation with Creativity Index ?

---

> ### Comment · Reviewer_8K9m · 2024-11-25
> **DetectGPT**
>
> On another note authors claim about DetectGPT being the strongest zeroshot AI text detecting system is not true. Please refer to this paper (https://liamdugan.com/files/raid.pdf)
>
> Binoculars [https://arxiv.org/abs/2401.12070] is a lot better than DetectGPT. Actually DetectGPT is a lot worse than several content detectors.
>
> In the light of this and no rebuttal from authors inspite of several concerns raised I am leaning towards reducing my scores. Authors if you think there are issues with the points raised in my review please respond as the discussion period is ending

---

> ### Author Response · Authors · 2024-11-26
>
> We thank the reviewer for their detailed feedback, constructive critique, and recognition of the importance of our work in exploring how “trace text from pre-training is an excellent opportunity for the future of work and creative labor.” We also appreciate the acknowledgment of our “detailed experiments,” the “intuitive” nature of our idea, the “great structure and organization of the paper,” and the “very valuable contribution from an engineering standpoint.” Below, we address the key concerns and questions raised.
>
> ## Re: Ablation for Decoding Temperature
>
> Thank you for the suggestion to experiment with decoding temperature! We fixed p=1.0, varied the decoding temperature from 0 to 1, and computed the CREATIVITY INDEX. We observed that a higher temperature resulted in a marginally higher CREATIVITY INDEX. Specifically, ChatGPT's CREATIVITY INDEX was 8.4% higher with a decoding temperature of 1.0 compared to 0 (p = 0.16; N = 600). However, this difference was minimal compared to the gap between humans and LLMs. The CREATIVITY INDEX of human authors was 49.1% higher than that of ChatGPT at a decoding temperature of 1.0 (p = 6.9 × 10^{-17}; N = 600).
> ## Re: Whether Proposed CREATIVITY INDEX Measure Creativity
>
>
> Thank you for the suggestion to expand on this discussion! We respond below and have also updated the paper draft to to include additional discussion.
>
> As we mentioned in the paper, CREATIVITY INDEX is designed to measure one specific aspect of creativity—linguistic creativity (i.e., the novelty in composing words and phrases). This is approximated by computing originality of the text with respect to a large pre-training corpus.  While effective at capturing statistical differences between professional human writing and seemingly remarkable outputs from LLMs, it may not comprehensively reflect other dimensions of creativity, such as flexibility or elaboration as featured in the Torrance Test. Therefore, the CREATIVITY INDEX alone is insufficient for drawing definitive conclusions about overall creativity, especially when analyzing nuanced variations among human authors.
>
> Additional factors, such as writing style and the time of composition, can also influence the CREATIVITY INDEX. For example, older English writings may exhibit higher scores, as they are more difficult to reconstruct from the pretraining corpora of modern English. Furthermore, CREATIVITY INDEX assumes the input text is of sufficient quality, as our study focuses on outputs from recent LLMs that are already fluent and coherent. When applied to broader contexts, the CREATIVITY INDEX can be complemented by standard automatic measures of text quality, such as fluency classifiers or perplexity-based evaluations, to provide a more comprehensive assessment. We have incorporated discussions of these considerations into the paper.
>
> To further validate the CREATIVITY INDEX, we computed its correlation with existing human evaluation-based creativity metrics, specifically the Torrance Test of Creative Writing (TTCW) proposed by Chakrabarty et al. [1] This framework evaluates the creativity of stories written by LLMs and human authors using expert evaluations based on a carefully designed rubric inspired by the Torrance Test of Creative Thinking (TTCT) (**See General Response**).
>
>
> We found that the CREATIVITY INDEX positively correlates with all dimensions of TTCW, showing particularly strong correlations with high statistical significance in dimensions such as “Emotional Flexibility,” “Narrative Pacing,” “Structural Flexibility,” and “Originality in Thought.”
> Again, as the CREATIVITY INDEX is specifically designed to measure linguistic creativity, it may correlate with other dimensions of creativity but does not comprehensively capture all of them. Therefore, it serves as a complementary metric to existing creativity measurement methods.
>
> [1] Chakrabarty, Tuhin et al. “Art or Artifice? Large Language Models and the False Promise of Creativity.” Proceedings of the CHI Conference on Human Factors in Computing Systems (2023): n. pag.
>
> **(1 / 3, to be continued)**

---

> ### Author Response · Authors · 2024-11-26
> **Re: DetectGPT**
>
> Thank you for bringing up more recent work in zero-shot machine text detection! We have revised the paper to remove the claim that DetectGPT is the strongest zero-shot detector. In the next revision, we will include comparisons with more recent detection systems, such as Binoculars. The primary goal of this paper is to analyze statistical differences between professional human writing and seemingly amazing outputs from LLMs. Applying the CREATIVITY INDEX to machine text detection is intended as an exploratory effort for novel applications.
>
> We also apologize for the delayed response. The additional experiments took a while to conclude due to limited resources. Additionally, the extension period has been extended to December 2nd!

---

### Official Review · Reviewer_ZD3D · 2024-11-02

**Soundness:** 3
**Presentation:** 4
**Contribution:** 4
**Rating:** 8
**Confidence:** 3

**Summary:**

The authors present two methods for analyzing text creativity:

DJ Search - A clever algorithm for finding the longest novel n-grams in a text by checking against a reference corpus

This paper is likely to have significant practical impact. Unlike many ML papers that require a cluster of A100s just to run inference, this could actually be deployed in real-world applications. The algorithmic approach means it can run on modest hardware and scale linearly with input size.

Plus, it's refreshing to see someone tackle the creativity measurement problem from first principles rather than just throwing more parameters at it. The information retrieval perspective brings some much-needed rigor to a field that's been getting a bit hand-wavy lately.

Accept. This paper makes a solid contribution to the field with practical, implementable algorithms. While there are some limitations, the core ideas are sound and the engineering considerations are thoroughly addressed. It's the kind of paper that will actually be useful to practitioners, not just citation fodder.

This is what good systems papers should look like - solid theoretical foundations, practical implementation details, and clear engineering considerations. Accept with enthusiasm, and looking forward to the GitHub repo (you are open-sourcing this, right?).

**Strengths:**

The paper's main contribution isn't just another transformer architecture trained on twice as much data - it's a genuinely novel algorithmic approach to measuring text creativity. The DJ Search algorithm is particularly elegant, reducing what could be an O(n²) operation to O(n) through a smart two-pointer approach. As someone who appreciates algorithmic elegance, this sparks joy.

The implementation details in Appendix A are great - they actually thought through the real-world engineering challenges instead of handwaving them away with "we leave this as an implementation detail." The optimization of Word Mover's Distance computation is particularly well done, showing they really care about making this practical.

They've included all the nitty-gritty details about corpus deduplication, handling edge cases, and optimizing the search process. This isn't just academic fluff - it's the kind of paper that an engineer could actually implement from.

The authors clearly understand the scale challenges of working with massive reference corpora. The BM25 pre-filtering approach for WMD is a smart compromise between accuracy and computational efficiency.

**Weaknesses:**

Classic ML paper moment - everything's tested on English only. Would love to see how this performs on languages with different n-gram distributions or morphological complexity.

The paper waves away the choice of α=0.9 and β=0.3 thresholds for corpus deduplication with a "in practice, we set..." Hand-waving intensifies. Some ablation studies would've been nice here.

The method's effectiveness is heavily dependent on the quality and coverage of the reference corpus. While they address this somewhat with the deduplication approach, there could be more discussion of how to maintain and update these corpora over time.

**Questions:**

Where is the github repo with code to make it easy to run this?

---

> ### Author Response · Authors · 2024-11-27
>
> We thank the reviewer for their encouraging and constructive feedback! We are delighted to hear that they consider our work to have “significant practical impact,” and we greatly appreciate their recognition of its “solid theoretical foundations, practical implementation details, and clear engineering considerations.”
>
> ## CREATIVITY INDEX on Multilingual Data
>
> Thank you for the suggestion to expand our analysis to multilingual data! We conducted additional experiments to compute the CREATIVITY INDEX for ChatGPT and human authors in French and German within the novel domain.
>
> For human-written French novels, we randomly sampled 10 books from the “Contemporary French Books” list on Goodreads (https://www.goodreads.com/shelf/show/contemporary-french). From each book, we randomly selected 10 snippets (i.e. continuous sequences of K sentences with a total length exceeding 256 words) as the human text. Similarly, for German novels, we randomly sampled 10 books from the “Modern German Novel Books” list on Goodreads (https://www.goodreads.com/shelf/show/modern-german-novel). For machine texts, we prompted ChatGPT to generate continuations for these novels starting with an initial sentence from the human writings, following the same setup as in the paper.
>
> As a reference corpus, we used the Common Corpus from PleIAs, the most recent multilingual pre-training dataset. Due to time and resource constraints, we were not able to build an Infini-gram search index for the entire corpus. Instead, we subsampled approximately 20B tokens each from the French and German portions of the Common Corpus to serve as the reference.
>
> We found that human writings consistently exhibit a significantly higher CREATIVITY INDEX than ChatGPT in both French and German, aligning with our main results. Specifically, the CREATIVITY INDEX of human authors was 46.1% higher than ChatGPT in French novel writing (p = 7.5 * 10^{-13}; n = 100) and 54.3% higher in German (p = 1.2 * 10^{-15}; n = 100). We plan to expand these experiments to additional languages in the next revision.
>
> ## Additional Ablation for Deduplication Threshold
>
> Thank you for the suggestion to expand our ablation on deduplication thresholds! We analyzed the relationship between the degree of overlap S(x,d) and the likelihood that the reference document d contains an exact copy or quotations of the input text x.
>
> | overlap degree S(x,d)    | 0.1 | 0.2  | 0.3  | 0.4  | 0.5  | 0.6  | 0.7  | 0.8  | 0.9  | 1.0 |
> |--------------------------|-----|------|------|------|------|------|------|------|------|-----|
> | percentage of exact copy | 0.0 | 0.0  | 0.0  | 0.03 | 0.12 | 0.28 | 0.49 | 0.77 | 0.94 | 1.0 |
> | percentage of quotation  | 0.0 | 0.02 | 0.04 | 0.11 | 0.15 | 0.13 | 0.05 | 0.02 | 0.0  | 0.0 |
>
> We found that when S(x, d) < 0.3, there are no exact copies, and almost no quotations of the input text x in the reference document d. Conversely, when S(x,d) > 0.9, the reference document d almost always contains an exact copy of x. For cases where 0.3 < S(x, d) < 0.9, all three scenarios (exact copy, quotation, or none) are plausible. Therefore, we chose to prompt an LLM to determine whether d contains exact copies or quotations of x in this range, ensuring accurate deduplication.
>
>
> ## Maintain and Update Reference Corpus
>
> The effectiveness of our method indeed relies on the quality and coverage of the reference corpus. Building a high-quality and high-coverage pre-training corpus remains an important research question that the community is actively addressing with notable progress. At the time of our experiments, RedPajama was the largest open-source pre-training corpus available. Since then, newer large-scale pre-training corpora, such as DOLMA [1], have been released and incorporated into the infini-gram search API. Additionally, Common Corpus from PleIAs, which includes extensive multilingual data, is another resource we plan to integrate into the infini-gram search API.
>
> We will continue to monitor advancements and integrate new pretraining corpora into DJ SEARCH as part of the reference corpus. Additionally, one of our future directions is to create our own large-scale, reliable reference corpus with fast search capability.
>
> ## Open-Source DJ SEARCH
>
> Yes! We will definitely open-source the code, and we also provide an online demo of DJ SEARCH for easy and accessible use.
>
>
> [1] Soldaini, Luca et al. “Dolma: an Open Corpus of Three Trillion Tokens for Language Model Pretraining Research.” ArXiv abs/2402.00159 (2024): n. Pag.
>
> [2] PleIAs. “PleIAs/Common_corpus · Datasets at Hugging Face.” PleIAs/common_corpus · Datasets at Hugging Face. Accessed November 26, 2024. https://huggingface.co/datasets/PleIAs/common_corpus.

---

> > ### Comment · Reviewer_ZD3D · 2024-11-29
> > **Thank you to the authors.**
> >
> > Thank you for running additional experiments and resolving my concerns. This is a good paper, but raising to 10 is a very high bar. I will maintain my rating but will slightly raise the other scores in recognition.

---

### Official Review · Reviewer_5RYT · 2024-11-02

**Soundness:** 3
**Presentation:** 4
**Contribution:** 3
**Rating:** 6
**Confidence:** 5

**Summary:**

The author presents Creativity Index that quantifies the linguistic creativity of text by reconstructing it from n-gram text segments from web sources.  The proposed indexing algorithm combines strict verbatim matching on text snippets and near-verbatim semantic matching on word embeddings. They have conducted comprehensive experiments using the proposed index to evaluate creativity between various human groups and both open-source and proprietary LLMs.
The findings indicate that the current generative model reduces the ability to produce original and unique content, both verbatim and semantically, from human writing. An intriguing findings by the author is that preference alignment could significantly harm the model's creativity, decreasing it by 30% compared to the pre-trained model.
Overall, it is a well-organized and comprehensive research on the quantified assessment of model creativity.

**Strengths:**

- Well-crafted paper that clearly demonstrates the metric algorithm, experimental settings, and conclusions.
- The motivation is well described at the beginning of this paper. The author managed to implement an effective method that is greatly aligned with the intention.
- The author conducted a multifaceted and multiangled extension of discussions, directly comparing LM and humans, exploring various matching algorithms, assessing RLHF on creativity, and potential usage on AI-generated text detection.

**Weaknesses:**

The metric method proposed in this article does not adequately support some of the findings presented. The main issues are as follows:
- Timeliness of Data: The majority of the text provided by RedPajama originates from internet sources dated between 2014 and 2023, with no precise creation dates included in the data. Many works by authors from past decades lack a sufficiently reliable text history to effectively apply the Creativity Index for analysis. Thus, it cannot be concluded that distinguished authors necessarily possess high levels of creativity.
- Depth of Matching Method Exploration: The article’s proposed method, which combines text and semantic matching, lacks an analysis of how each method contributes to the effectiveness of the metrics.
- Lack of Baseline Metric Comparison and Analysis

The author mentioned the proposed DJ search using the two pointers to achieve linear search complexity. An analysis on searching common segments in the database would inform readers.

**Questions:**

Please see the weaknesses

---

> ### Author Response · Authors · 2024-11-30
>
> We thank the reviewer for their constructive comments and appreciate the acknowledgment of the paper as “well-crafted,” “greatly aligned with the intention,” and conducting a “multifaceted and multiangled extension of discussions.”
>
> ## Re: Timeliness of Data
>
> As mentioned in the original draft, “the computation of the CREATIVITY INDEX is constrained by the reference corpus used for DJ SEARCH.” We acknowledged that “Without incorporating private data used to train closed source LLMs such as ChatGPT into the reference corpus of DJ SEARCH, the CREATIVITY INDEX of closed-source LLMs may be somewhat inflated.” Similarly, we agree that human writing from older times might also exhibit somewhat inflated CREATIVITY INDEX, as the reference corpus may not cover older English as comprehensively as modern English. We have added further discussion on this topic in the paper.
>
> The main focus of this work is to understand the differences between seemingly amazing machine text and professional human writing by quantifying some aspect of linguistic creativity through the lens of n-gram comparisons with web text. Precisely distinguishing creativity differences within human writing remains a challenging research problem [1, 2, 3], which is beyond the primary scope of this work.
>
> Nonetheless, it is interesting to explore whether the CREATIVITY INDEX remains robust when the reference corpus has much smaller coverage. To investigate, we conducted experiments using the older pre-training corpus, Pile, which is approximately 1/5 the size of RedPajama. Even with this much smaller reference corpus, we found that human writings still exhibit a significantly higher CREATIVITY INDEX than any LLM. Specifically, based on the Pile, the average CREATIVITY INDEX of humans was 42.8% higher than that of LLMs in novel writing (p=1.4 × 10^{-26}; n = 100), consistent with our main findings.
>
>
> ## Re: Depth of Matching Method Exploration
>
> In the original draft, we conducted detailed experiments to analyze the effect of semantic matching versus exact matching, as described in the experiment section “How do different matching criteria affect creativity measurement?”
>
> We found that: “First, the creativity gap between humans and LLMs becomes even larger when considering semantic matches in addition to verbatim matches (Fig. 3d). Averaged across all models, the CREATIVITY INDEX of human, based on both verbatim and semantic matches, is 102.5% higher than LLMs in novel writing (p = 2.6 × 10−12; N = 600), whereas based on verbatim matches alone, the CREATIVITY INDEX of human is 52.2% higher than LLMs. Second, semantic matches provide more signal for analyzing the uniqueness of longer n-grams (Fig. 3e). For example, while the gap in L-uniqueness at L = 11 between human text and machine text from OLMo Instruct is 3.7% based on verbatim matches alone, this gap widens to 16.3% when considering both verbatim and semantic matches (p = 3.1 × 10−7 ; N = 600). This indicates that although some of the longer n-grams in machine text may appear unique at the verbatim level, they are similar to certain text snippets in the reference corpus at the content level.”

---

> > ### Author Response · Authors · 2024-11-30
> >
> > ## Re: Lack of Baseline Metric
> >
> > Thank you for the suggestion! To our knowledge, we could not find a directly comparable baseline that statistically reveals some aspects of creativity differences between humans and LLMs.
> >
> > Instead, we compared our approach with previous human-evaluation based creativity measures. Please refer to the **General Response** for more details.
> >
> >
> > ## Re: Searching Common Segments in the Database
> >
> > The closest work we can find for searching common segments in datasets is What’s in my Big Data [4], which identifies the most common n-grams in pre-training corpora like RedPajama. While this approach provides some insights into a specific pre-training dataset, it is not directly applicable to our focus. Our work aims to understand the differences between seemingly amazing machine texts and professional human writings by statistically comparing the overlap of their n-grams with a pretraining corpus to assess their level of linguistic creativity.
> >
> > Nonetheless, we conducted additional runtime analysis of DJ SEARCH. To verify linear complexity, we measured the runtime of DJ SEARCH with respect to increasing input text lengths using RedPajama as the reference corpus, averaging the results over 500 examples. The results are as follows:
> >
> > | Input Length      | 64  | 128  | 256  | 512  |
> > |-------------------|-----|------|------|------|
> > | Runtime (seconds) | 9.2 | 16.9 | 30.7 | 62.3 |
> >
> > We observed that the runtime of DJ SEARCH indeed increases linearly with the length of the input text.
> >
> > We also measured the runtime of DJ SEARCH with respect to increasing sizes of the reference corpus while keeping the input length fixed at 256 words. In addition to RedPajama, which contains approximately 1.4 trillion tokens, we experimented with the Pile (380B tokens) and DOLMA (2.6 trillion tokens). The results are as follows:
> >
> > | Reference Corpus Size | 380B Tokens | 1.4T Tokens | 2.6T Tokens |
> > |-----------------------|-------------|-------------|-------------|
> > | Runtime (seconds)     | 24.1        | 30.7        | 37.1        |
> >
> > We found that the runtime increase for DJ SEARCH is minimal as the size of the reference corpus grows. This is likely due to the high optimization of the Infini-gram, which enables it to retrieve matched n-grams within milliseconds.
> >
> > [1] EL-MURAD, JAAFAR, and DOUGLAS C. WEST. “The Definition and Measurement of Creativity: What Do We Know?” Journal of Advertising Research 44.2 (2004): 188–201. Web.
> >
> > [2]Sternberg, Robert J., and Todd I. Lubart. “An Investment Theory of Creativity and Its Development.” Human Development, vol. 34, no. 1, 1991, pp. 1–31. JSTOR, http://www.jstor.org/stable/26767348. Accessed 29 Nov. 2024.
> >
> > [3] Boden, M. “Dimensions of creativity.” The Journal of Aesthetic Education 30 (1996): 120.
> >
> > [4] Elazar, Yanai et al. “What's In My Big Data?” ArXiv abs/2310.20707 (2023): n. pag.

---

> > > ### Comment · Reviewer_5RYT · 2024-12-02
> > > **Thanks for the responses**
> > >
> > > Thanks for the author's responses to the experimental results. I raised the presentation score and recommend accepting this paper.

---

### Official Review · Reviewer_Kcvs · 2024-11-02

**Soundness:** 4
**Presentation:** 4
**Contribution:** 3
**Rating:** 8
**Confidence:** 3

**Summary:**

The paper discusses work on measuring the creativity of a language model’s generated content using a large compilation of web texts as reference corpus called Creativity Index. This metric quantifies how much the machine-generated text is remixed and matched from the reference corpus through an n-gram based analysis (re: L uniqueness, # of words outside n-grams normalized by words in text). This is then packaged into a DP-based search mechanism called DJ Search that efficiently attributes pieces of texts from the reference corpus. The paper has a number of well-discussed, well-visualized interesting results. This includes humans showing a consistent higher level of creativity compared to LLMs (52% higher than LLMs in novel writing, 31.1% higher in poetry composition, 115.3% higher in speech drafting). Supporting ablation experiments were also done by the authors on quantifying how much RLHF hurts creativity (30+% reduction), measuring creativity outside of the scope of the reference corpus, variance across humans’ level of creativity, and comparison to existing machine generated text detectors.

Overall I think this is a good paper for ICLR and has the level of completeness and technical rigor required for the conference.

**Strengths:**

The paper presents a good empirical investigation on the creativity (and and lack of) of large language models compared to creative inputs from humans. The proposed Creative Index is simple and easy to understand. The paper is readable, contains quality visualizations to complement experiment results, and has the level of completeness in terms of depth of experiments, discussion coverage, and insights. I also like and agree with the author/s’s analogy regarding how LLMs can be similar to DJ remixing information from different sources rather than a novelty-driven agent like real humans. Overall, the paper is a good addition to the conference.

**Weaknesses:**

I don’t have serious concerns from the paper, but a number of points can be considered to improve the overall quality of the work:

1. The first thing that readers will think of the proposed DJ Search is that searching for any form of patterns from an extremely large corpus such as RedPajama would be time inefficient (esp with size of the data). Even if the author/s say this is efficient, it should be emphasized in the main body of the paper itself including details of time complexity as I feel this part is not that stark / outright upon reading the paper. It might also be a nice-to-have to provide visualizations on how much time it would take for the algorithm to complete processing a sample dataset compared to other methods of searching in a visualized manner. The paper has some great visualizations so this part would be minimum effort.

2. Evaluating LLM’s quantified creativity is not new, therefore readers would expect some comparison between what is being proposed (Creativity Index) and what other creativity metrics are in literature. For example, does Creativity Index observe or correlate with some of the results from other tests such as Torrance Test of Creative Writing (TTCW, CHI2024) (https://arxiv.org/pdf/2309.14556)?  The authors can use the exact same test set with results from human annotations from this work and run them through the Creativity Index formula to analyze the results and identify which of the 14 aspects correlate together.

3. The vanilla version of WMD is old can faster variations have been proposed through the years including relaxed version (https://github.com/src-d/wmd-relax) and supervised version (https://papers.nips.cc/paper_files/paper/2016/hash/10c66082c124f8afe3df4886f5e516e0-Abstract.html). I suggest the authors explore these variations and document improvements to the paper.

**Questions:**

1. Can WMD be switched with simple Euclidean/Sinkhorn-based distance measures? What's the author/s' thoughts on this? Are there are any possible downgrades?
2. Is RedPajama the largest webcrawled data to use as reference? How about RefineWeb (https://arxiv.org/abs/2306.01116)?

---

> ### Author Response · Authors · 2024-11-29
>
> We thank the reviewers for their encouraging and constructive feedback! We appreciate the recognition of the “simple and easy to understand” idea, the “level of completeness in terms of depth of experiments, discussion coverage, and insight,” as well as the “quality visualization.”
>
> ## Re: Runtime of DJ SEARCH
>
> Thank you for the suggestion to include additional runtime analysis!
>
> To verify linear complexity, we measured the runtime of DJ SEARCH with respect to increasing input text lengths using RedPajama as the reference corpus, averaging the results over 500 examples. The results are as follows:
>
> | Input Length (words)     | 64  | 128  | 256  | 512  |
> |-------------------|-----|------|------|------|
> | Runtime (seconds) | 9.2 | 16.9 | 30.7 | 62.3 |
>
> We observed that the runtime of DJ SEARCH indeed increases linearly with the length of the input text.
>
> We also measured the runtime of DJ SEARCH with respect to increasing sizes of the reference corpus while keeping the input length fixed at 256 words. In addition to RedPajama, which contains approximately 1.4 trillion tokens, we experimented with the Pile (380B tokens) and DOLMA (2.6 trillion tokens). The results are as follows:
>
> | Reference Corpus Size | 380B Tokens | 1.4T Tokens | 2.6T Tokens |
> |-----------------------|-------------|-------------|-------------|
> | Runtime (seconds)     | 24.1        | 30.7        | 37.1        |
>
> We found that the runtime increase for DJ SEARCH is minimal as the size of the reference corpus grows. This is likely due to the high optimization of the Infini-gram, which enables it to retrieve matched n-grams within milliseconds.
>
> ## Re: Compare with Previous Work
>
> Thank you for the suggestion! We compared our CREATIVITY INDEX with the Torrance Test of Creative Writing (TTCW) proposed by Chakrabarty et al. Please refer to the General Response for further details.
>
> We found that the CREATIVITY INDEX positively correlates with all dimensions of TTCW, showing particularly strong correlations with high statistical significance in dimensions such as “Emotional Flexibility,” “Narrative Pacing,” “Structural Flexibility,” and “Originality in Thought.”
>
> ## Re: Variation of WMD
>
> Thank you for the suggestion! We experimented with different variations of transport-inspired distances, including relaxed WMD, supervised WMD [1], and Sinkhorn distance [2].
>
> We computed the CREATIVITY INDEX for ChatGPT and human writing in the novel domain based on semantic matches using relaxed WMD, supervised WMD, and Sinkhorn distance. Regardless of the distance metric used, human writings consistently exhibited a significantly higher CREATIVITY INDEX than ChatGPT, aligning with our main results. Specifically, based on relaxed WMD, the CREATIVITY INDEX of human authors was 97.6% higher than ChatGPT (p=2.5 × 10^{-13}; n=100). Based on supervised WMD, the CREATIVITY INDEX of human authors was 79.3% higher than ChatGPT (p=1.9 × 10^{-7}; n = 100). Based on Sinkhorn distance, the CREATIVITY INDEX of human authors was 91.3% higher than ChatGPT (p=3.5 × 10^{-12}; n = 100).
>
> As mentioned in the original draft, the motivation for using transport-inspired distances—which combine word embedding distances for each n-gram's words rather than directly computing the Euclidean distance between embeddings of two n-grams—is rooted in efficiency. The latter requires computing n-gram embeddings for every n-gram in the input text and every document in the reference corpus during DJ SEARCH, which is computationally infeasible. Ideally, a good transport-inspired distance should approximate the Euclidean distance between the embeddings of two n-grams.
>
> To evaluate this, we computed the Pearson correlation between our implementations of WMD, relaxed WMD, supervised WMD, and Sinkhorn distance with the Euclidean distance between the embeddings of two n-grams to assess how closely they correlate.
>
> |         | Our MWD        | Relaxed MWD    | Supervised MWD | Sinkhorn distance |
> |---------|----------------|----------------|----------------|-------------------|
> | r-value | 0.842          | 0.823          | 0.716          | 0.808             |
> | p-value | 5.7 * 10^{-11} | 4.1 * 10^{-10} | 8.3 * 10^{-4}  | 9.1 * 10^{-8}     |
>
> We found that all transport-inspired distances correlate highly with Euclidean distance, showing strong statistical significance. The slightly lower correlation with supervised MWD may be due to its training objective, which focuses on distinguishing between different categories of documents.

---

> ### Author Response · Authors · 2024-11-29
>
> ## Re: Euclidean/Sinkhorn-based Distance
>
> As mentioned in the original draft, the motivation for using transport-inspired distances—which combine word embedding distances for each n-gram's words rather than directly computing the Euclidean distance between embeddings of two n-grams—is rooted in efficiency. The latter requires computing n-gram embeddings for every n-gram in the input text and every document in the reference corpus during DJ SEARCH, which is computationally infeasible. We also experimented with Sinkhorn distance alongside various versions of Word Mover’s Distance; please see the response above.
>
> ## Re: Reference Copus
>
> At the time of our experiments, RedPajama was the largest open-source pre-training corpus available. Since then, newer large-scale pre-training corpora, such as DOLMA [1], have been released and incorporated into the infini-gram search API. Additionally, RefinedWeb and Common Corpus from PleIAs [3] have also become available, and we plan to integrate them into the infini-gram search API.
>
> Building a high-quality and high-coverage pre-training corpus remains an important research question that the community is actively addressing with notable progress. We will continue to monitor advancements and integrate new pretraining corpora into DJ SEARCH as part of the reference corpus. Additionally, one of our future directions is to create our own large-scale, reliable reference corpus with fast search capability.
>
>
> [1] Huang, Gao et al. “Supervised Word Mover's Distance.” Neural Information Processing Systems (2016).
>
> [2] Cuturi, Marco. “Sinkhorn Distances: Lightspeed Computation of Optimal Transport.” Neural Information Processing Systems (2013).
>
> [3] PleIAs. “PleIAs/Common_corpus · Datasets at Hugging Face.” PleIAs/common_corpus · Datasets at Hugging Face. Accessed November 26, 2024. https://huggingface.co/datasets/PleIAs/common_corpus.

---

> > ### Comment · Reviewer_Kcvs · 2024-11-30
> > **Appreciate the effort in addressing even the smallest suggestions to improve work**
> >
> > Dear authors,
> >
> > Thank you for conducting additional experiments to address my suggestions on runtime and other similarity algorithms. Please include these results in the Appendix if possible, as these can make readers appreciate your contribution, particularly how fast the DJ Search (with the help of Infinigram) can be even with a large reference corpus such as RedPajama.
> >
> > My score remains favorable (8) based on the level of completeness of the paper and especially with the technicality of DJ Search and its simplicity and effectiveness for detecting AI-generated texts. While other reviewers have raised important points on measuring creativity, I have to acknowledge that this falls outside my core expertise, hence my confidence score.

---

### Author Response · Authors · 2024-11-26
**General Response**

We thank the reviewers for their thoughtful feedback and constructive comments. We particularly appreciate the encouraging remarks, such as the “simple and easy to understand” idea, the “level of completeness in terms of depth of experiments, discussion coverage, and insight” (Kcvs), the “multifaceted and multiangled extension of discussions” (5RYT), the “solid contribution to the field with practical, implementable algorithms” (ZD3D), the “great structure and organization of the paper,” and the “very valuable contribution from an engineering standpoint” (8K9m).

We will address some common concerns in the general response and respond to the remaining points from each reviewer individually below. Additionally, we have updated the paper draft based on the reviewers’ feedback.

---

> ### Author Response · Authors · 2024-11-26
> **Compare with Existing Human Evaluation-based Creativity Metrics**
>
> Several reviewers suggested comparing our CREATIVITY INDEX with existing human evaluation-based creativity measures. A recent study by Chakrabarty et al. [1] evaluates the creativity of stories written by LLMs and professional human authors through expert evaluations based on a carefully designed rubric inspired by the Torrance Test of Creative Thinking (TTCT). This work introduces the Torrance Test of Creative Writing (TTCW), which consists of 14 binary questions grouped into four dimensions: Fluency, Flexibility, Originality, and Elaboration. To conduct the evaluation, ten expert creative writers were recruited to evaluate 48 stories—written either by LLMs or published in The New York Times,—using the TTCW rubric.
>
> For each story, three expert evaluators are tasked with answering 14 binary questions, each designed to quantify a specific dimension of creativity. For example, within the Fluency dimension, "Understandability & Coherence" is assessed by asking, “Do the different elements of the story work together to form a unified, engaging, and satisfying whole?” In the Flexibility dimension, "Emotional Flexibility" is evaluated with the question, “Does the story achieve a good balance between interiority and exteriority, in a way that feels emotionally flexible?” Similarly, within the Originality dimension, "Originality in Thought" is measured by asking, “Is the story an original piece of writing without any clichés?” For each of the 14 binary questions, the final answer is determined by majority vote among the expert evaluators.
>
> To evaluate the correlation between our CREATIVITY INDEX and Chakrabarty et al.'s TTCW, we compute the CREATIVITY INDEX for each of the 48 stories. For each binary question in the TTCT (e.g., "Understandability & Coherence" or "Emotional Flexibility"), we calculate the point biserial correlation between the CREATIVITY INDEX values and the corresponding binary expert ratings for that question across all 48 stories, as shown below.
>
> |  | Emotional Flexibility | Narrative Pacing | Structural Flexibility | Originality in Thought | Scene vs Summary | Rhetorical Complexity | Language Proficiency | World Building and Setting | Narrative Ending | Originality in Form and Structure | Understandability and Coherence | Perspective and Voice Flexibility | Originality in Theme and Content | Character Development |
> |-----------|-----------------------|------------------|------------------------|------------------------|------------------|-----------------------|----------------------|----------------------------|------------------|-----------------------------------|---------------------------------|-----------------------------------|----------------------------------|-----------------------|
> | r-value   | 0.6826                | 0.6524           | 0.5968                 | 0.5968                 | 0.5812           | 0.5810                | 0.5702               | 0.5170                     | 0.4871           | 0.4626                            | 0.4495                          | 0.4467                            | 0.3582                           | 0.2299                |
> | p-value   | 0.0028                | 0.0051           | 0.0135                 | 0.0135                 | 0.0135           | 0.0135                | 0.0204               | 0.0426                     | 0.0617           | 0.0816                            | 0.0941                          | 0.0969                            | 0.1231                           | 0.1453                |
>
>
> We found that the CREATIVITY INDEX positively correlates with all dimensions of TTCW, showing particularly strong correlations with high statistical significance in dimensions such as “Emotional Flexibility,” “Narrative Pacing,” “Structural Flexibility,” and “Originality in Thought.”
>
> Quantifying creativity precisely remains a challenging research question, as different works adopt varying definitions of creativity [2, 3, 4]. Even when using the same carefully designed rubric, such as in TTCW, expert annotations do not always agree with each other due to subjectivity. In TTCW, the Fleiss' 𝜅 agreement across expert annotations ranges from 0.27 to 0.66, indicating low to moderate agreement. The primary goal of our paper is to explore the differences between seemingly amazing machine text and professional human writing by quantifying some aspect of linguistic creativity through the lens of n-gram comparisons with web text. While our approach provides valuable insights, it alone might not be a definitive measure of creativity or capture all dimensions of creativity, and is therefore complementary to existing creativity measurement methods.

---

> > ### Comment · Reviewer_8K9m · 2024-11-27
> > **Thank you for this rebuttal**
> >
> > I appreciate the rebuttal. I read the updated draft now and I wish the authors were thoughtful to admit all these before submitting. If you submitted the current draft with all the limitations of CREATIVITY INDEX I don't think the reviews would have been favourable. The language in the paper had a lot of overclaims/wrong claims and honestly a lot of assumptions. As it goes with the ethics of doing good research please think through carefully before saying something. Creativity is a human centered trait and not something that you can quantify with a number.
> >
> > Regarding correlation I don't think you can draw much inference. Further its really interesting here that out of the 6 tests where the p-value is not significant 2 are Originality related tests, which is literally what your metric is built to capture or based on
> >
> > Narrative Ending (p=0.0617)
> > Originality in Form and Structure (p=0.0816)
> > Understandability and Coherence (p=0.0941)
> > Perspective and Voice Flexibility (p=0.0969)
> > Originality in Theme and Content (p=0.1231)
> > Character Development (p=0.1453)
> >
> > Did you move New Yorker paragraphs from RedPajama before running the experiments ? I just look at Chakrabarty et al's github repo and noticed these two stories are in RedPajama so i hope you took that into account when calculation
> >
> > 1) https://www.newyorker.com/books/flash-fiction/the-facade-renovation-thats-going-well
> > 2) https://www.newyorker.com/books/flash-fiction/the-kingdom-that-failed
> >
> > In general you should put these results in your paper so that the community can decide whether to adopt to your metric or not.

---

> > > ### Author Response · Authors · 2024-12-02
> > >
> > > ## Re: Quantifying Linguistic Creativity
> > >
> > > On one hand, we agree with the 8k9m that creativity is an inherently challenging concept to successfully quantify. On the other hand we respectfully disagree with the reviewer that researchers must not make an attempt at quantifying at least some aspects of linguistic creativity, especially in the context of people at large giving potentially out-of-proportion credits to language models’ seemingly creative capacity. It is important to understand the underlying mechanism by quantifying where that perceived creativity comes from, and how we can make a step toward quantitatively measuring it.
> > >
> > > ## Re: Correlation with Previous Human-evaluation Based Creativity Measure
> > >
> > > Different prior works adopt varying definitions of creativity [2, 3, 4]. Even when using the same carefully designed rubric to measure creativity, such as in TTCW, expert annotations often do not agree with each other due to subjectivity. In TTCW, the Fleiss' 𝜅 agreement across expert annotations ranges from 0.27 to 0.66, indicating low to moderate agreement.
> > >
> > >
> > > The primary goal of this work is to understand the differences between seemingly amazing machine text and professional human writing by quantifying some aspect of linguistic creativity through the lens of n-gram comparisons with web text. While the proposed CREATIVITY INDEX provides valuable insights for this purpose, it is reasonable that this index correlates to varying degrees with other dimensions of creativity defined by previous human-evaluation based creativity measure. The motivation behind the CREATIVITY INDEX is to analyze LLMs’ perceived creativity through the lens of n-gram comparisons with web text. Comprehensive quantification of creativity, however, lies beyond the primary scope of this work.
> > >
> > > Additionally, we confirm that we performed deduplication and removed New Yorker paragraphs from RedPajama before running the experiments reported above.
> > >
> > > [1] Chakrabarty, Tuhin et al. “Art or Artifice? Large Language Models and the False Promise of Creativity.” Proceedings of the CHI Conference on Human Factors in Computing Systems (2023): n. Pag.
> > >
> > > [2] EL-MURAD, JAAFAR, and DOUGLAS C. WEST. “The Definition and Measurement of Creativity: What Do We Know?” Journal of Advertising Research 44.2 (2004): 188–201. Web.
> > >
> > > [3] Sternberg, Robert J., and Todd I. Lubart. “An Investment Theory of Creativity and Its Development.” Human Development, vol. 34, no. 1, 1991, pp. 1–31. JSTOR, http://www.jstor.org/stable/26767348. Accessed 29 Nov. 2024.
> > >
> > > [4] Boden, M. “Dimensions of creativity.” The Journal of Aesthetic Education 30 (1996): 120.

---

> > > > ### Comment · Reviewer_8K9m · 2024-12-02
> > > > **Comprehensive quantification of creativity, however, lies beyond the primary scope of this work ?**
> > > >
> > > > If comprehensive quantification of creativity, however, lies beyond the primary scope of this work, then do not add clickbait titles and metric name. A metric that is called CREATIVITY INDEX should measure Creativity comprehensively.
> > > >
> > > > I just checked the TTCW paper and it has a moderate average agreement (which is quite good for subjective tasks) it seems while you didn't even do a human evaluation. So I am not sure your point holds
> > > >
> > > > CREATIVITY CANNOT BE MEASURED BY NGRAM COMPOSITION. I have pointed it through many examples and how your metric systematically puts simple evocative writing at a disadvantage by calling it less creative (which you haven't acknowledged)
> > > > [Example from George Saunders and other examples i put in my original review]. Ofcourse as authors you don't understand the multifaceted aspect of Creativity which led you to say things like "Hemingway is more creative than people publishing books in 2023"
> > > >
> > > > The paper should not be judged because you optimized a computationally expensive search algorithm using DP. It should be judged on the claims it makes and the premise. Many of which is fundamentally wrong
> > > >
> > > > And if you are throwing random citations , here is a citation for one of the highest cited Creativity Evaluation paper
> > > >
> > > > **Social psychology of creativity: A consensual assessment technique.** which states that Creativity is best judged by experts in the field.
> > > >
> > > > Its unfortunate that there needs to be so much back and forth. I am going to stop here. You have favorable reviews from 2 reviewers so I guess it doesn't matter. If I were to make recommendations to AC , I would not be in favor of accepting this paper in its current form. Good luck

---

> > > > > ### Author Response · Authors · 2024-12-02
> > > > >
> > > > > ## Re: Flaws in CREATIVITY INDEX
> > > > >
> > > > > We computed the CREATIVITY INDEX for the George Saunders example provided by reviewer 8K9m and its LLM-generated counterpart. Across all the LLMs studied in the paper, George Saunders' writing consistently exhibited a much higher CREATIVITY INDEX than any LLMs, consistent with the main findings in our paper. For example:
> > > > >
> > > > > **Context**: One Saturday morning, the man who had actually killed the former premier went to his dead uncle’s garage,
> > > > >
> > > > > **ChatGPT Generated Paragraph**: and opened up the old wooden chest hidden in the corner. As he flipped open the lid, a rush of memories flooded back to him. It had been years since he had last seen the contents of the chest - reminders of a long-forgotten past that he had tried to bury deep within his soul. Among the items in the chest was a tattered journal, filled with cryptic entries and codes that only he could decipher. The journal held the key to his uncle's most prized possession - a map that led to a hidden treasure buried deep in the woods. The man's heart raced with excitement as he realized that this was his chance to finally uncover the secrets of his uncle's mysterious past. Without hesitation, he grabbed the journal and map, and made his way into the forest. The trees loomed ominously overhead, casting eerie shadows on the forest floor. As he followed the map's directions, he could feel the weight of his guilt and regret lifting off his shoulders with each step. The treasure hunt was becoming more than just a search for riches - it was a journey of self-discovery and redemption.
> > > > >
> > > > > The CREATIVITY INDEX of George Saunders's writing is 60.9, while that of ChatGPT's generation is 47.9. The CREATIVITY INDEX of George Saunders's writing is 27.1% higher than that of ChatGPT.
> > > > >
> > > > > **LLaMA 2 Chat Generated Paragraph**: One Saturday morning, the man who had actually killed the former premier went to his dead uncle's garage. He had been avoiding this place for years, ever since his uncle's death in a tragic accident involving a faulty lawnmower. But today, he felt compelled to come here, to confront the memories and secrets that lay hidden within these walls. As he entered the dimly lit space, the smell of gasoline and oil filled his nostrils, transporting him back to countless hours spent tinkering with engines and working on cars with his uncle. He walked over to the workbench where his uncle used to sit, surrounded by tools and parts scattered all around. The man's eyes scanned the room, taking in every detail, every memory etched into the surface of things. There was the old engine block his uncle had been working on when he died, still covered in grease and oil. Next to it stood an assortment of wrenches and sockets, each one bearing witness to countless hours of hard labor and sweat. And then there were the photographs on the wall, depicting a life well-lived, full of love and laughter. The man couldn't help but feel a pang of regret as he gazed upon these images, knowing that his own actions had ripped apart the very fabric of this family.
> > > > >
> > > > > The CREATIVITY of LLaMA 2 Chat is 46.6. The CREATIVITY INDEX of George Saunders's writing is 30.7% higher than that of LLaMA 2 Chat.
> > > > >
> > > > > The higher CREATIVITY INDEX reported by reviewer 8K9m for AI-generated text is likely due to the specific LLM they used. As mentioned in the original draft, “the computation of the CREATIVITY INDEX is constrained by the reference corpus used for DJ SEARCH.” In our demo, we used RedPajama as the reference corpus, which is suitable for analyzing LLMs primarily pre-trained on web data available before the cutoff date of the RedPajama corpus, such as GPT-3.5-turbo, LLaMA 2 Chat, Tulu 2, and OLMo Instruct. We explicitly mention this in both the original draft and the disclaimer on our demo page.
> > > > >
> > > > > At the time of our experiments, RedPajama was the largest open-source pre-training corpus available. Since then, newer large-scale pre-training corpora, such as DOLMA, have been released and incorporated into the infini-gram search API. Additionally, RefinedWeb and Common Corpus from PleIAs have also become available, and we plan to integrate them into the infini-gram search API to support the analysis of newer LLMs, such as GPT-4 and LLaMA 3 Chat. We will continue to monitor advancements and integrate new pretraining corpora into DJ SEARCH as part of the reference corpus.

---

> > > > > > ### Comment · Reviewer_8K9m · 2024-12-02
> > > > > > **Acknowledgement**
> > > > > >
> > > > > > Fair enough. My entire point is attribution to pre-training is great but you can’t use it to justify claims about  individual creativity (especially humans). And while you did mention some points about how your metric can work for newer models I think this goes on to prove that without availability of pre-training data we cannot make claims about  creativity of any LLM generated paragraph.
> > > > > >
> > > > > >  I generated it using Claude3.5Sonnet. The issue is also that people are no longer using Llama2Chat or GPT3.5turbo and its not your fault and neither you should be penalized for it.
> > > > > >
> > > > > > I am increasing my score to 6. Please add the disclaimer that Creativity Index might be inflated for any modern LLM (like GPT4, Claude, Llama3 etc)

---

> ### Comment · Reviewer_8K9m · 2024-11-27
> **Fundamental Flaws in your metric**
>
> Now coming to "Linguistic Creativity" . What is its definition ? Can you cite it ? What do you mean by it? From what I understand there needs to be certain degree of appropriateness. LLMs generate a hell lot of awkward phrases
>
> Now here is why I think your metric has fundamental flaws. I took an excerpt from a George Saunders Flash Fiction that was published on August 29 2024 after Red Pajamas cut off data
>
> Original : One Saturday morning, the man who had actually killed the former premier went to his dead uncle’s garage, where he’d hidden the red car, started it up, then sat behind the wheel, thinking. Once already, in the name of goodness, he’d sullied himself with murder. Shouldn’t he at least try to kill this new, even more evil, premier? Otherwise, what had it all been about?Killing the new premier would entail killing another set of unlucky diners. For him: more panic attacks, fresh nightmares. This time, if he succeeded, he would at least take some credit. Good God, yes, he would: no more silent suffering in anonymity. Afterward, he would climb up on the hood of his red car and, addressing the people, propose a new movement that would eschew violence and rule by justice and gentleness forever. Also firmness, of course. A merciful firmness the people would instantly respect.Or he could forget the whole thing, go back to being the man he’d been before the first killing: a frightened, passive nobody.This thought made him sicker than the thought of another round of screaming and crunching.He leaped out of the still running red car, threw the garage door open, and, feeling alive and decisive, stood blinking in the morning sunlight, readying himself for his task.
>
>
> LLM Generated Paragraph with same content details : In the crimson Mazda's cocoon, he shudders as phantom screams pierce his consciousness—not his own, but those of the former premier, whose last breath had tasted of copper and surprise. His fingers dance across the dashboard, muscle memory recalling how easily the garrote had slipped around his target's throat, smooth as silk until it wasn't. The garage's stale air, thick with motor oil and his uncle's lingering cigarette ghost, feels like a confessional booth as he weighs his soul against the current premier's sins. Every sunrise brings fresh headlines of corruption that make his predecessor seem saintly, yet each night drags him through a carousel of terrors—dreams where his victims' faces morph into his own, where blood flows upward like reverse rain. The prospect of retreating into anonymous mediocrity nauseates him more than the memories; he's tasted the intoxicating wine of consequence now, sampled the heady bouquet of being someone who matters. His manifesto sits heavy in his breast pocket, each word a bullet waiting to reshape reality. The car door flies open with the force of destiny itself, and he emerges transformed—no longer a mere assassin, but a revolutionary being born in his dead uncle's garage, ready to paint his name across history in whatever shade of red fate demands.
>
>
> Now both are written on the exact same content. I ran some quick experiments using Infinigram ( 5<=n<=11). The scores are Original George Saunders written Paragraph has a CREATIVITY INDEX of 59.5 while LLM generated paragraph has a CREATIVITY INDEX of 73.67 (I might be a little bit off with the numbers but these are close approximates)
>
> Now I find it hilarious because George Saunders is one of the biggest writers alive (https://en.wikipedia.org/wiki/George_Saunders)
>
> In the LLM generation these expressions are very very awkward (makes very little sense)
> - garage's stale air, thick with motor oil and his uncle's lingering cigarette ghost, feels like a confessional booth (???)
> - last breath had tasted of copper and surprise (???)
> - each night drags him through a carousel of terrors (???)
> - blood flows upward like reverse rain (???)
> - tasted the intoxicating wine of consequence now, sampled the heady bouquet (???)
> - each word a bullet waiting to reshape reality (???)
> - ready to paint his name across history in whatever shade of red fate demands. (???)
>
> I don't think these qualify as linguistically creative and now your metric has claim AI to be better than an expert
>
> This is exactly what i feared because your metric will force people to call LLMs more creative because they produce fancy ngrams with no overlap in pre-training (even though this was not your intention). Memorization is interesting to study but when you use it to claim anything with real humans you have to understand the consequences.
>
> How do you justify this ?

---

> > ### Comment · Reviewer_8K9m · 2024-11-27
> > **Thank you for acknowledging you metric only works on fluent and coherent text**
> >
> > Your revision says "Third, CREATIVITY INDEX assumes that the input text is of sufficient quality, as our study focuses on outputs from recent LLMs that are already fluent and coherent. For less refined texts, our metric can be complemented with standard automatic quality measures, such as fluency classifiers or perplexity-based evaluations, to provide a more comprehensive assessment."
> >
> > This is a bigger limitation. Now say people adapt your metric. But any leaderboard submission can be gamed by inserting 2-3 garbled sentence with all novel ngrams. Would it not inflate the CREATIVITY INDEX?
> >
> > I think you should rename your metric. It should not be CREATIVITY INDEX coz its not really measuring Creativity. Call it NOVELTY INDEX

---

> ### Comment · Reviewer_ZD3D · 2024-11-27
> **Another reviewer is not convinced by this reviewers critique**
>
> This is kind of a bogus critique in my opinion. In regards to "gaming" benchmarks, that's just par for the course for any kind of benchmark outside of LLM-as-judge or chatbot arena.
>
> I think that there are many notions of "creativity" or "diversity", and nearly all of them will have pathological cases, including stuff worse than the case you outlined here.
>
> More benchmarks, even with some issues, are better than less.
>
> I will maintain my high score. This is great work.

---

> > ### Comment · Reviewer_8K9m · 2024-11-27
> > **Stick to your own opinion**
> >
> > You are entitled to your opinion. My points are very clearly made and supported with evidence instead of Ad hominem claims. And we are all trying to inform the AC to make a better decision.
> >
> > "I think that there are many notions of "creativity" or "diversity", and nearly all of them will have pathological cases, including stuff worse than the case you outlined here." -----> Absolutely no. Research doesn't depend on vibe or notions but on anecdotal evidence. If you were informed by literature you would not equate creativity and diversity. They are absolutely different things.

---

### Meta-Review · Area_Chair_mhNr · 2024-12-15

**Metareview:**

All the reviewers unanimously appreciated the paper. I am also glad to see significant interactions between authors and reviewers, which made many things clear. I request the authors to incorporate the discussions into the final version of the paper. In particular, please take into account the following points:

1. Reviewer 8K9m made major criticism regarding the overclaiming of the statements and lack of proper comparison in the paper. I suggest the authors to please rephrase those statements and make appropriate adjustments.

2. Other reviewers also repeatedly mentioned the lack of baselines for comparison. Creativity is highly subjective. The authors should consider assessing and comparing the proposed metric from every angle.

3. The authors may test this metric on languages with different n-gram distributions or morphological complexity.

**Additional Comments On Reviewer Discussion:**

This seems to be a perfect example of how author-reviewer interaction should take place. I really appreciate all the reviewers and authors. In fact, in some cases, authors delayed responding to the comments and the reviewers had to ask the authors for the rebuttal. I really enjoyed reading the entire discourse.

In general, the major criticism pointed out by almost all the reviewers is the lack of comparison and over-claims here and there in the paper. The authors attempted to address some of these comments and convinced the reviewers. Overall, it is a good paper.

---

### Decision · Program_Chairs · 2025-01-22

Accept (Oral)